# FORMALIZING LEARNING FROM LANGUAGE FEEDBACK WITH PROVABLE GUARANTEES

## ABSTRACT

Interactively learning from observation and language feedback is an increasingly studied area driven by the emergence of large language model (LLM) agents. While impressive empirical demonstrations have been shown, so far a principled framing of these decision problems remains lacking. In this paper, we formalize the Learning from Language Feedback (LLF) problem, assert sufficient assumptions to enable learning despite latent rewards, and introduce *transfer eluder dimension* as a measure to characterize the hardness of LLF problems. We formalize the intuition that information in the feedback governs the learning complexity of LLF problems. We demonstrate cases where learning from rich language feedback can be exponentially faster than learning from reward. We develop a no-regret algorithm, called HELiX, that provably solves LLF problems through sequential interactions, with performance guarantees that scale with the transfer eluder dimension of the problem. Across several empirical domains, we show that HELiX performs well even when repeatedly prompting LLMs does not work reliably. Our contributions mark an important step towards designing principled interactive learning algorithms from generic language feedback.

## 1 INTRODUCTION

Large language models (LLMs) have reshaped the landscape of how machines learn and interact with the world across a wide range of tasks (Bommasani et al., 2021; BIG-bench authors, 2023; Anil et al., 2024; Hurst et al., 2024; Jaech et al., 2024; Guo et al., 2025; Yamada et al., 2025). Trained on large corpra of web data, these models can interact with the world through natural language, opening up new settings for sequential decision-making problems. Unlike traditional sequential decision-making approaches where agents learn from scalar reward signals (Sutton & Barto, 2018), LLM can act as agents that interpret and reason with natural language feedback such as critique (Du et al., 2023; Akyürek et al., 2023a), guidance (Branavan et al., 2012; Harrison et al., 2017; Scheurer et al., 2023; Nie et al., 2023; Fu et al., 2024; Wei et al., 2024; Cheng et al., 2024), or detailed explanations (Andreas et al., 2017; Chen et al., 2023; Cheng et al., 2023).

Consider an LLM agent that produces a summary of a story and receives feedback: "The summary is mostly accurate, but it overlooks the main character's motivation." Such feedback conveys notably richer information than a numerical score, e.g., 0.7 out of 1, as it identifies a specific flaw and suggests a direction for improvement. With LLMs' abilities to understand and respond in natural language Touvron et al. (2023), such feedback can be used to drastically increase learning efficiency. This represents a fundamental shift in how AI systems can learn through continuous, rich interactions beyond rewards only (Silver & Sutton, 2025). Despite early works on this topic pre-LLM (Gauthier & Mordatch, 2016; Andreas, 2022) and promising recent empirical results in utilizing language feedback for sequential decision-making (Liu et al., 2023; Chen et al., 2024; Xie et al., 2024), a rigorous theoretical framework remains lacking.

We introduce a formal mathematical framework of Learning from Language Feedback (LLF) in sequential decision making. The LLF paradigm was introduced in (Cheng et al., 2023) as an interface to benchmark LLM agents' ability to learn from text feedback in lieu of numerical reward. However, it is unclear when LLF is feasible or whether it is harder to solve than the more traditional reward-aware bandit setting. Intuitively, one might think language feedback can provide more information to help learning. Indeed, people have empirically found constructive feedback to be more effective

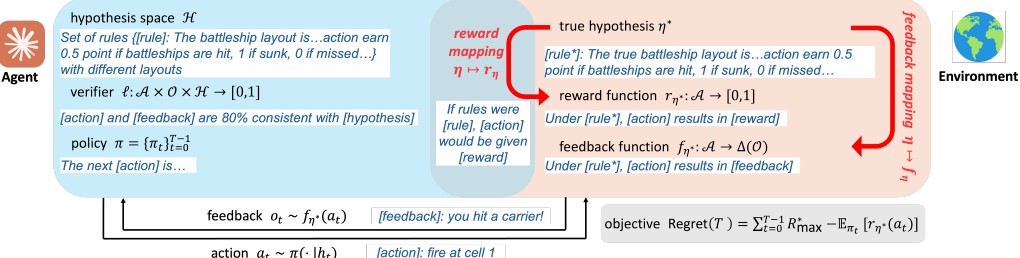

Figure 1: The LLF setup using battleship as a concrete example. The environment has a hypothesis $\eta^*$ representable via text tokens unknown to the agent. Reward as a function of $\eta^*$ is latent and used only to benchmark the agent via regret to an optimal policy. Feedback as a function of $\eta^*$ is observed by the agent. Three ingredients are sufficient for no-regret learning: feedback is *unbiased* (Assumption 3), agent can interpret feedback (Assumption 2), and agent considers hypotheses $\mathcal{H}$ including $\eta^*$ (precursor to Assumption 1).

for LLM agents to learn from than conveying reward alone in words (Mu et al., 2022; Liu et al., 2024; Zhong et al., 2024; Xie et al., 2024). But feedback can also mislead agents. The complexity and generality of language make it difficult to formally quantify the effect of language feedback.

For general language feedback, can we precisely define helpful and unhelpful feedback? Can we capture the complexity of LLF based on the information in the feedback, and does helpful feedback indeed imply a lower problem complexity? Can we design a provably correct algorithm that learns solely from language? The goal of this paper is to provide constructive answers to all these questions:

**Language feedback can be formalized through hypothesis and verifier.** To work with the generality of language, we rely on the concept of hypothesis testing and elimination in machine learning (De Jong et al., 1993; Lehmann & Romano, 2022), except with hypotheses that can be expressed in words. We formalize the interface in which agents sequentially interact while reasoning with feedback produced by an underlying hypothesis (summarized by Fig. 1). We also define a verifier which evaluates the semantic consistency between candidate hypotheses and observed feedback. Through the notion of hypothesis and verifier, we give a precise definition of informative feedback and establish conditions such that LLF is feasible and can be efficiently solved.

**Hardness of LLF is determined by information in feedback.** We capture the learning difficulty with a new notion of complexity based on eluder dimension (Russo & Van Roy, 2013), which we call *transfer eluder dimension*. This complexity measure captures how efficiently language feedback can reduce uncertainty about rewards. While many existing settings consider feedback in place of scalar rewards (Wang et al., 2003; Kocák et al., 2014; Bartók et al., 2014; Fürnkranz et al., 2012), they commonly assume that the only useful information the feedback encodes is the underlying reward and focuses on decoding it accurately. As an example, IGL (Xie et al., 2021) posits a decoder capable of extracting reward estimates from a rich feedback vector, and treats the remaining components as distractions. In contrast, our work emphasizes on the importance of extracting useful learning signals other than reward, and we show regimes where LLF is strictly *easier* than reward-based learning.

**LLF can provably have no regret.** We develop HELiX, a provably efficient algorithm for LLF. We prove that HELiX achieves a regret bound that scales gracefully with the transfer eluder dimension and time horizon $T$, establishing a formal connection between no-regret learning and language feedback. Crucially, our analysis shows that in certain environments, HELiX can be *exponentially* more efficient than learning from reward alone. We introduce a meta-algorithm that enables LLMs to perform inference-time exploration and exploitation using HELiX, inspired by how thinking tokens are used in large reasoning models (LRMs) (Guo et al., 2025). We empirically validate the efficacy of our implementation on Wordle, Battleship and Minesweeper. We show that HELiX and its variants consistently outperform in-context learning LLM baselines. Altogether, our work contributes a principled framework for understanding and designing learning agents guided by language.

## 2 RELATED WORK

While using LLMs for general problem solving has been studied for a long time (Xie et al., 2022a; Guo et al., 2024; Akyürek et al., 2023b), relatively fewer prior works studied the use of LLMs for sequential decision-making. There are roughly two routes to improving the agent's performance with language feedback. One is to directly deploy LLMs as agents in decision-making problems by incorporating feedback into subsequent prompts or an external memory buffer (Yao et al., 2023; Brooks et al., 2023; Shinn et al., 2023; Wang et al., 2024; Krishnamurthy et al., 2024; Nie et al., 2024; Xi et al., 2025). Another route is to process this feedback and use it to finetune a model's weights (Chen et al., 2024; Scheurer et al., 2022; Raparthy et al., 2023; Lee et al., 2023; Qu et al., 2025). More recent work has investigated more sophisticated methods to improve exploration with LLMs, such as directly learning exploration behavior through supervised fine-tuning (Nie et al., 2024), preference-based learning (Tajwar et al., 2025), or reinforcement learning (Schmied et al., 2025), or prompting LLMs to mimic a perfect Bayesian learner (Arumugam & Griffiths, 2025). However, these results have been empirical.

We aim to bridge this gap by introducing a formal framework and guarantees for learning from language feedback. Our framework is closely related to multi-armed bandits (Lai & Robbins, 1985) and contextual bandits (Langford & Zhang, 2007). The class of algorithms that achieve diminishing long-term average reward are termed "no-regret algorithms" (Auer et al., 2002; Thompson, 1933; Russo et al., 2018). One widely adopted strategy relies on the "optimism in the face of uncertainty" principle. Our algorithm design follows the same spirit as UCB (Auer et al., 2002). A key difference is that our algorithm does not observe rewards at all, but instead rely on decoding information in the feedback through a verifier loss to construct the confidence set. A recent line of work utilizes UCB-like heuristics for LLM agents, but they either consider hypotheses as code that specifies an MDP (Tang et al., 2024), and/or assume that the agent observes the ground-truth numerical reward (Tang et al., 2024; N et al., 2024; Nie et al., 2024).

Another line of research has leveraged natural language as an auxiliary signal to improve learning in sequential decision-making. Early studies showed that agents can benefit from textual guidance, such as game manuals, to inform policies or features (Branavan et al., 2012). Subsequent approaches explored grounded language to shape behavior (Gauthier & Mordatch, 2016), guide exploration (Harrison et al., 2017), or learn from feedback (Andreas et al., 2017). More recently, LDD (Zhong et al., 2024) pre-trains agents on language-annotated demonstrations to learn environment dynamics, then fine-tunes with RL to improve sample efficiency and generalization. While these approaches show empirical success, they lack a formal framework and theoretical guarantees.

Beyond scalar rewards, many learning settings offer richer forms of feedback. Prior work has explored bandits with side observations (Wang et al., 2003; Kocák et al., 2014), partial monitoring (Bartók et al., 2014), and preference-based feedback (Fürnkranz et al., 2012). To characterize sample complexity in reward-aware RL, Russo & Van Roy (2013) introduces the eluder dimension. Our work extends this notion beyond reward learning (for a detailed discussion and illustration of the relationship of LLF to existing paradigms, see Fig. 3 in Appendix A).

## 3 FORMULATING LEARNING FROM LANGUAGE FEEDBACK

Our first contribution is to give a formal mathematical model to describe the LLF process (illustrated by Fig. 1) and introduce natural assumptions to frame the learning problem so that LLF can be rigorously studied. In what follows, we first define the interaction setup. Then we introduce the notion of text hypotheses for world modeling. Finally, we define the verifier to evaluate hypothesis-feedback consistency, which later gives a measure on the informativeness of feedback. These constructions provide a basis for studying LLF's learnability and analyzing regret in the next section.

### 3.1 FORMAL SETUP OF LLF

Let $\mathcal{T}$ be a finite set of tokens. We denote the set of all finite token sequences by $\mathcal{T}^+ = \cup_{k \geq 1} \mathcal{T}^k \cup \{\emptyset\}$, where $\mathcal{T}^k$ denotes the set of length-$K$ token sequences. There is a set $\mathcal{O} \subset \mathcal{T}^+$ of token sequences that we refer to as the *feedback* space. For an arbitrary set $\mathcal{X}$, we use $\Delta(\mathcal{X})$ to denote the set of all probability distributions with support on $\mathcal{X}$.

We define the problem of Learning from Language Feedback (LLF)[1] with a finite action set $\mathcal{A}$. At time step $t$, the agent interacts with the environment by executing an action $A_t \in \mathcal{A}$ and observing feedback $O_t \in \mathcal{O}$ sampled from a feedback distribution $f^* : \mathcal{A} \to \Delta(\mathcal{O})$; a reward $R_t = r^*(A_t)$ is incurred, based on a reward function $r^* : \mathcal{A} \to [0, 1]$, though $R_t$ is not revealed to the agent. Here we suppose the reward is generated by a deterministic function $r^*$; our results can be extended to stochastic rewards. A policy is a distribution on $\mathcal{A}$. We denote $\Pi = \Delta(\mathcal{A})$ and the agent's policy at time step $t$ for sampling $A_t$ as $\pi_t$. We measure the performance of the agent in the LLF setup by regret, which is defined as $\mathrm{Regret}(T) = \sum_{t=0}^{T-1} R_{\max}^* - \mathbb{E}_{\pi_t}[R_t]$, where $T$ is the total number time steps, $R_{\max}^* = \max_{a \in \mathcal{A}} r^*(a)$, and the expectation is taken over feedback randomness and the algorithm's inner randomization.

This setup is similar to a bandit problem, and the goal of the agent is to find actions that maximize the reward. However, unlike RL, here the agent *does not observe the rewards* $\{R_t\}$, and must learn to maximize the reward solely using natural language feedback $\{O_t\}$.

**Remark 1.** The setup above can be naturally extended to a contextual setting (an analogy of contextual bandit problems; please see Appendix D.2 for details), where the agent receives a context in each time step before taking an action. While the feedback in the context-less setting here may be viewed similar to a context, the main difference is that the optimal actions in the context-less setting do not change between iterations; on the other hand, in the contextual setting, the optimal actions in each time step depend on the context presented to the agent at that point.

## 3.2 Environment Model and Text Hypothesis

The environment in the LLF setup is defined by a feedback function $f^* : A \to \Delta(\mathcal{O})$ and a reward function $r^* : \mathcal{A} \to [0, 1]$. We suppose they are "parameterized" by some text description, which we call a *hypothesis*, belonging to a (possibly exponentially large) hypothesis space $\mathcal{H} \subset \mathcal{T}^+$. One can think of a hypothesis as describing the learning problem and mechanism of generating feedback in texts such as natural language or codes. For example, in a recommendation environment, a hypothesis can be a text description of a user's interests, e.g., "the user enjoys fantasy movies produced in the 21st century..."; in a video game environment, a hypothesis can describe the game's code logic, "`<rule of the game><inferred hidden game state><inferred reward mechanism>`". A hypothesis can also represent a finite-sized numerical array along with operations to decode it into reward and feedback. In short, a hypothesis is a sufficient text description of the learning problem such that the reward and the feedback functions can be fully determined.

With the hypothesis space $\mathcal{H}$, we model the feedback mechanism through a *feedback mapping* $\eta \mapsto f_\eta$ that maps each hypothesis $\eta \in \mathcal{H}$ to a *feedback function* $f_\eta : \mathcal{A} \to \Delta(\mathcal{O})$. Similarly, we model a *reward mapping* $\eta \mapsto r_\eta$ that maps a hypothesis $\eta \in \mathcal{H}$ to a *reward function* $r_\eta : \mathcal{A} \to [0, 1]$. We denote by $\eta^* \in \mathcal{H}$ the true hypothesis of the environment, and use shorthand $f^* = f_{\eta^*}$ and $r^* = r_{\eta^*}$. This construction is reminiscent of classical bandit settings where the reward function is parameterized, such as the linear case $r^*(a) = \phi(a)^\top \theta^*$ for some known feature map $\phi$ and unknown ground-truth parameter $\theta^*$. We generalize this by using the reward mapping $\eta \mapsto r_\eta$ as an analogue of the feature map and the hypothesis $\eta^*$ as the parameter. Following the convention in the literature, we assume that the parameterization, i.e., the reward mapping $\eta \mapsto r_\eta$, is *known* to the agent, but the parameter $\eta^*$ is *unknown*. See Fig. 1 for an overview.

**Assumption 1.** We assume that the agent has access to the reward mapping $r_\eta : \eta \mapsto r_\eta$.

In practice, the reward mapping can be implemented using an LLM to process a given hypothesis text, e.g., to tell whether an action is correct/incorrect (Zheng et al., 2023; Weng et al., 2023; Gu et al., 2024). We do not assume knowing the feedback mapping $\eta \mapsto f_\eta$, however, as precisely generating language feedback in practice is difficult.

## 3.3 Measuring Information in Feedback

Without any connection between feedback and reward, learning to minimize regret from feedback is provably impossible. Intuitively, for LLF to be feasible, language feedback must contain information

---

[1]In the original formulation in (Cheng et al., 2023), a problem context is given before learning to provide background to interpret feedback. We omit writing the problem context for simplicity but equivalently *assume that the agent can interpret the feedback through the verifier* that we will introduce later.

that can infer the solution, like reward, action rankings, or whether an action is optimal. To study LLF learnability, we need a way to quantify this information. Since it is impossible to enumerate all possible language feedback, we adopt a weak, implicit definition based on a sensing function.

We introduce the notion of a *verifier* to formalize information the agent can extract from feedback. The verifier represents a mechanism that assesses whether a hypothesis is consistent with observed feedback given to an action; for example, a verifier implemented by an LLM may rule out hypotheses that are semantically incompatible with feedback observations.

**Assumption 2** (Verifier). We assume that there is a verifier, which defines a loss $\ell : \mathcal{A} \times \mathcal{O} \times \mathcal{H} \to [0, 1]$, and the agent has access to the verifer through $\ell$. For any action $a \in \mathcal{A}$, feedback $o \in \mathcal{O}$ and hypothesis $\eta \in \mathcal{H}$, the value $\ell(a, o, \eta)$ quantifies how well $\eta$ aligns with the feedback on action $a$. If $\eta$ is consistent with $o$ on action $a$, then $\ell(a, o, \eta) = 0$; otherwise, it returns a non-zero penalty.

A concrete example may help ground this abstract assumption. Suppose the agent chooses an action $a$ corresponding to a text summary of a story, and receives feedback $o$ in the form of text critique, such as: "The summary is mostly accurate, but it misses an important detail about the main character's motivation." Suppose each hypothesis $\eta \in \mathcal{H}$ corresponds to a set of rubrics to judge summaries. A verifier must output a score $\ell(a, o, \eta)$. If a rubric $\eta$ implies that summaries should capture the main character's motivation, then $\ell(a, o, \eta) = 0$, indicating consistency. Otherwise, the loss value is positive. Such a verifier can be implemented by prompting an LLM to assess whether the feedback $o$ is consistent with applying rubric $\eta$ to the summary $a$.

The set of feedback-consistent hypotheses naturally captures information in the feedback. Ideally, feedback generated from $f_\eta(\cdot)$ should be self-consistent, i.e., $\mathbb{E}_{O \sim f_\eta(a)}[\ell(a, O, \eta)] = 0$ for all $a \in \mathcal{A}$ and $\eta \in \mathcal{H}$. However, in practice, both the feedback and the verifier may be noisy or imperfect and there may be some $a \in \mathcal{A}$ such that $\mathbb{E}_{O \sim f^*(a)}[\ell(a, O, \eta^*)] > 0$. To accommodate this potential noise while preserving learnability, we adopt a weaker assumption than self-consistency: although the feedback may be noisy, it is *unbiased* such that each hypothesis minimizes the expected verifier loss under its induced distribution.

**Assumption 3** (Unbiased Feedback). We say $f_\eta$ is unbiased, if for all $a \in \mathcal{A}$ and $\eta \in \mathcal{H}$, $\eta \in \arg\min_{\eta' \in \mathcal{H}} \mathbb{E}_{O \sim f_\eta(a)}[\ell(a, O, \eta')]$.

The notion of verifier can be used to formalize *semantic equivalence* among hypotheses. In natural language, many token sequences share the same underlying semantic meaning. For LLF, such distinctions are not meaningful and should not affect the learning outcome. This invariance can be captured by the verifier introduced above. We deem hypotheses as equivalent whenever they induce identical loss functions across all inputs. We use this to define the geometry of the hypothesis space.

**Definition 1** (Hypothesis Equivalence). We define the distance between two hypotheses $\eta, \eta' \in \mathcal{H}$ as $d_{\mathcal{H}}(\eta, \eta') := \sup_{a \in \mathcal{A}, o \in \mathcal{O}} |\ell(a, o, \eta) - \ell(a, o, \eta')|$. If $d_{\mathcal{H}}(\eta, \eta') = 0$, we say $\eta$ and $\eta'$ are *equivalent*.

This definition provides a criteria to determine the equivalence of hypotheses, as two hypotheses with zero distance are indistinguishable from the agent's perspective. In applications involving LLM-generated feedback, the loss function $\ell$ can be designed to reflect semantic similarity, e.g., by assigning similar values to outputs that are paraphrases of one another, based on token-level matching, embedding-based metrics, or LLM-prompted judgments (Wang & Yu, 2023; Chuang et al., 2022; Asai & Hajishirzi, 2020; Bubeck et al., 2023).

**Remark 2.** Readers familiar with reinforcement learning from human feedback (RLHF) or AI feedback (RLAIF) may wonder if such a loss structure is necessary. Indeed, one may alternatively define a scoring function $g : \mathcal{A} \times \mathcal{O} \to [0, 1]$ that directly evaluates an action-feedback pair and impose some relationships between the scoring function and the underlying reward. This construction is a special case to our framework, which we discuss in detail in Section 4.3.

## 4 LEARNABILITY AND PROVABLE ALGORITHM

Compared to numerical rewards, feedback can potentially carry more information. In LLF, to interpret this feedback and guide learning, the agent is equipped with: *1)* The verifier loss function $\ell$ and *2)* The reward mapping $\eta \mapsto r_\eta$. This structure reflects a central feature of LLF: the agent must reason over the hypothesis space $\mathcal{H}$ via the verifier to minimize regret of the hidden rewards.

But can an agent learn to maximize reward despite not observing it? For instance, if feedback does not convey useful information for problem solving, it is unrealistic to expect any learning to happen. On the other hand, if feedback directly reveals the optimal action, then the problem can be solved in two steps. Naturally, one would expect the learnability and complexity of LLF problems to depend on the information that feedback conveys. The goal of this section is to give natural structures and assumptions to the LLF setup that characterizes the difficulty of the learning problem.

## 4.1 TRANSFER ELUDER DIMENSION

To quantify information in the feedback, we propose a new complexity measure called *transfer eluder dimension* based on the eluder dimension (Russo & Van Roy, 2013) using the verifier in Section 3.3. At a high level, transfer eluder dimension characterizes how effectively information in the feedback reduces uncertainty about the unknown reward function. When it is small, a single piece of feedback carries a lot of information about the reward, which enables LLF to be much more efficient than learning from reward.

**Definition 2.** Define $\ell_\eta^{min}(a) := \min_{\eta'} \mathbb{E}_{O \sim f_\eta(a)}[\ell(a, O, \eta')]$. Given a verifier loss $\ell$, an action $a \in \mathcal{A}$ is $\epsilon$-*transfer dependent* on actions $\{a_1, \ldots, a_n\} \subset \mathcal{A}$ with respect to $\mathcal{H}$ if any pair of hypotheses $\eta, \eta' \in \mathcal{H}$ satisfying $\sum_{i=1}^n \left( \mathbb{E}_{o \sim f_{\eta'}(a_i)}[\ell(a_i, o, \eta)] - \ell_{\eta'}^{\min}(a_i) \right) \leq \epsilon^2$, also satisfies $|r_\eta(a) - r_{\eta'}(a)| \leq \epsilon$. Further, $a$ is $\epsilon$-*transfer independent* of $\{a_1, \ldots, a_n\}$ with respect to $\mathcal{H}$ if $a$ is not $\epsilon$-transfer dependent on $\{a_1, \ldots, a_n\}$.

This definition says that an action $a$ is transfer independent of $\{a_1, \ldots, a_n\}$ if two hypotheses that give similar feedback according to the verifier at $\{a_1, \ldots, a_n\}$ can differ significantly in their reward predictions at $a$. This differs from the dependency condition used in eluder dimension (Definition 4), which measures discrepancies in both the history and new observation using reward.

**Definition 3** (Transfer eluder dimension). The $\epsilon$-*transfer eluder dimension* $\dim_{TE}(\mathcal{H}, \ell, \epsilon)$ of $\mathcal{H}$ with respect to the verifier loss $\ell$ is the length $d$ of the longest sequence of elements in $\mathcal{A}$ such that, for some $\epsilon' \geq \epsilon$, every action element is $\epsilon'$-transfer independent of its predecessors.

Unlike the eluder dimension, transfer eluder dimension measures dependence based on two quantities: the verifier loss and the reward function. This extension allows us to capture information in the feedback relevant to reward learning. Later in Section 4.4, we will present a provable algorithm that attains a sublinear regret rate in LLF in terms of the transfer eluder dimension.

## 4.2 INFORMATIVE FEEDBACK REDUCES LEARNING COMPLEXITY EXPONENTIALLY

We discuss several example forms of feedback and compute the corresponding transfer eluder dimensions. The nature of feedback critically affects learning efficiency: uninformative feedback (e.g., random text) leads to infinite transfer eluder dimension, while some feedback can provide more information than reward and accelerate learning. For example, in a constraint satisfaction problem, feedback that reveals satisfied constraints can shrink the set of potentially true hypotheses. In the toy example below, reward-only learning requires exponential time ($2^L$), whereas the transfer eluder dimension is 1, so LLF has the potential for an exponential speed up.

**Example 1** (Bitwise feedback on 0-1 string). Consider an action set $\mathcal{A} = \{0, 1\}^L$. The space of hypotheses $\mathcal{H}$ contains all possible length-$L$ 0-1 strings. Each hypothesis $\eta$ contains a particular fixed target string $s(\eta)$ and the corresponding text instruction to provide reward and feedback about the target. The reward function $r_\eta$ corresponding to a hypothesis $\eta$ is such that $r(a) = 1$ if $a = s(\eta)$ and $r(a) = 0$ otherwise. In other words, rewards are sparse and every suboptimal arm incurs a regret of 1. Feedback to an action $a = (a_1, \ldots, a_L)$ is bitwise, which tells in words the correctness of each bit in the 0-1 string (i.e. whether $a_i = s_i$ for $s(\eta) = (s_1, \ldots, s_L)$. Equivalently, we can abstract the feedback as $f_\eta(a) = (\mathbb{1}\{a_i = s_i\})_{i=1}^L$ and define the loss function $\ell(a, o, \eta) = \frac{1}{L} \sum_{i=1}^L \mathbb{1}\{o_i \neq \mathbb{1}\{a_i = s_i\}\}$ to measure the discrepancy between the feedback and the correctness indicated by hypothesis $\eta$. For any $\epsilon < \frac{1}{L}$, the transfer eluder dimension $\dim_{TE}(\mathcal{H}, \ell, \epsilon) = 1$, as for any action $a'$, the expected loss $\mathbb{E}_{O \sim f_{\eta'}(a')}[\ell(a', O, \eta)] < \frac{1}{L}$ iff $\eta = \eta'$.

We can also use feedback to reveal information e.g. about the optimality of selected actions, improving directions, or explanation of mistakes.

**Example 2** (Reasoning steps)**.** Consider a math reasoning problem where one tries to construct a hidden sequence of $L$-step reasoning $a^* = (s_1^*, \ldots, s_L^*)$, where each $s_i \in \mathcal{S} \subset \mathcal{T}^+$ is a token sequence that represents a correct reasoning at step $i$, and $\mathcal{S}$ is a finite set of token sequences that represent possible reasoning steps. The action set $\mathcal{A} = \cup_{k=1}^{L}(\mathcal{T}^+)^k$ consists of all possible reasoning of $L$ steps. Each hypothesis represents a full solution to the problem and rubrics to critique partial answers with. Reward is 1 if all steps are correct and 0 otherwise. Below we show the transfer eluder dimension with $\epsilon < \frac{1}{2L}$ for different feedback (see Appendix C.4 for the exact forms of verifiers and proofs). We consider four feedback types, which corresponds to the reward, hindsight-negative, hindsight-positive, and future-positive feedback, respectively, in the LLF's feedback taxonomy proposed in (Cheng et al., 2023). Directly learning from rewards incurs exponential complexity, as the agent must enumerate all possible sequences. Feedback that identifies the first mistake enables stage-wise decomposition and yields exponential improvement in $L$, though each stage still requires brute-force search. If the feedback is more constructive, showing not only where the first mistake is but also how to correct for it, the problem complexity does not depend on $|\mathcal{S}|$. Finally, if the feedback tells the answer right away, the complexity becomes constant, as the agent can learn the solution immediately after one try.

| Feedback | $\dim_{TE}(\mathcal{H}, \ell, \epsilon)$ |
|---|---|
| 1. (reward) binary indicator of whether all steps are correct | $O(|\mathcal{S}|^L)$ |
| 2. (explanation) index of the first incorrect step | $O(|\mathcal{S}|L)$ |
| 3. (suggestion) give correction for the first mistake | $O(L)$ |
| 4. (demonstration) all the correct steps | $O(1)$ |

### 4.3 Learning from Feedback Is No Harder Than Learning from Reward

We have shown examples where the transfer eluder dimension is bounded and decreases as the feedback provides more information than reward. Here we prove the generality of this observation. Below we show that if feedback discriminates between rewards, then the transfer eluder dimension of LLF is no larger than the traditional eluder dimension of RL in Definition 4.

**Definition 4** (Eluder Dimension)**.** An action $a \in \mathcal{A}$ is $\epsilon$-dependent on actions $\{a_1, \ldots, a_n\} \subset \mathcal{A}$ with respect to a reward class $\mathcal{R}$ if any $r, r' \in \mathcal{R}$ satisfying $\sum_{i=1}^{n}(r(a_i) - r'(a_i))^2 \leq \epsilon^2$, also satisfies $|r(a) - r'(a)| \leq \epsilon$. Further, $a$ is $\epsilon$-independent of $\{a_1, \ldots, a_n\}$ if it is not $\epsilon$-dependent on $\{a_1, \ldots, a_n\}$. The $\epsilon$-eluder dimension $\dim_E(\mathcal{R}, \epsilon)$ of $\mathcal{R}$ is the length $d$ of the longest sequence of elements in $\mathcal{A}$ such that, for some $\epsilon' \geq \epsilon$, every action element is $\epsilon'$-independent of its predecessors.

First, by using the verifier, we define the statement "feedback discriminates between rewards".

**Definition 5** (Discriminative feedback)**.** The feedback function $f_\eta$ is *discriminative* of $r_\eta$ with respect to the verifier $\ell$ if there is $C_F > 0$ such that $\forall \eta' \in \mathcal{H}$, $a \in \mathcal{A}$, $|r_\eta(a) - r_{\eta'}(a)|^2 \leq C_F \mathbb{E}_{o \sim f_\eta(a)}[\ell(a, o, \eta') - \ell_\eta^{min}(a)]$. We say an LLF problem is *discriminative* if $(f^*, r^*, \ell)$ satisfies the above condition.

This definition states that the verifier can distinguish hypotheses based on feedback to the same extent as their reward differences. In other words, if two hypotheses differ in their corresponding rewards, then the verifier can tell they are different. Therefore, problems where feedback encodes the reward and verifier can decode it (e.g., classical RL) are subsumed as a special case of discriminative LLF. We discuss the relationship of LLF with discriminative feedback and IGL (Xie et al., 2022b) in Appendix A.

A discriminative feedback example is when the unobserved reward is a function of the feedback. Concretely, suppose $r_\eta(a) = \mathbb{E}_{o \sim f_\eta(a)}[g(a, o)]$ for some known $g : \mathcal{A} \times \mathcal{O} \to [0, 1]$. Note that the reward mapping $\eta \mapsto r_\eta$ is known, but the reward function itself is still hidden from the agent (since $\eta^*$ is unknown). Consider $\ell(a, o, \eta) := (g(a, o) - r_\eta(a))^2 = (g(a, o) - \mathbb{E}_{o' \sim f_\eta(a)}[g(a, o')])^2$. Then one can verify that $\eta \in \arg\min_{\eta' \in \mathcal{H}} \mathbb{E}_{o \sim f_\eta(a)}[\ell(a, o, \eta')]$ and show that this feedback-verifier pair is discriminative. (see Appendix C.3). In addition to this example, one can check that the forms of feedback used in Section 4.2 are discriminative too (see Appendix C.4). Discriminative feedback can contain information other than reward as shown in Section 4.2.

With this definition in place, we show that if feedback can discriminate rewards, the transfer eluder dimension is no larger than the eluder dimension for the reward class induced by $\mathcal{H}$.

---

**Algorithm 1** HELiX: Hypothesis Elimination using Language-informed Exploration

---

1: **Input** $\mathcal{A}, \mathcal{O}, T$, reward mapping $\eta \mapsto r_\eta$, verifier loss $\ell : \mathcal{A} \times \mathcal{O} \times \mathcal{H} \to [0, 1]$, confidence levels $\{\epsilon_t\}_{t=0}^{T-1}$
2: **Initialize** $t = 0$, $A_0 \sim \mathrm{Unif}(\mathcal{A})$, $\mathcal{H}_0 = \mathcal{H}$
3: **for** $t = 1, \ldots, T$ **do**
4:      observe $O_{t-1}$
5:      $\mathcal{H}_t \leftarrow \mathcal{H}_{t-1} \bigcap \{\eta \in \mathcal{H} : \frac{1}{t} \sum_i \ell(A_i, O_i, \eta) - \min_{\eta' \in \mathcal{H}} \frac{1}{t} \sum_i \ell(A_i, O_i, \eta') \leq \epsilon_t\}$
6:      $(\pi_p, \eta_p) \leftarrow \arg\min_{\pi \in \Pi} \max_{\eta \in \mathcal{H}_t} \left[ r_\eta(\pi_\eta) - r_\eta(\pi) \right]$
7:      **if** $r_{\eta_p}(\pi_{\eta_p}) - r_{\eta_p}(\pi_p) = 0$ **then**
8:          $A_t \sim \pi_p(\cdot)$     // Exploitation step: exploit if there is consensus
9:      **else**
10:         $(\pi_o, \eta_o) \leftarrow \arg\max_{\pi \in \Pi} \max_{\eta \in \mathcal{H}_t} r_\eta(\pi)$     // Exploration step: UCB-inspired
11:         $A_t \sim \pi_o(\cdot)$
12:      **end if**
13: **end for**

---

**Proposition 1.** *For discriminative LLF problems with $C_F$ as in Definition 5, it holds that $\dim_{TE}(\mathcal{H}, C_F\ell, \epsilon) \leq \dim_E(\mathcal{R}_{\mathcal{H}}, \epsilon)$, where $\mathcal{R}_{\mathcal{H}} = \{r_\eta : \eta \in \mathcal{H}\}$ is the effective reward class of $\mathcal{H}$.*

Proposition 1 implies that discriminative LLF problems are no harder than their reward-only counterparts, such as those solved by the standard UCB algorithm over the reward class $\mathcal{R}_{\mathcal{H}}$ using reward extracted from the language feedback by some LLM. It is important to note that general LLF problems are not necessarily discriminative. This separates LLF from existing frameworks such as IGL (Xie et al., 2021), as it allows LLF to handle cases where feedback contains much more *useful* information than reward. For instance, when feedback is not discriminative but reveals information about the optimal action, LLF captures the decrease in problem complexity compared to learning from reward, while the latter setting is vacuous for IGL.

### 4.4 HELiX ALGORITHM

To validate our characterization of learnability based on the transfer eluder dimension, we design a simple UCB-style algorithm, HELiX, outlined in Algorithm 1. HELiX uses feedback to guide exploration using the optimism principle. Given a hypothesis $\eta \in \mathcal{H}$, let $\pi_\eta$ denote its optimal policy. At step $t$, the algorithm maintains a confidence set $\mathcal{H}_t$ of hypotheses that remain approximately consistent with observed actions and feedback, as measured by cumulative verifier loss. The algorithm then identifies a hypothesis $\eta_o$ that achieve maximal optimal reward, and follows an optimal policy $\pi_o$ under this hypothesis. With a slight abuse of notation, we let $r_\eta(\pi) := \sum_{a \in \mathcal{A}} r_\eta(a)\pi(a)$ denote the expected reward of policy $\pi$. An additional design in HELiX compared to standard UCB is a stopping criterion. It checks for a consensus optimal action among all hypotheses in the confidence set. If the minimax regret $\min_{\pi \in \Pi} \max_{\eta \in \bar{\mathcal{H}}} r_\eta(\pi_\eta) - r_\eta(\pi) = 0$, then the minimizer policy only selects actions that are simultaneously optimal for all candidate hypotheses (see Lemma 5).

As discussed in Section 4.3, feedback in a trivial LLF problem can directly reveal the optimal action but nothing about the reward. In this case, the LLF problem is not discriminative, yet the stopping criteria ensures that the algorithm will not over-explore after identifying an optimal action.

HELiX is a concrete instantiation of how our conceptual LLF framework can inform algorithmic design, showing that LLF problems with finite transfer eluder dimensions can indeed be solved provably efficiently with a regret guarantee that depends sublinearly on the transfer eluder dimension.

**Theorem 1.** *Under Assumption 1 and Assumption 2, for all $T \in \mathbb{N}$, the regret of HELiX satisfies*

$$\mathrm{Regret}(T) \leq \widetilde{O}\left(T^{3/4}\left(\log N(\mathcal{H}, \epsilon_T^{\mathcal{H}}, d_{\mathcal{H}})\right)^{1/4}\sqrt{\dim_{TE}(\mathcal{H}, \ell, \epsilon_T^{\mathcal{H}})}\right),$$

*where $N(\mathcal{H}, \epsilon_T^{\mathcal{H}}, d_{\mathcal{H}})$ denotes the $\epsilon_t^{\mathcal{H}}$-covering number of $\mathcal{H}$ based on the pseudo-metric $d_{\mathcal{H}}$, $\dim_{TE}(\mathcal{H}, \ell, \epsilon_T^{\mathcal{H}})$ denotes the $\epsilon_T^{\mathcal{H}}$-transfer eluder dimension of $\mathcal{H}$, and $\epsilon_T^{\mathcal{H}} = \max\left\{\frac{1}{T^2}, \min_{a \in \mathcal{A}} \inf\{|r_\eta(a) - r^*(a)| : \eta \in \mathcal{H}, \eta \neq \eta^*\}\right\}$.*

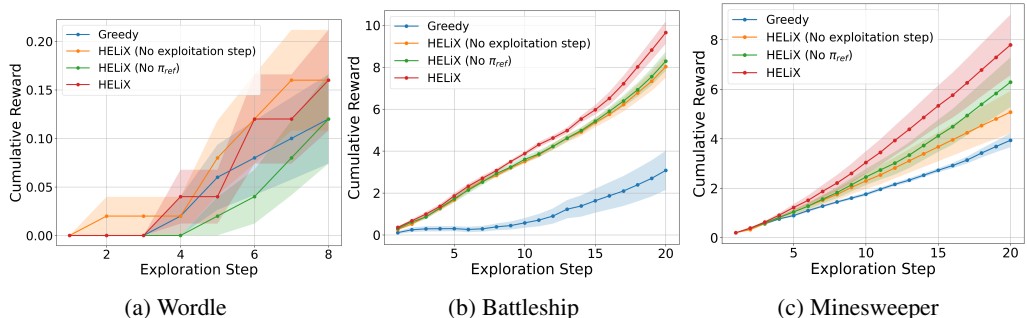

(a) Wordle      (b) Battleship      (c) Minesweeper

Figure 2: `HELiX` consistently outperforms the greedy baseline and `HELiX` variants. Shaded area represents the standard error of cumulative reward across different scenarios. We explain the ablations (no exploitation step; no $\pi_{\text{ref}}$) in Appendix F.

While the order $\widetilde{O}(T^{3/4})$ on the time horizon $T$ may appear suboptimal compared to classical $\widetilde{O}(\sqrt{T})$ optimal rates for bandit learning with direct reward feedback, this slower rate is in fact a principled consequence of our minimal assumptions. Specifically, our analysis makes no structural assumptions on the verifier loss $\ell$ beyond boundedness. If we have more structural knowledge of $\ell$, say, that it is a squared loss, then the bound can be tightened to match the optimal order $\widetilde{O}(\sqrt{T})$ in classical bandit learning (see Theorem 4 in Appendix B.4). We provide a sketch of the general argument in Theorem 1 in Appendix B.1, and include complete technical details in Appendix B.2.

Directly querying LLM for an action by prompting with the interaction history (with the lowest temperature) is similar to drawing actions from $\pi_\eta$ where $\eta$ is randomly sampled from $\arg\min_{\eta' \in \mathcal{H}} \sum_i \ell(A_i, O_i, \eta')$. In the RL setting, such a greedy algorithm does not explore and therefore does not always have low-regret. Since RL is a special case of discriminative LLF, we conjecture that this greedy algorithm also does not have regret guarantees for general LLF. We compare this baseline in all of our experiments and confirm that `HELiX` reliably outperforms it.

## 5   EMPIRICAL STUDIES

We validate a practical LLM-based approximation of our theoretical Algorithm 1 in experiments using three LLF problems (Wordle, Battleship and Minesweeper) constructed from the benchmark Tajwar et al. (2025). We provide the pseudocode for the practical implementation in Algorithm 2 (see Appendix E). Our algorithm selects actions based on multiple LLM thinking traces, treating them as samples from a space of hypotheses, while evolving this hypothesis space using past observations (see Figure 4). Please see Appendix F for details.

**Results** We consider the following LLF agents: `HELiX`, a ReAct agent (labeled as Greedy), and ablations of `HELiX`. We plot the cumulative reward as a function of the number of environment interaction steps on WORDLE, BATTLESHIP, and MINESWEEPER in Figure 6. We see that for all three environments, the ReAct agent (Greedy), where we only greedily choose the first action, performs worse generally. In environments where information-gathering is more necessary, such as in BATTLESHIP or in MINESWEEPER, agents designed to conduct strategic explorations tend to outperform the greedy base LLM by a large margin. As shown, `HELiX` consistently outperforms both the greedy baseline and `HELiX` variants. In particular, in BATTLESHIP and MINESWEEPER, `HELiX` performs significantly better than the baselines. We leave further analysis to Appendix F.

## 6   DISCUSSION

One might wonder if the transfer eluder dimension forms a lower bound for LLF. The answer, however, is negative, as some LLF problems are trivially solvable despite having infinite transfer eluder dimension. For example, our LLF framework does not preclude problem instances where rewards are arbitrary but feedback always reveals an optimal action. The transfer eluder dimension is unbounded in this case, yet the learning problem is easy and `HELiX` can also solve it in one step.

The difference between this case where the transfer eluder dimension is unbounded and the earlier demonstration case in Example 2 is that latter's reward class are constrained to be binary and the optimal action is unique, which keeps the transfer eluder dimension finite. We highlight that this

argument assumes worst-case verifier behavior, while LLMs in practice impose inductive biases on how feedback is interpreted. Empirically, we find that when explicitly presented with an optimal action, LLMs tend to trust and act on it, bypassing further learning to infer full rewards. HELiX captures this using the early stopping criterion (line 8), whereas naïve reward-driven UCB fails. This counterexample points to a gap in our current understanding: the true complexity of LLF may lie between worst-case reward identification and optimal behavior learning. Closing this gap by refining the transfer eluder dimension to lower-bound regret remains an important open question.

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

# A  LLF AND ITS RELATIONSHIP TO EXISTING PARADIGMS

To better understand the position of LLF among existing paradigms of learning from feedback, we provide an in-depth review in this section, as alluded to in Fig. 3. In all discussed paradigms, we focus our comparison on how different forms of feedback are subsumed within LLF, while other environment parameters are loosely assumed to be included in the LLF agent's hypothesis space. LLF covers the following learning paradigms commonly discussed in the literature:

| Learning Framework | Feedback Type | Discriminative? | LLF Verifier |
|---|---|---|---|
| Reinforcement Learning | Reward $r_{\eta^*} \in \mathbb{R}$ | Yes | $\ell(r_\eta(a), r_{\eta^*}(a))$ |
| Multi-objective RL | Reward vector $r_{\eta^*} \in \mathbb{R}^d$ | Yes | $\ell(r_\eta(a), r_{\eta^*}(a))$ |
| Interaction-Grounded Learning (IGL) | Rich feedback $y \in \mathcal{Y}$ s.t. $\exists \psi^* : \mathcal{Y} \times \mathcal{A} \to \mathbb{R} \approx r_{\eta^*}$ | Yes | Consistency loss: $\ell(y, a, \eta)$ (modeling $\psi^*$ is optional) |
| Preference-based RL | Comparison: $a_1 \overset{\eta^*}{\succ} a_2$ | No | $\mathbb{I}\left[ a_1 \overset{\eta}{\succ} a_2 \right]$ |
| Imitation Learning | Expert actions $a \in \mathcal{A}_{\eta^*}^*$ | No | $\mathbb{I}\left[ a \in \mathcal{A}_\eta^* \right]$ |

Table 1: Comparison of different learning frameworks and their feedback signals. All these learning paradigms are subsumed under the LLF framework with the last column specifying possible verifier losses for an LLF agent.

**Reinforcement learning (RL)**  In RL, upon seeing an environment state $x_t \in \mathcal{X}$, the agent chooses an action $a_t \in \mathcal{A}$ and observes a scalar reward feedback $r_t \in \mathbb{R}$. The rewards and states observed by the agent at any decision step $t$, can depend on the past observed states and actions. In LLF, the agent's hypothesis $\eta \in \mathcal{H}$ returns a reward function $r_\eta : \mathcal{A} \times \mathcal{X} \to [0, 1]$, while the feedback function is exactly the same: $f_\eta = r_\eta$. Hence, RL is trivially subsumed by LLF.

**Partial Monitoring Games**  In Partial Monitoring (Bartók et al., 2014), the agent observes an abstract feedback signal (not necessarily reward for its chosen action) and must deduce reward-optimal actions indirectly. The function that maps actions to feedbacks (signal function) is assumed known to the agent, and the challenge is to explore and infer optimal actions indirectly by leveraging the known signal function. In contrast, LLF assumes that the feedback function is unknown, and agents must interpret natural language feedback through a verifier to ascertain semantic consistency with hypotheses. The unknown feedback mapping in LLF fundamentally alters the learning challenge, requiring ways to extract insights from potentially ambiguous language feedback, and thus capturing a broader class of interactive learning scenarios.

**Interaction-grounded Learning (IGL) (Xie et al., 2021)**  In IGL, the environment generates a latent scalar reward $r(x, a) \in [0, 1]$ but only reveals a rich feedback vector $y \in \mathcal{Y}$. To enable learning, IGL framework assumes reward decodability, i.e., the existence of a decoder $\psi \in \Psi$, such that

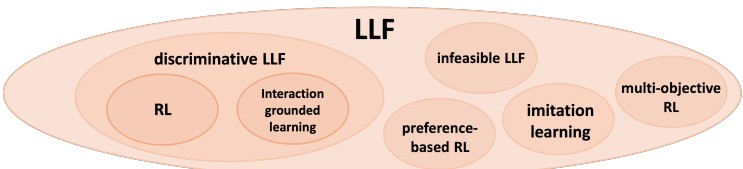

Figure 3: **LLF and its relationship to existing paradigms**. LLF covers many existing paradigms: (1) reinforcement learning (RL): agent learning from a scalar reward signal, (2) interaction-grounded learning (IGL) (Xie et al., 2021): agent observes a generic feedback vector that can decode a latent reward signal, (3) discriminative LLF: agent observes language feedback that discriminates between rewards, (4) multi-objective RL: extension of RL to problems with multiple objectives, combined via a utility function, (5) preference-based RL: feedback provides a comparison between two actions, (6) imitation learning: feedback provides expert demonstrations.

$\psi : \mathcal{Y} \times \mathcal{A} \to [0, 1]$, capable of extracting reward estimates for the agent. The remaining information in the feedback vector is regarded as distractions to learning and assumed to be distinguishable by the decoder. In contrast, LLF naturally accommodates information extraction by modeling both the latent reward $r_\eta$ and the feedback mapping $f_\eta$ (hence the feedback $y$), allowing the agent to reason about the consistency between the decoded rewards and the observed feedback vectors without needing to identify the true decoder $\psi^*$ or the true feedback function $f^*$. Furthermore, we only make discriminative assumptions about LLF so as to compare our algorithm to baselines in terms of reward. In fact, the generality of LLF allows it to handle cases where feedback contains much more *useful* information than reward. Specifically, when feedback is not discriminative but reveals information about the optimal action, LLF captures the decrease in problem complexity compared to learning from reward, while the latter setting is vacuous for IGL.

**Discriminative LLF**    Discriminative LLF, defined formally in Definition 5, subsumes the special case where the latent reward function is itself a function of the observed feedback (Xie et al., 2024). This framework generalizes both RL and IGL, and shares similarity with IGL with action-inclusive feedback (Xie et al., 2022b). In Xie et al. (2022b), the authors consider binary rewards, and assume that there exists a perfect reward decoding function $\psi^*$ such that $\mathbb{E}[\psi^*(o, a)|r = 1] - \mathbb{E}[\psi^*(o, a)|r = 0] = 1$ for decodability. Definition 5 generalizes this to the LLF framework where the LLF agent achieves this discriminative property via the LLF verifier loss for any two hypotheses $\eta, \eta'$ given action $o$ and feedback $a$. Thus discriminative LLF framework generalizes both RL and IGL, capturing scenarios where feedback is rich and structured (e.g., language) but ultimately reflects reward. As discussed in Section 4.3, this class of LLF problems can be no harder than the reward-only setting and may even improve sample efficiency by leveraging structure in the feedback to recover the reward signal more effectively.

The general LLF framework is a strict superset of both IGL and discriminative LLF as it accommodates scenarios where the reward is not decodable from the environment feedback.

**Multi-objective RL (MORL)**    MORL extends the standard RL framework to environments that return vector-valued rewards rather than a single scalar. The central challenge in MORL is balancing trade-offs across multiple objectives, often handled via scalarization methods (see single-policy learning approaches in (Roijers et al., 2013; Zhang & Golovin, 2020)) or Pareto front exploration (Mossalam et al., 2016). In LLF, this is naturally captured by allowing the agent's hypothesis to represent vector-valued reward functions. Furthermore, the verifier loss $\ell : \mathcal{A} \times \mathcal{O} \times \mathcal{H}$ can be extended accordingly. Since the reward vector may be under-determined with respect to the underlying utility function, we treat MORL as distinct from discriminative LLF (Definition 5), which assumes informativeness of feedback with respect to scalar reward.

**Preference-based RL**    In PbRL, the environment does not reveal scalar reward feedback. Instead, the agent receives pairwise preferences over actions (or trajectories), e.g., that action $a$ is preferred over action $a'$. These comparisons may be between actions selected by the agent or between one agent-chosen action and a reference provided by the environment. LLF captures this setting by modeling the feedback function $f_\eta$ as a binary comparator over pairs of actions such that $f_\eta(a, a') \in \{0, 1\}$ indicates the binary preference. The underlying reward model can be implicitly defined in the hypothesis $\eta$ such that it induces such preferences. Thus, this preference based structure fits within LLF.

**Imitation learning (IL)**    In IL, the agent learns from demonstrations of expert behavior rather than explicit feedback or rewards. To make a closer comparison with LLF, we can consider the interactive imitation learning setting, where the agent observes expert actions (corrections) for the all environment observations. IL can be modeled within the LLF framework by considering expert actions as a form of feedback $f_\eta^* = a^*$. Any hypothesis $\eta \in \mathcal{H}$ considered by the LLF agent can evaluate a verifier loss which corresponds to the discrepancy between the optimal action of the hypothesis $a_\eta^*$ and expert action $a^*$. IL is thus a special case of LLF where the feedback space is the action space itself, and consistency between the agent's output and expert-labeled actions is the verifier loss.

## B  REGRET ANALYSIS

### B.1  PROOF SKETCH

We sketch the regret analysis in four main steps. The full proof is presented in Appendix B.2.

**Step 1: Define confidence sets** For each hypothesis $\eta \in \mathcal{H}$, we define $\mathcal{L}_t(\eta) = \sum_{i=0}^{t-1} \left( \mathbb{E}_{O \sim f_{\eta^*}(A_i)}[\ell(A_i, O, \eta)] - \ell_{\eta^*}^{\min}(A_i) \right)$ to be the cumulative population prediction error and $L_t(\eta) = \sum_{i=0}^{t-1} \ell(A_i, O_i, \eta) = \sum_{i=0}^{t-1} \ell_i(\eta)$ to be the cumulative empirical verifier loss. We define confidence sets $\mathcal{H}_t = \{\eta \in \mathcal{H} : L_t(\eta) \le \min_{\eta' \in \mathcal{H}} L_t(\eta') + \beta_t\}$ where $\beta_t$ is a confidence parameter.

**Step 2: Regret decomposition** We let the width of a subset $\mathcal{V} \subseteq \mathcal{H}$ at an action $a \in \mathcal{A}$ be $w_{\mathcal{V}}(a) = \sup_{\overline{\eta} \in \mathcal{V}} |r_{\overline{\eta}}(a) - r^*(a)|$. Then, we can decompose the regret in terms of version space widths: $\text{Regret}(T, \eta^*) \le \sum_{t=0}^{T-1} \mathbb{E}\left[ w_{\mathcal{V}_t}(A_t) \cdot \mathbb{1}\{\eta^* \in \mathcal{V}_t\} + \mathbb{1}\{\eta^* \notin \mathcal{V}_t\} \right].$

**Step 3: Bounding the sum of widths via transfer eluder dimension** The key step is to show that if the width $w_{\mathcal{H}_t}(A_t) > \epsilon$ for some $\epsilon > 0$, then $A_t$ must be $\epsilon$-dependent on only $O(\beta_t/\epsilon^2)$ disjoint historical action sequences, where $\beta_t$ is the confidence parameter. By the definition of the transfer eluder dimension $d_{TE} = \dim_{TE}(\mathcal{H}, \ell, \epsilon)$, in any sequence of $N$ actions, there must be some action that is $\epsilon$-dependent on at least $\Omega(N/d)$ previous ones. Combining these facts forces the number of large-width version spaces $\sum_{t=0}^{T-1} \mathbb{1}\{w_{\mathcal{H}_t}(A_t) > \epsilon\}$ to be bounded by $O(\beta_T d/\epsilon^2)$. Rearranging terms and choosing a suitable sequence of $\epsilon$ gives that with high probability, $\sum_{t=0}^{T-1} w_{\mathcal{V}_t}(A_t) \le O(d_{TE} + 2\sqrt{3 d_{TE} \beta_T T})$. Note that when the stopping criteria is triggered, the per-step regret of all following steps become zero, and so the regret of HELiX is always bounded above by that without the stopping criteria.

**Step 4: Prove high-probability confidence set concentration** It remains to define suitable $\beta_t$'s and show that $\eta^* \in \mathcal{V}_t$ for all $t \in \mathbb{N}$ with high probability. Depending on what structural assumptions are known for the verifier loss $\ell$, we determine the rate of decay of $\beta_t$. If we only make the minimal assumption that $\ell$ is bounded, then $\beta_T = \widetilde{O}(\sqrt{T})$. Putting everything together proves Theorem 1.

### B.2  FULL ANALYSIS

We first define the version spaces used in the algorithm. As shorthand notations, define

$$\mathcal{L}_t(\eta) = \sum_{i=0}^{t-1} \left( \mathbb{E}_{O \sim f_{\eta^*}(A_i)}[\ell(A_i, O, \eta)] - \ell_{\eta^*}^{\min}(A_i) \right)$$

to be the cumulative population prediction error and

$$L_t(\eta) = \sum_{i=0}^{t-1} \ell(A_i, O_i, \eta) = \sum_{i=0}^{t-1} \ell_i(\eta)$$

to be the cumulative empirical verifier loss. A small value of $L_t(\eta)$ means $\eta$ is close to consistent with observed feedback. Let $\mathcal{V}_t \subseteq \mathcal{H}$ be the version space of all hypotheses still plausible after $t$ rounds of interactions. Concretely,

$$\mathcal{V}_t = \{\eta \in \mathcal{H} : L_t(\eta) \le \min_{\eta' \in \mathcal{H}} L_t(\eta') + \beta_t\}, \tag{1}$$

where $\beta_t > 0$ is an appropriately chosen confidence parameter so that we do not throw away the true hypothesis $\eta^*$ due to noise.

A useful approach to bounding the regret is to decompose it in terms of version spaces. We define the width of a subset $\mathcal{V} \subseteq \mathcal{H}$ at an action $a \in \mathcal{A}$ by

$$w_{\mathcal{V}}(a) = \sup_{\overline{\eta} \in \mathcal{V}} |r_{\overline{\eta}}(a) - r^*(a)|.$$

**Proposition 2** (Regret decomposition). *Fix any sequence $\{\mathcal{V}_t : t \in \mathbb{N}\}$, where $\mathcal{V}_t \subseteq \mathcal{H}$ is measurable with respect to $\sigma(H_t)$. Then for any $T \in \mathbb{N}$,*

$$\text{Regret}(T, \eta^*) \le \sum_{t=0}^{T-1} \mathbb{E}\left[ w_{\mathcal{V}_t}(A_t) \cdot \mathbb{1}\{\eta^* \in \mathcal{V}_t\} + \mathbb{1}\{\eta^* \notin \mathcal{V}_t\} \right].$$

*Proof.* We define the upper bound $U_t(a) = \sup\{r_\eta(a) : \eta \in \mathcal{V}_t\}$ and let $a^* \in \arg\max_{a \in \mathcal{A}} r^*(a)$. When $\eta^* \in \mathcal{V}_t$, the bound $r^*(a) \leq U_t(a)$ hold for all actions. This implies

$$r^*(\eta^*) - r^*(A_t) \leq (U_t(a^*) - r^*(A_t)) \cdot \mathbb{1}\{\eta^* \in \mathcal{V}_t\} + \mathbb{1}\{\eta^* \notin \mathcal{V}_t\}$$
$$\leq w_{\mathcal{V}_t}(A_t) \cdot \mathbb{1}\{\eta^* \in \mathcal{V}_t\} + \mathbb{1}\{\eta^* \notin \mathcal{V}_t\} + [U_t(a^*) - U_t(A_t)] \cdot \mathbb{1}\{\eta^* \in \mathcal{V}_t\}.$$

Since the algorithm selects an action $A_t$ that maximizes $U_t(a)$, the conclusion follows by taking the expectation and summing over all $t = 0, \ldots, T-1$. $\qquad\square$

This proposition reduces upper bounding the regret to bounding the expected sum of widths $\sum_{t=0}^{T-1} \mathbb{E}[w_{\mathcal{V}_t}(A_t)]$ if the version spaces $\mathcal{V}_t$ are constructed such that they contain $\eta^*$ with high probability.

We first introduce a class of Martingale exponential inequalities that will be useful throughout our analysis, including bounding the sum of widths and proving the high-confidence events $\eta^* \in \mathcal{V}_t$. For random variables $(X_t | t \in \mathbb{N})$ adapted to the filtration $(\mathcal{F}_t | t \in \mathbb{N})$, let us assume that $\mathbb{E}[\exp(\lambda X_t)]$ is finite for all $\lambda$ and $\mathbb{E}[X_t | \mathcal{F}_{t-1}] = 0$. We assume that there is a uniform upper bound on the cumulant generating function (i.e., log moment generating function) for the conditional distribution of $X_t$.

**Lemma 1** (Cumulant generating function). *If there is a sequence of convex functions $\{\psi_t : [0, \infty) \to \mathbb{R}\}_{t=0}^{\infty}$ with $\psi_t(0) = 0$ such that, for all $t \in \mathbb{N}$ and all $\lambda \in [0, \infty)$,*

$$\log \mathbb{E}\left[e^{\lambda|X_t|}|\mathcal{F}_{t-1}\right] \leq \psi_t(\lambda),$$

*then for all $\delta \in (0, 1)$ and $T \in \mathbb{N}$, with probability $1 - \delta$,*

$$\left|\sum_{t=0}^{T-1} X_t\right| \leq \inf_{\lambda \in [0,\infty)} \left\{\frac{\sum_{t=0}^{T-1} \psi_t(\lambda) + \log(2/\delta)}{\lambda}\right\}.$$

*Proof.* Let $S_T = \sum_{t=0}^{T-1} X_t$. By Markov's inequality, for all $u \in \mathbb{R}$ and $\lambda \in [0, \infty)$,

$$\mathbb{P}(S_T \geq u) = \mathbb{P}\left(e^{\lambda S_T} \geq e^{\lambda u}\right) \leq \frac{\mathbb{E}[e^{\lambda S_T}]}{e^{\lambda u}} = \frac{\mathbb{E}[\mathbb{E}[e^{\lambda S_T}|\mathcal{F}_{T-1}]]}{e^{\lambda u}} = \frac{\mathbb{E}[e^{\lambda \sum_{t=0}^{T-2} X_t}\mathbb{E}[e^{\lambda X_{T-1}}|\mathcal{F}_{T-1}]]}{e^{\lambda u}}$$
$$\leq \frac{\mathbb{E}[e^{\lambda \sum_{t=0}^{T-2} X_t}]\exp(\psi_{T-1}(\lambda))}{e^{\lambda u}} \leq \cdots \leq \frac{\exp(\sum_{t=0}^{T-1} \psi_t(\lambda))}{e^{\lambda u}}.$$

This gives

$$\mathbb{P}(S_T \geq u) \leq \exp\left(-\lambda u + \sum_{t=0}^{T-1} \psi_t(\lambda)\right)$$

for all $\lambda \in [0, \infty)$. Applying the same argument to $-X_t$, we have

$$\mathbb{P}(S_T \leq -u) = \mathbb{P}(-S_T \geq u) \leq \exp\left(-\lambda u + \sum_{t=0}^{T-1} \psi_t(\lambda)\right).$$

Solving for $u$ to achieve a $\delta/2$ probability for each side, and taking the infimum over $\lambda \in [0, \infty)$, we have with probability at least $1 - \delta$,

$$S_T \leq \inf_{\lambda \in [0,\infty)} \left\{\frac{\sum_{t=0}^{T-1} \psi_t(\lambda) + \log(2/\delta)}{\lambda}\right\}.$$

$\qquad\square$

We now proceed to bounding the sum of widths $\sum_{t=0}^{T-1} \mathbb{E}[w_{\mathcal{V}_t}(A_t)]$ when the event $\eta^* \in \mathcal{V}_t$ holds. As a first step, we show that there cannot be many version spaces $\mathcal{V}_t$ with a large width. For all $t \in \mathbb{N}$ and $\eta, \eta' \in \mathcal{H}$, we define the martingale difference

$$Z_t(\eta, \eta') = \mathbb{E}_{O \sim f_{\eta^*}(A_t)}[\ell(A_t, O, \eta) - \ell(A_t, O, \eta')|\mathcal{G}_{t-1}] - (\ell(A_t, O_t, \eta) - \ell(A_t, O_t, \eta')).$$

Notice that $Z_t$ have expectation zero and constitutes a martingale difference sequence adapted to the filtration $(\mathcal{G}_t | t \in \mathbb{N})$ where $\mathcal{G}_t$ is the $\sigma$-algebra generated by all observations $\{(a_0, o_1), \ldots, (a_t, o_t)\}$ up to time $t$.

**Proposition 3.** *If the conditions in Lemma 1 holds for $(Z_t | t \in \mathbb{N})$ adapted to $(\mathcal{G}_t | t \in \mathbb{N})$ with cumulative generating function bound $(\psi_t | t \in \mathbb{N})$, $(\beta_t \geq 0 | t \in \mathbb{N})$ in (1) is a nondecreasing sequence such that for all $t \in \mathbb{N}$, $\beta_t \geq \inf_{\lambda \in [0,\infty)} \left\{ \frac{\sum_{i=0}^{t-1} \psi_i(\lambda) + \log(10t^2/3\delta)}{\lambda} \right\}$, then for all $\delta \in (0,1)$, with probability at least $1 - \delta$,*

$$\sum_{t=0}^{T-1} \mathbb{1}\{w_{\mathcal{V}_t}(A_t) > \epsilon\} \cdot \mathbb{1}\{\eta^* \in \mathcal{V}_t\} \leq \left( \frac{3\beta_T}{\epsilon^2} + 1 \right) \dim_{TE}(\mathcal{H}, \ell, \epsilon)$$

*for all $T \in \mathbb{N}$ and $\epsilon > 0$.*

*Proof.* We first show that if $w_{\mathcal{V}_t}(A_t) > \epsilon$ and $\eta^* \in \mathcal{V}_t$, then with high probability, $A_t$ is $\epsilon$-dependent on fewer than $O(\beta_t/\epsilon^2)$ disjoint subsequences of $(A_0, A_1, \ldots, A_{t-1})$. If $w_{\mathcal{V}_t}(A_t) > \epsilon$ and $\eta^* \in \mathcal{V}_t$, there exists $\overline{\eta} \in \mathcal{V}_t$ such that $|r_{\overline{\eta}}(A_t) - r_{\eta^*}(A_t)| > \epsilon$. By definition, if $A_t$ is $\epsilon$-dependent on a subsequence $(A_{i_1}, \ldots, A_{i_k})$ of $(A_0, \ldots, A_{t-1})$, then we have that

$$\sum_{j=1}^{k} \left( \mathbb{E}_{O \sim f_{\eta^*}(A_{i_j})}[\ell(A_{i_j}, O, \overline{\eta})] - \ell_{\eta^*}^{\min}(A_{i_j}) \right) > \epsilon^2.$$

It follows that if $A_t$ is $\epsilon$-dependent on $K$ disjoint subsequences of $(A_0, \ldots, A_{t-1})$ then

$$\sum_{i=0}^{t-1} \left( \mathbb{E}_{O \sim f_{\eta^*}(A_i)}[\ell(A_i, O, \overline{\eta})] - \ell_{\eta^*}^{\min}(A_i) \right) > K\epsilon^2.$$

Then

$$\sum_{i=0}^{t-1} \left( \mathbb{E}_{O \sim f_{\eta^*}(A_i)}[\ell(A_i, O, \overline{\eta})] - \ell_{\eta^*}^{\min}(A_i) \right)$$

$$= \sum_{i=0}^{t-1} \mathbb{E}_{O \sim f_{\eta^*}(A_i)} [\ell(A_i, O, \overline{\eta}) - \ell(A_i, O, \eta^*)]$$

$$= \left[ \sum_{i=0}^{t-1} \ell(A_i, O_i, \eta^*) - \min_{\eta' \in \mathcal{H}} \sum_{i=0}^{t-1} \ell(A_i, O_i, \eta') \right] - \left[ \sum_{i=0}^{t-1} \ell(A_i, O_i, \overline{\eta}) - \min_{\eta' \in \mathcal{H}} \sum_{i=0}^{t-1} \ell(A_i, O_i, \eta') \right]$$

$$+ \left[ \sum_{i=0}^{t-1} [\ell(A_i, O_i, \overline{\eta}) - \ell(A_i, O_i, \eta^*)] - \sum_{i=0}^{t-1} \mathbb{E}_{O \sim f_{\eta^*}(A_i)} [\ell(A_i, O, \overline{\eta}) - \ell(A_i, O, \eta^*)] \right]$$

$$\leq \left| \sum_{i=0}^{t-1} \ell(A_i, O_i, \eta^*) - \min_{\eta' \in \mathcal{H}} \sum_{i=0}^{t-1} \ell(A_i, O_i, \eta') \right| + \left| \sum_{i=0}^{t-1} \ell(A_i, O_i, \overline{\eta}) - \min_{\eta' \in \mathcal{H}} \sum_{i=0}^{t-1} \ell(A_i, O_i, \eta') \right|$$

$$+ \left[ \sum_{i=0}^{t-1} [\ell(A_i, O_i, \overline{\eta}) - \ell(A_i, O_i, \eta^*)] - \sum_{i=0}^{t-1} \mathbb{E}_{O \sim f_{\eta^*}(A_i)} [\ell(A_i, O, \overline{\eta}) - \ell(A_i, O, \eta^*)] \right]$$

$$\leq 2\beta_t + \sum_{i=0}^{t-1} [\ell(A_i, O_i, \overline{\eta}) - \ell(A_i, O_i, \eta^*)] - \sum_{i=0}^{t-1} \mathbb{E}_{O \sim f_{\eta^*}(A_i)} [\ell(A_i, O, \overline{\eta}) - \ell(A_i, O, \eta^*)]$$

$$= 2\beta_t - \sum_{i=0}^{t-1} Z_i(\overline{\eta}, \eta^*).$$

Using Lemma 1,

$$\mathbb{P}\left( \left| \sum_{i=0}^{t-1} Z_i(\overline{\eta}, \eta^*) \right| > \inf_{\lambda \in [0,\infty)} \left\{ \frac{\sum_{i=0}^{t-1} \psi_i(\lambda) + \log(2/\delta)}{\lambda} \right\} \right) \leq \delta.$$

We choose a sequence $\{\delta_t\}_{t \in \mathbb{N}_{>0}}$ where $\delta_t = \frac{3\delta}{5t^2}$, and so $\sum_{t=1}^{\infty} \delta_t < \delta$. Using a union bound over all $t \in \mathbb{N}_{>0}$, we have that with probability at least $1 - \delta$, for all $t \in \mathbb{N}$,

$$\left| \sum_{i=0}^{t-1} Z_i(\overline{\eta}, \eta^*) \right| \leq \inf_{\lambda \in [0,\infty)} \left\{ \frac{\sum_{i=0}^{t-1} \psi_i(\lambda) + \log(10t^2/3\delta)}{\lambda} \right\} \leq \beta_t.$$

Since $\{\beta_t\}_{t\in\mathbb{N}}$ is nondecreasing in $t$, we have that with probability at least $1 - \delta$, $K\epsilon^2 \leq 3\beta_T$. It then follows that with probability at least $1 - \delta$, $K \leq 3\beta_T/\epsilon^2$.

Next, we take any action sequence $(a_1, \ldots, a_\tau)$ and show that there is some element $a_j$ that is $\epsilon$-dependent on at least $\tau/d - 1$ disjoint subsequences of $(a_1, \ldots, a_{j-1})$, where $d = \dim_{TE}(\mathcal{H}, \ell, \epsilon)$. For an integer $K$ satisfying $Kd + 1 \leq \tau \leq Kd + d$, we will construct $K$ disjoint subsequences $B_1, \ldots, B_K$ inductively starting with $B_i = (a_i)$ for $i = 1, \ldots, K$. If $a_{K+1}$ is $\epsilon$-dependent on each subsequence $B_1, \ldots, B_K$, we are done. Otherwise, there must be at least one subsequence for which $a_{K+1}$ is $\epsilon$-independent. We choose such a subsequence and append $a_{K+1}$ to it. We will repeat this process for $a_j$ with $j = K + 2, K + 3, \ldots$ until either $a_j$ is $\epsilon$-dependent on each subsequence or $j = \tau$. If the first case occurs, we are done. If $j = \tau$, we necessarily have that $\sum |B_i| \geq Kd$. Since each element of a subsequence $B_i$ is $\epsilon$-independent of its predecessors, $|B_i| = d$. By the definition of $\dim_{TE}(\mathcal{H}, \ell, \epsilon)$, $a_\tau$ must be $\epsilon$-dependent on each subsequence.

We now take $(A_1, \ldots, A_\tau)$ to be the subsequence $(A_{t_1}, \ldots, A_{t_\tau})$ of $(A_1, \ldots, A_T)$ where for each $A_t$, we have $w_{\mathcal{V}_t}(A_t) > \epsilon$. As we have shown first, with probability at least $1 - \delta$, each $A_{t_j}$ is $\epsilon$-dependent on fewer than $3\beta_T/\epsilon^2$ disjoint subsequences of $(A_1, \ldots, A_{j-1})$. As we have shown in the preceding paragraph, there is some $a_j$ that is $\epsilon$-dependent on at least $\tau/d - 1$ disjoint subsequences of $(a_1, \ldots, a_{j-1})$. Combining these two facts, we may conclude that $\tau/d - 1 \leq 3\beta_T/\epsilon^2$. It follows that with probability at least $1 - \delta$, $\tau \leq \left(3\beta_T/\epsilon^2 + 1\right)d$ as desired. $\square$

We are now ready to bound the sum of widths $\sum_{t=0}^{T-1} \mathbb{E}[w_{\mathcal{V}_t}(A_t)]$ when the event $\eta^* \in \mathcal{V}_t$ holds. Consider the $\epsilon_T^{\mathcal{H}}$-transfer eluder dimension of $\mathcal{H}$, where

$$\epsilon_t^{\mathcal{H}} = \max\left\{\frac{1}{t^2}, \min_{a\in\mathcal{A}}\inf\{|r_\eta(a) - r^*(a)| : \eta \in \mathcal{H}, \eta \neq \eta^*\}\right\}. \tag{2}$$

**Lemma 2.** *If the conditions in Lemma 1 holds for $(Z_t | t \in \mathbb{N})$ adapted to $(\mathcal{G}_t | t \in \mathbb{N})$ with cumulative generating function bound $(\psi_t | t \in \mathbb{N})$, $(\beta_t \geq 0 | t \in \mathbb{N})$ in (1) is a nondecreasing sequence such that for all $t \in \mathbb{N}$, $\beta_t \geq \inf_{\lambda\in[0,\infty)}\left\{\frac{\sum_{i=0}^{t-1}\psi_i(\lambda) + \log(10t^2/3\delta)}{\lambda}\right\}$, then for all $\delta \in (0, 1)$, with probability at least $1 - \delta$,*

$$\sum_{t=0}^{T-1} w_{\mathcal{V}_t}(A_t) \cdot \mathbb{1}\{\eta^* \in \mathcal{V}_t\} \leq \frac{1}{T} + \min\left\{\dim_{TE}(\mathcal{H}, \ell, \epsilon_T^{\mathcal{H}}), T\right\} + 2\sqrt{3\dim_{TE}(\mathcal{H}, \ell, \epsilon_T^{\mathcal{H}})\beta_T T}$$

*for all $T \in \mathbb{N}$.*

*Proof.* Let $d_T = \dim_{TE}(\mathcal{H}, \ell, \epsilon_T^{\mathcal{H}})$ and $w_t = w_{\mathcal{V}_t}(A_t)$. Reorder the sequence $(w_1, \ldots, w_T) \to (w_{i_1}, \ldots, w_{i_T})$ where $w_{i_1} \geq w_{i_2} \geq \cdots \geq w_{i_T}$. We have

$$\sum_{t=0}^{T-1} w_{\mathcal{V}_t}(A_t) \cdot \mathbb{1}\{\eta^* \in \mathcal{V}_t\}$$

$$= \sum_{t=0}^{T-1} w_{i_t} \cdot \mathbb{1}\{\eta^* \in \mathcal{V}_{i_t}\}$$

$$= \sum_{t=0}^{T-1} w_{i_t} \cdot \mathbb{1}\{\eta^* \in \mathcal{V}_{i_t}\} \cdot \mathbb{1}\{w_{i_t} > \epsilon_T^{\mathcal{H}}\} + \sum_{t=0}^{T-1} w_{i_t} \cdot \mathbb{1}\{\eta^* \in \mathcal{V}_{i_t}\} \cdot \mathbb{1}\{w_{i_t} \leq \epsilon_T^{\mathcal{H}}\}$$

$$\leq \frac{1}{T} + \sum_{t=0}^{T-1} w_{i_t} \cdot \mathbb{1}\{\eta^* \in \mathcal{V}_{i_t}\} \cdot \mathbb{1}\{w_{i_t} > \epsilon_T^{\mathcal{H}}\}.$$

The last inequality follows since either $\epsilon_T^{\mathcal{H}} = 1/T^2$ and $\sum_{t=0}^{T-1}\epsilon_T^{\mathcal{H}} = 1/T$ or $\epsilon_T^{\mathcal{H}}$ is set below the smallest possible width and hence $\mathbb{1}\{w_{i_t} \leq \epsilon_T^{\mathcal{H}}\}$ never occurs. We have that $w_{i_t} \leq 1$. Also, $w_{i_t} > \epsilon \iff \sum_{k=0}^{T-1}\mathbb{1}\{w_{\mathcal{V}_k}(a_k) > \epsilon\} \geq t$. By Proposition 3, with probability at least $1 - \delta$, this can only happen if $t < \left(3\beta_T/\epsilon^2 + 1\right)\dim_{TE}(\mathcal{H}, \ell, \epsilon)$. For $\epsilon \geq \epsilon_T^{\mathcal{H}}$, since $\dim_{TE}(\mathcal{H}, \ell, \epsilon')$ is non-increasing in $\epsilon'$, $\dim_{TE}(\mathcal{H}, \ell, \epsilon) \leq \dim_{TE}(\mathcal{H}, \ell, \epsilon_T^{\mathcal{H}}) = d_T$. Therefore, when $w_{i_t} > \epsilon \geq \epsilon_T^{\mathcal{H}}$,

$t \leq \left(3\beta_T/\epsilon^2 + 1\right) d_T$, implying $\epsilon \leq \sqrt{\frac{3\beta_T d_T}{t - d_T}}$. So if $w_{i_t} > \epsilon_T^{\mathcal{H}}$, then $w_{i_t} \leq \min\{1, \sqrt{\frac{3\beta_T d_T}{t - d_T}}\}$. Thus,

$$\sum_{t=0}^{T-1} w_{i_t} \cdot \mathbb{1}\{\eta^* \in \mathcal{V}_{i_t}\} \cdot \mathbb{1}\{w_{i_t} > \epsilon_T^{\mathcal{H}}\} \leq d_T + \sum_{t=d_T+1}^{T-1} \sqrt{\frac{3\beta_T d_T}{t - d_T}}$$

$$\leq d_T + \sqrt{3\beta_T d_T} \int_{t=1}^{T-1} \frac{1}{\sqrt{t}} dt$$

$$= d_T + 2\sqrt{3\beta_T d_T T}.$$

Since the sum of widths is always bounded by $T$, this implies that with probability $1 - \delta$,

$$\sum_{t=0}^{T-1} w_{\mathcal{V}_t}(a_t) \cdot \mathbb{1}\{\eta^* \in \mathcal{V}_t\}$$

$$\leq \min \left\{ T, \frac{1}{T} + \dim_{TE}(\mathcal{H}, \ell, \epsilon_T^{\mathcal{H}}) + 2\sqrt{3 \dim_{TE}(\mathcal{H}, \ell, \epsilon_T^{\mathcal{H}})\beta_T T} \right\}$$

$$\leq \frac{1}{T} + \min \left\{ \dim_{TE}(\mathcal{H}, \ell, \epsilon_T^{\mathcal{H}}), T \right\} + 2\sqrt{3 \dim_{TE}(\mathcal{H}, \ell, \epsilon_T^{\mathcal{H}})\beta_T T}.$$

$\square$

So far, we have only considered HELiX without the exploitation step. We remark that by Lemma 6, when the exploitation step is triggered, the per-step regret of all following steps become zero, and so the regret of the full HELiX is always bounded above by that without the exploitation step. Combining this observation with Lemma 2 and Proposition 2, we arrive at the following abstract regret bound in terms of the version space confidence parameter $\beta_T$.

**Theorem 2.** *If it holds that for some $\delta \in (0, 1)$, with probability at least $1 - \delta$, $\eta^* \in \mathcal{V}_t$ for all $t$, then for all $T \in \mathbb{N}$,*

$$\mathrm{Regret}(T) \leq 1 + \frac{1}{T} + \min\{\dim_{TE}(\mathcal{H}, \ell, \epsilon_T^{\mathcal{H}}), T\} + 2\sqrt{3 \dim_{TE}(\mathcal{H}, \ell, \epsilon_T^{\mathcal{H}})\beta_T T}.$$

The dominant term in the regret bound is

$$2\sqrt{3 \dim_{TE}(\mathcal{H}, \ell, \epsilon_T^{\mathcal{H}})\beta_T T}.$$

For our main theorem, it remains to design suitable version spaces $\mathcal{V}_t$ and show that they contain the true hypothesis $\eta^*$ with high probability. Crucially, the rate at which the confidence parameters $\beta_t$ of these version spaces shrink depends on concentration properties of the verifier loss function $\ell$. Note that for the general LLF framework, we have assumed only that $\ell$ is a bounded function taking values in $[0, 1]$. If we have more structural assumptions on the verifier loss $\ell$, for example, that $\ell$ is $\alpha$-strongly convex, then we may arrive at a tighter regret bound up to order $\sqrt{T}$ by taking $\beta_T$ to be of constant order.

### B.3 VERSION SPACE CONSTRUCTION FOR GENERAL BOUNDED LOSS

Consider the most general case with minimal assumptions on the loss function, namely, that it is bounded between $[0, 1]$ for all inputs. Then we prove the following high-probability event:

**Lemma 3** (High-probability event). *For all $\delta > 0$, $\eta, \eta' \in \mathcal{H}$,*

$$\mathbb{P}\left( \mathcal{L}_T(\eta') \geq \mathcal{L}_T(\eta) + L_T(\eta') - L_T(\eta) - \sqrt{2T \log\left(\frac{10T^2}{3\delta}\right)}, \quad \forall T \in \mathbb{N} \right) \geq 1 - \delta.$$

*Proof.* For each $t = 1, \ldots, T$, define the Martingale difference sequence

$$X_t = \mathbb{E}_{O \sim f_{\eta^*}(A_t)}\left[\ell(A_t, O, \eta) - \ell(A_t, O, \eta')\right] - \left(\ell(A_t, O_t, \eta) - \ell(A_t, O_t, \eta')\right).$$

$$\mathcal{L}_T(\eta') - \mathcal{L}_T(\eta) - (L_T(\eta') - L_T(\eta))$$

$$= \sum_{t=0}^{T-1} \left( \mathbb{E}_{O \sim f_{\eta^*}(A_t)}[\ell(A_t, O, \eta)] - \mathbb{E}_{O \sim f_{\eta^*}(A_t)}[\ell(A_t, O, \eta')] \right) - \sum_{t=0}^{T-1} \left( \ell(A_t, O_t, \eta) - \ell(A_t, O_t, \eta') \right)$$

$$= \sum_{t=0}^{T-1} \mathbb{E}_{O \sim f_{\eta^*}(A_t)}\left[ \ell(A_t, O, \eta) - \ell(A_t, O, \eta') \right] - \sum_{t=0}^{T-1} \left( \ell(A_t, O_t, \eta) - \ell(A_t, O_t, \eta') \right)$$

$$= \sum_{t=0}^{T-1} X_t.$$

Notice that $X_t$ have expectation zero and constitutes a Martingale difference sequence adapted to the filtration $\{\mathcal{G}_t\}_{t \geq 1}$ where $\mathcal{G}_t$ is the $\sigma$-algebra generated by all observations $\{(A_0, O_1), \ldots, (A_t, O_t)\}$ up to time $t$. Since feedback losses $\ell(a, o, \eta)$ are uniformly bounded between $[0, 1]$, we have that $X_t \in [-2, 2]$ with probability 1. Using Lemma 1 with $\psi_t(\lambda) = \lambda^2 / 2$ and taking the infimum over $\lambda$, we get

$$\mathbb{P}\left( \left| \sum_{t=0}^{T-1} X_t \right| > \sqrt{2T \log(2/\delta)} \right) \leq \delta.$$

We choose a sequence $\{\delta_T\}_{T \in \mathbb{N}_{>0}}$ where $\delta_T = \frac{3\delta}{5T^2}$ such that $\sum_{T=1}^{\infty} \delta_T < \delta$. Using a union bound over all $T \in \mathbb{N}_{\geq 0}$, we have that with probability at least $1 - \delta$,

$$|\mathcal{L}_T(\eta') - \mathcal{L}_T(\eta) - (L_T(\eta') - L_T(\eta))| \leq \sqrt{2T \log\left(\frac{2}{\delta_T}\right)} = \sqrt{2T \log\left(\frac{10T^2}{3\delta}\right)} \quad \forall T \in \mathbb{N}.$$

$$\square$$

Since $\eta^*$ is the true hypothesis, by Assumption 3, it minimizes the population loss $\mathcal{L}_T(\eta)$ for all $T \in \mathbb{N}$. That is, for all $\eta \in \mathcal{H}$,

$$\mathcal{L}_T(\eta^*) \leq \mathcal{L}_T(\eta) \quad \forall T \in \mathbb{N}.$$

Suppose $m = |\mathcal{H}| < \infty$. By Lemma 3, for any $\eta \in \mathcal{H}$, with probability at least $1 - \delta/m$, for all $T \in \mathbb{N}$,

$$L_T(\eta^*) - L_T(\eta) \leq \mathcal{L}_T(\eta^*) - \mathcal{L}_T(\eta) + \sqrt{2T \log\left(\frac{10T^2}{3\delta}\right)} \leq \sqrt{2T \log\left(\frac{10mT^2}{3\delta}\right)}.$$

Using a union bound over $\mathcal{H}$, with probability at least $1 - \delta$, the true hypothesis $\eta^*$ is contained in the version space

$$\mathcal{V}_T = \left\{ \eta \in \mathcal{H} : L_T(\eta) \leq \min_{\eta' \in \mathcal{H}} L_T(\eta') + \sqrt{2T \log\left(\frac{10|\mathcal{H}|T^2}{3\delta}\right)} \right\}$$

for all $T \in \mathbb{N}$. To extend this to a space of infinite hypotheses, we measure the set $\mathcal{H}$ by some discretization scale $\alpha$. Recall that we define distances in the hypothesis space in terms of the loss function $\ell$:

$$d_{\mathcal{H}}(\eta, \eta') = \sup_{a \in \mathcal{A}, o \in \mathcal{O}} |\ell(a, o, \eta) - \ell(a, o, \eta')|.$$

**Lemma 4.** $d_{\mathcal{H}}(\cdot, \cdot)$ *is a pseudometric on* $\mathcal{H}$.

*Proof.* We check the axioms for a pseudometric.

- nonnegativity: $d_{\mathcal{H}}(\eta, \eta) = 0$ and $d_{\mathcal{H}}(\eta, \eta') \geq 0$ for all $\eta, \eta' \in \mathcal{H}$.

- symmetry: $d_{\mathcal{H}}(\eta, \eta') = d_{\mathcal{H}}(\eta', \eta)$.

- triangle inequality: for each $a \in \mathcal{A}$ and $o \in \mathcal{O}$, $|\ell(a, o, \eta) - \ell(a, o, \eta'')| \leq |\ell(a, o, \eta) - \ell(a, o, \eta')| + |\ell(a, o, \eta') - \ell(a, o, \eta'')|$. Taking the supremum over $\mathcal{A}$ and $\mathcal{O}$ yields the desired property.

$$\square$$

Let $N(\mathcal{H}, \alpha, d_{\mathcal{H}})$ denote the $\alpha$-covering number of $\mathcal{H}$ in the pseudometric $d_{\mathcal{H}}$, and let

$$\beta_t^*(\mathcal{H}, \delta, \alpha) := \sqrt{2t \log\left(\frac{10N(\mathcal{H}, \alpha, d_{\mathcal{H}})t^2}{3\delta}\right)} + 2\alpha t. \tag{3}$$

**Proposition 4.** *For $\delta > 0$, $\alpha > 0$, and $T \in \mathbb{N}$, define*

$$\mathcal{V}_T := \left\{ \eta \in \mathcal{H} : L_T(\eta) \leq \min_{\eta' \in \mathcal{H}} L_T(\eta') + \beta_T^* \right\}$$

*Then it holds that*

$$\mathbb{P}\left(\eta^* \in \bigcap_{T=1}^{\infty} \mathcal{V}_T\right) \geq 1 - \delta.$$

*Proof.* Let $\mathcal{H}^\alpha \subseteq \mathcal{H}$ be an $\alpha$-cover of $\mathcal{H}$ in the pseudometric $d_{\mathcal{H}}$. In other words, for any $\eta \in \mathcal{H}$, there is an $\eta^\alpha \in \mathcal{H}^\alpha$ such that $d_{\mathcal{H}}(\eta, \eta^\alpha) \leq \alpha$. A union bound over $\mathcal{H}^\alpha$ gives that with probability at least $1 - \delta$,

$$(\mathcal{L}_T(\eta^\alpha) - L_T(\eta^\alpha)) - (\mathcal{L}_T(\eta^*) - L_T(\eta^*)) \leq \sqrt{2T \log\left(\frac{10|\mathcal{H}^\alpha|T^2}{3\delta}\right)}$$

$$\implies (\mathcal{L}_T(\eta) - L_T(\eta)) - (\mathcal{L}_T(\eta^*) - L_T(\eta^*)) \leq \sqrt{2T \log\left(\frac{10|\mathcal{H}^\alpha|T^2}{3\delta}\right)}$$

$$+ \underbrace{(\mathcal{L}_T(\eta) - L_T(\eta)) - (\mathcal{L}_T(\eta^\alpha) - L_T(\eta^\alpha))}_{\text{discretization error}}.$$

The discretization error can be expanded and bounded as

$$\sum_{t=0}^{T-1} \left[ \mathbb{E}_{O \sim f_{\eta^*}(A_t)}\left[\ell(A_t, O, \eta) - \ell(A_t, O, \eta^\alpha)\right] - \ell(A_t, O_t, \eta) + \ell(A_t, O_t, \eta^\alpha) \right] \leq 2\alpha T.$$

Since $\eta^*$ is a minimizer of $\mathcal{L}_T(\cdot)$, we have that with probability at least $1 - \delta$,

$$L_T(\eta^*) - L_T(\eta) \leq \sqrt{2T \log\left(\frac{10|\mathcal{H}^\alpha|T^2}{3\delta}\right)} + 2\alpha T.$$

We take the infimum over the size of $\alpha$-covers, which results in the bound

$$L_T(\eta^*) - L_T(\eta) \leq \sqrt{2T \log\left(\frac{10N(\mathcal{H}, \alpha, d_{\mathcal{H}})T^2}{3\delta}\right)} + 2\alpha T.$$

$$\square$$

Taking $\delta = \frac{1}{T}$ and plugging $\beta_T = \beta_T^*(\mathcal{H}, \delta, \epsilon_T^{\mathcal{H}})$ into the abstract regret bound in Theorem 2 proves the following main theorem.

**Theorem 3.** *For all $T \in \mathbb{N}$,*

$$\text{Regret}(T) \leq 1 + \frac{1}{T} + \min\{\dim_{TE}(\mathcal{H}, \ell, \epsilon_T^{\mathcal{H}}), T\}$$

$$+ 2\sqrt{3\sqrt{2} \log\left(\frac{10N(\mathcal{H}, \alpha, d_{\mathcal{H}})T^2}{3\delta}\right)^{1/2} \dim_{TE}(\mathcal{H}, \ell, \epsilon_T^{\mathcal{H}})T^{3/2} + 6\dim_{TE}(\mathcal{H}, \ell, \epsilon_T^{\mathcal{H}})}.$$

*Proof.*

$$\text{Regret}(T)$$

$$\leq 1 + \frac{1}{T} + \min\{\dim_{TE}(\mathcal{H}, \ell, \epsilon_T^{\mathcal{H}}), T\} + 2\sqrt{3\dim_{TE}(\mathcal{H}, \ell, \epsilon_T^{\mathcal{H}})\beta_T^*(\mathcal{H}, \delta, \epsilon_T^{\mathcal{H}})T}$$

$$= 1 + \frac{1}{T} + \min\{\dim_{TE}(\mathcal{H}, \ell, \epsilon_T^{\mathcal{H}}), T\}+$$

$$+ 2\sqrt{3\dim_{TE}(\mathcal{H}, \ell, \epsilon_T^{\mathcal{H}})\left(\sqrt{2T\log\left(\frac{10N(\mathcal{H}, \epsilon_T^{\mathcal{H}}, d_{\mathcal{H}})T^2}{3\delta}\right)} + 2\epsilon_T^{\mathcal{H}}T\right)T}$$

$$= 1 + \frac{1}{T} + \min\{\dim_{TE}(\mathcal{H}, \ell, \epsilon_T^{\mathcal{H}}), T\}+$$

$$+ 2\sqrt{3\sqrt{2}\log\left(\frac{10N(\mathcal{H}, \alpha, d_{\mathcal{H}})T^2}{3\delta}\right)^{1/2}\dim_{TE}(\mathcal{H}, \ell, \epsilon_T^{\mathcal{H}})T^{3/2} + 6\epsilon_T^{\mathcal{H}}\dim_{TE}(\mathcal{H}, \ell, \epsilon_T^{\mathcal{H}})T^2}$$

$$\leq 1 + \frac{1}{T} + \min\{\dim_{TE}(\mathcal{H}, \ell, \epsilon_T^{\mathcal{H}}), T\}+$$

$$+ 2\sqrt{3\sqrt{2}\log\left(\frac{10N(\mathcal{H}, \alpha, d_{\mathcal{H}})T^2}{3\delta}\right)^{1/2}\dim_{TE}(\mathcal{H}, \ell, \epsilon_T^{\mathcal{H}})T^{3/2} + 6\dim_{TE}(\mathcal{H}, \ell, \epsilon_T^{\mathcal{H}})},$$

where the last inequality follows since $\epsilon_T^{\mathcal{H}} \leq 1/T^2$ by definition. $\qquad\square$

The leading term in the regret bound is of order

$$T^{3/4}\left(\log N(\mathcal{H}, \epsilon_T^{\mathcal{H}}, d_{\mathcal{H}})\right)^{1/4}\sqrt{\dim_{TE}(\mathcal{H}, \ell, \epsilon_T^{\mathcal{H}})}.$$

**Remark 3.** As noted earlier on, while the order $\widetilde{O}(T^{3/4})$ on the time horizon $T$ may appear suboptimal compared to classical $\widetilde{O}(\sqrt{T})$ optimal rates for bandit learning with direct reward feedback, this slower rate is in fact a principled consequence of our minimal assumptions. Specifically, our analysis makes no structural assumptions on the verifier loss $\ell$ beyond boundedness. If we have more structural knowledge of $\ell$, say, that it is $\alpha$-strongly convex, then the bound can be tightened to match the optimal order $\widetilde{O}(\sqrt{T})$. A notable instance is when $\ell$ is a squared loss. A refined analysis on the drift of conditional mean losses allows us to choose the confidence parameters $\beta_T$ for the version spaces to be of order $\widetilde{O}(\log(1/\delta))$, which results in the tight $\widetilde{O}(\sqrt{T})$ regret rate.

### B.4 RATE-OPTIMAL BOUND FOR SQUARED LOSS

In this section, we consider a special case of a verifier $\ell$, taking the discriminative example introduced in Section 4.3 and detailed in Section C.3.

**Theorem 4.** *Suppose* $r_\eta(a) = \mathbb{E}_{o\sim f_\eta(a)}[g(a, o)]$ *for some known* $g : \mathcal{A} \times \mathcal{O} \to [0, 1]$ *and* $\ell(a, o, \eta) = (g(a, o) - r_\eta(a))^2 = (g(a, o) - \mathbb{E}_{o'\sim f_\eta(a)}[g(a, o')])^2$. *Suppose for all* $t \in \mathbb{N}$, $g(A_t, O_t) - \mathbb{E}_{O'\sim f_\eta(A_t)}[g(A_t, O')]$ *conditioned on* $(\mathcal{G}_t, A_t)$ *is* $\sigma$-*sub-Gaussian. For all* $T \in \mathbb{N}$, *the regret of LLF-UCB satisfies*

$$\text{Regret}(T) \leq \widetilde{O}\left(\sqrt{T\log N(\mathcal{H}, \epsilon_T^{\mathcal{H}}, d_{\mathcal{H}})\dim_{TE}(\mathcal{H}, \ell, \epsilon_T^{\mathcal{H}})}\right),$$

*where* $N(\mathcal{H}, \epsilon_T^{\mathcal{H}}, d_{\mathcal{H}})$ *denotes the* $\epsilon_t^{\mathcal{H}}$-*covering number of* $\mathcal{H}$ *based on the pseudo-metric* $d_{\mathcal{H}}$, $\dim_{TE}(\mathcal{H}, \ell, \epsilon_T^{\mathcal{H}})$ *denotes the* $\epsilon_T^{\mathcal{H}}$-*transfer eluder dimension of* $\mathcal{H}$, *and* $\epsilon_T^{\mathcal{H}} = \max\left\{\frac{1}{T^2}, \min_{a\in\mathcal{A}}\inf\{|r_\eta(a) - r^*(a)| : \eta \in \mathcal{H}, \eta \neq \eta^*\}\right\}$.

## C PROOFS FOR SUPPORTING LEMMAS AND PROPOSITIONS

### C.1 PROOF FOR PROPOSITION 1

*Proof.* Let $\tilde{\ell} = C_F\ell$. Let $d_{TE} = \dim_{TE}(\mathcal{H}, \tilde{\ell}, \epsilon)$ be the shorthand for the $\epsilon$-transfer eluder dimension of $\mathcal{H}$ with respect to $\tilde{\ell}$. Then, there exists a length $d_{TE}$ sequence of elements in $\mathcal{A}$ such

that for some $\tilde{\epsilon} \geq \epsilon$, every action element is $\tilde{\epsilon}$-transfer independent of its predecessors. We denote such a sequence as $(a_0, \ldots, a_{d_{TE}-1})$. By definition of the transfer eluder dimension, for any $k \in \{0, \ldots, d_{TE} - 2\}$, there exists a pair of hypotheses $\eta, \eta' \in \mathcal{H}$ satisfying

$$\sum_{i=0}^{k} \left( \mathbb{E}_{o \sim f_{\eta'}}(a_i)[\tilde{\ell}(a_i, o, \eta)] - \tilde{\ell}_{\eta'}^{\min}(a_i) \right) \leq \tilde{\epsilon}^2$$

but $|r_\eta(a_{k+1}) - r_{\eta'}(a_{k+1})| > \tilde{\epsilon}$. Using the definition for reward-discriminative verifiers,

$$\sum_{i=0}^{k} \left( r_\eta(a_i) - r_{\eta'}(a_i) \right)^2 \leq C_F \sum_{i=0}^{k} \left( \mathbb{E}_{o \sim f_{\eta'}}(a_i)[\ell(a_i, o, \eta)] - \ell_{\eta'}^{\min}(a_i) \right)$$

$$= \sum_{i=0}^{k} \left( \mathbb{E}_{o \sim f_{\eta'}}(a_i)[\tilde{\ell}(a_i, o, \eta)] - \tilde{\ell}_{\eta'}^{\min}(a_i) \right) \leq \tilde{\epsilon}^2.$$

By the definition of the (regular) eluder dimension, every action in the sequence $(a_0, \ldots, a_{d_{TE}-1})$ is $\epsilon$-independent of its predecessors. Therefore, $d_{TE} \leq \dim_E(\mathcal{R}, \epsilon)$ since the latter is the length of the longest sequence of independent actions. We may conclude that $\dim_E(\mathcal{R}, \epsilon) \geq \dim_{TE}(\mathcal{H}, C_F \ell, \epsilon)$.

$\square$

## C.2 PROOF FOR LEMMA 5

**Lemma 5.** *Consider some $\bar{\mathcal{H}}$. Suppose $\min_{\pi \in \Pi} \max_{\eta \in \bar{\mathcal{H}}} r_\eta(\pi_\eta) - r_\eta(\pi) = 0$. Let $\hat{\pi}$ be a minimizer. Let $\mathcal{A}_\eta^*$ denote the set of optimal actions with respect to $r_\eta$. Then $\mathrm{supp}(\hat{\pi}) \subseteq \mathcal{A}_\eta^*$, for all $\eta \in \bar{\mathcal{H}}$.*

*Proof.* We prove by contradiction. Suppose $\hat{\pi}$ takes some action $a'$ outside of $\mathcal{A}_\eta^*$ for some $\eta \in \bar{\mathcal{H}}$ with probability $p'$. Let $\pi' = \hat{\pi} - p'\mathbb{1}[a = a'] + p'\mathrm{Unif}[a \in \mathcal{A}_\eta^*]$. Then it follows $r_\eta(\pi') > r_\eta(\hat{\pi})$, which is a contradiction. Therefore, $\mathrm{supp}(\hat{\pi}) \subseteq \mathcal{A}_\eta^*$, for all $\eta \in \mathcal{H}$. $\square$

## C.3 PROOF OF THE DISCRIMINATIVE FEEDBACK EXAMPLE

Suppose $r_\eta(a) = \mathbb{E}_{o \sim f_\eta(a)}[g(a, o)]$ for some known $g : \mathcal{A} \times \mathcal{O} \to [0, 1]$. Note that the reward mapping $\eta \mapsto r_\eta$ is known, but the reward function itself is still hidden from the agent (since $\eta^*$ is unknown). We define $\ell(a, o, \eta) := (g(a, o) - r_\eta(a))^2 = (g(a, o) - \mathbb{E}_{o' \sim f_\eta(a)}[g(a, o')])^2$, which gives

$$\mathbb{E}_{o \sim f_\eta(a)}[\ell(a, o, \eta')] = \mathbb{E}_{o \sim f_\eta(a)}\left[ (g(a, o) - \mathbb{E}_{o' \sim f_{\eta'}(a)}[g(a, o')])^2 \right].$$

One can easily verify that $\eta \in \arg\min_{\eta' \in \mathcal{H}} \mathbb{E}_{o \sim f_\eta(a)}[\ell(a, o, \eta')]$. With this definition, we have that

$$|r_\eta(a) - r_{\eta'}(a)|^2 = (\mathbb{E}_{o \sim f_\eta(a)}[g(a, o)] - \mathbb{E}_{o \sim f_{\eta'}(a)}[g(a, o)])^2$$

$$= (\mathbb{E}_{o \sim f_\eta(a)}[g(a, o) - \mathbb{E}_{o' \sim f_{\eta'}(a)}[g(a, o')]])^2$$

$$\leq \mathbb{E}_{o \sim f_\eta(a)}[(g(a, o) - \mathbb{E}_{o' \sim f_{\eta'}(a)}[g(a, o')])^2]$$

$$= \mathbb{E}_{o \sim f_\eta(a)}[\ell(a, o, \eta')]$$

This shows the feedback is discriminative.

## C.4 PROOF OF REASONING EXAMPLE

**binary indicator of whether all steps are correct** This problem is equivalent to a bandit problem with $|\mathcal{S}|^L$ arms. Here $f_\eta(a) = r(a)$, so the transfer eluder dimension reduces to the standard eluder dimension, which is bounded by the size of the action space.

**index of the first incorrect step**     Here we prove for $\epsilon < 1/2L$. Given the rubric of $\eta^*$, partition the action space into $L$ sets, where $\mathcal{A}_l = \{(s_1, \ldots, s_L) | s_1, \ldots, s_{l-1}$ are correct and $s_l$ is incorrect$\}$ for $l = 1, \ldots, L$, where $\mathcal{A}_0$ denotes sequences where $s_1$ is incorrect. By this definition, we have $\mathcal{A}_i \bigcap \mathcal{A}_j = \emptyset$, for $i \neq j$, and $\mathcal{A}^* \bigcup (\bigcup_{l=1}^{L} \mathcal{A}_l) = \mathcal{A}$, where $\mathcal{A}^* = \{a^*\}$

Suppose we have an independent action sequence $(a_1, \ldots, a_K)$ in the sense of Definition 3 where each action is $\epsilon$-independent of their predecessors. We show it can have no more than $|\mathcal{S}|$ actions from each $\mathcal{A}_l$ for $l \in [1, L]$. By definition of the feedback, for $a \in \mathcal{A}_l$, $f_\eta^*(a) = l$. Suppose we have more than $|\mathcal{S}|$ actions from $\mathcal{A}_l$. It implies that a token must be used twice at the $l$th position. Say it's $s_l$ and it's shared by $a^1, a^2 \in \mathcal{A}_l$. Then we show $a^2$ is $\epsilon$-dependent on $a^1$ when $\epsilon < 1/L$. For $\eta \in \mathcal{H}$, satisfying $\mathbb{E}_{o \sim f^*(a^0)}[|o - f_\eta(a^0)|^2 / L^2] = |l - f_\eta(a^0)|^2 / L^2 \leq \epsilon^2$, we have $l - L\epsilon \leq f_\eta(a^0) \leq l + L\epsilon$. Since $\epsilon < 1/2L$ and $f_\eta(a^0)$ is an integer, this implies $f_\eta(a^0) = l$. That is, for such an $\eta$ satisfying the constraint given by $a^0$, $s_l$ is incorrect. This implies $f_\eta(a^1) \leq l$. Therefore, $r_\eta(a^0) = r_\eta(a^1) = 0$.

Therefore, the length of independent action sequences is bounded by $|\mathcal{S}|L + |\mathcal{A}^*| = |\mathcal{S}|L + 1$.

**give correction for the first mistake**     In this case, the feedback not only returns the index of the first incorrect step $l$, but also reveals the correct reasoning action $s_l^*$. Let $a_\eta^* = (s_1(\eta), \ldots, s_L(\eta))$ denote the $L$ reasoning steps based on the hypothesis $\eta$. The reward function of any action $a$ and hypothesis $\eta$ is $r_\eta(a) = \mathbb{I}\{a_\eta^* = a\}$. For an action $a = (s_1, \ldots, s_L)$ and feedback $o := (l, s_l(\eta))$ generated based on $f_\eta(a)$, we have $s_j = s_j(\eta)$ for all $j < l$ and $s_l \neq s_l(\eta)$. Now, given any feedback $o := (l, s_l^*)$, we define the following loss $\ell(a, o, \eta) = \frac{1}{L} \left( \sum_{j=1}^{l-1} \mathbb{I}\{s_j(\eta) = s_j\} + \mathbb{I}\{s_l(\eta) = s_l^*\} \right)$. This verifier loss evaluates whether $\eta$ and $\eta'$ have the same first $l$ reasoning steps.

For $\epsilon < 1$, suppose an action sequence $(a_1, \ldots, a_K)$ where each action is $\epsilon$-independent of their predecessors. If action $a$ is $\epsilon$-independent, there exists $\eta, \eta'$ such that $\sum_{i=1}^{K} \mathbb{E}_{o_i \sim f_{\eta'}(a)}[l(a_i, o_i, \eta)] \leq \epsilon$ and $|r_\eta(a) - r_{\eta'}(a)| > \epsilon$. By definition of the feedback and loss, we know $\eta, \eta'$ have the same initial $\max_i l_i$ reasoning steps. However, we know that $r_\eta(a) \neq r_{\eta'}(a)$ indicating at least one index $l > \max_i l_i$ where $s_l \in \{s_l(\eta), s_l(\eta')\}$ and $s_l(\eta) \neq s_l(\eta')$, resulting in feedback $o = (l, s_l(\eta'))$ for $a$. Thus, the sequence of indices in feedback $o_1, o_2, \ldots$ is monotonic. As we have $L$ reasoning steps, for any pair $\eta, \eta'$, the sequence length is bounded by $L$.

**demonstration**     Here, the feedback directly demonstrates correct reasoning sequence $a^* = (s_1^*, \ldots, s_L^*)$ and is independent of the agent's action sequence. For action $a = (s_1, \ldots, s_L)$ and hypothesis $\eta$, we define the loss as $\ell(a, o, \eta) = \mathbb{I}\{o = a_\eta^*\}$. Therefore, for any $\eta, \eta'$ and $\epsilon < 1$, if $a$ satisfies: $\mathbb{E}_{o \sim f_{\eta'}(a)} \ell(a, o, \eta) \leq \epsilon$, we have $a_\eta^* = a_{\eta'}^*$, implying $r_\eta(a) = r_{\eta'}(a)$ for all $a \in |\mathcal{S}|^L$ and a transfer Eluder dimension of 1.

## C.5   Proof for Lemma 6

**Lemma 6.** *Suppose for some $t_0 \geq 0$, we have that $\min_{\pi \in \Pi} \max_{\eta \in \mathcal{H}_{t_0}} |r_\eta(\pi_\eta) - r_\eta(\pi)| = 0$ in Algorithm 1. Then for all $t > t_0$, $\min_{\pi \in \Pi} \max_{\eta \in \mathcal{H}_t} |r_\eta(\pi_\eta) - r_\eta(\pi)| = 0$.*

*Proof.* We prove by induction. Suppose the conclusion holds for $t > t_0$, we prove that it holds for $t + 1$ as well. At time $t$, the induction hypothesis implies that $\min_{\pi \in \Pi} \max_{\eta \in \mathcal{H}_t} |r_\eta(\pi_\eta) - r_\eta(\pi)| = 0$. Since $\mathcal{H}_{t+1} \subseteq \mathcal{H}_t$, $\max_{\eta \in \mathcal{H}_{t+1}} |r_\eta(\pi_\eta) - r_\eta(\pi)| \leq \max_{\eta \in \mathcal{H}_t} |r_\eta(\pi_\eta) - r_\eta(\pi)|$ for all $\pi \in \Pi$. Thus, $\min_{\pi \in \Pi} \max_{\eta \in \mathcal{H}_{t+1}} |r_\eta(\pi_\eta) - r_\eta(\pi)| \leq \min_{\pi \in \Pi} \max_{\eta \in \mathcal{H}_t} |r_\eta(\pi_\eta) - r_\eta(\pi)| = 0$.

$\square$

# D   Extensions

## D.1   Special Case of Reward-Agnostic Feedback

Text feedback may contain information beyond what is relevant to the reward. In particular, one could imagine a special case, where feedback does not reveal much about the reward, but still provides enough to identify an optimal action over time. One simple example is when the feedback

directly reveals the optimal action, regardless of the action chosen. In this case, the transfer eluder dimension as defined could be arbitrarily large, but ideally an efficient LLF agent should choose the optimal action in the following steps instead of trying to identify the mean reward for each action.

## D.2 Extension to Contextual Bandits and Stateful Interactions

Our formulation can be modified slightly to accommodate learning with a context. In a contextual problem, a Markov process $X_t$ independently takes values in a set $\mathcal{X}$ that the agent views as contexts. We may define the full set of actions to be the set of context-action pairs $\mathcal{A} := \{(x, a) : x \in \mathcal{X}, a \in \mathcal{A}(x)\}$, where $\mathcal{A}(x)$ is the set of available actions under the context $x$. Instead of having a fixed action space $\mathcal{A}$ across time, consider time-varying action sets $\mathcal{A}_t := \{(X_t, a) : a \in \mathcal{A}(X_t)\}$. At each time $t$, an action $a_t \in \mathcal{A}_t$ will be selected. In accordance, the policy $\pi = \{\pi_t | t \in \mathbb{N}\}$ is now a sequence of functions indexed by time, each mapping the history $H_t = (\mathcal{A}_0, A_0, R_0, \dots, \mathcal{A}_{t-1}, A_{t-1}, R_{t-1}, \mathcal{A}_t)$ to a distribution over $\mathcal{A}$ with support $\mathcal{A}_t$. Our analysis for the context-free setting directly carries over.

While our framework focuses on stateless settings similar to bandits (and contextual bandits in the extension discussed above), extending this formulation to stateful interactions is an exciting avenue for future work. A most straightforward extension is treating history as contexts and following the contextual extension described above. However, this will induce an exponential explosion in the state space, and the regret guarantees will become too loose. A more careful treatment might involve formulating LLF problems in an MDP setting, and designing algorithms capable of deep exploration.

## D.3 Alternative Formulation of Feedback Generation

The LLF formulation we have presented so far assumes that feedback arises from a fixed mapping $\eta \mapsto f_\eta$ with each hypothesis $\eta \in \mathcal{H}$. While this "model-based" view simplifies both the design of exploration strategies and the complexity analysis via the transfer eluder dimension, it imposes a structural constraint that may be too restrictive in settings where feedback is generated by a more complex or even adversarial process. An alternative, entirely "model-free" formulation allows feedback to be generated arbitrarily from an oracle in a streaming fashion, without the need to explicitly model a feedback mapping $\eta \mapsto f_\eta$. Concretely, at each time $t$, the agent executes an action $A_t \in \mathcal{A}$ and observes feedback $O_t \in \mathcal{O}$. We denote the history of interactions as $I_t = (A_0, O_0, \dots, A_t, O_t)$ and write $\mathcal{I}$ for the set of all possible histories. A (history-dependent) policy $\pi : \mathcal{I} \to \Delta(\mathcal{A})$ maps each history $h \in \mathcal{I}$ to a distribution over actions.

This streaming-oracle perspective subsumes both stochastic and adversarial feedback models, and can capture scenarios where the dependence on $\eta$ is unknown or too complex to parameterize. In this setting, one must replace the hypothesis-indexed complexity measures by complexity metrics defined directly over the space of oracles or possible histories. Although this general approach will likely incur additional technical overhead, it also broadens the applicability of our LLF framework to encompass richer feedback protocols beyond the hypothesis-testing paradigm. An interesting future direction is to develop performance guarantees under the more general feedback generation model.

# E Implementing HELiX with Large Language Models

We provide a practical implementation of HELiX using an LLM. LLMs with advanced reasoning capabilities can produce chain-of-thoughts that often contain guesses and reasoning traces of the environment (Wei et al., 2022; Guo et al., 2025; Gandhi et al., 2025). We propose to leverage LLMs' knowledge about the world to enhance decision-making. In particular, we treat an LLM's thinking tokens before deciding on an action as "hypotheses". These thinking tokens can be sampled by prompting the LLM to output its reasoning before an action with prompts in the form of "`<Thought>` `<Action>`".

We provide the pseudocode for the practical implementation in Algorithm 2 and illustrate a corresponding flow-graph in Figure 4. The algorithm takes as inputs the following LLM-based components:

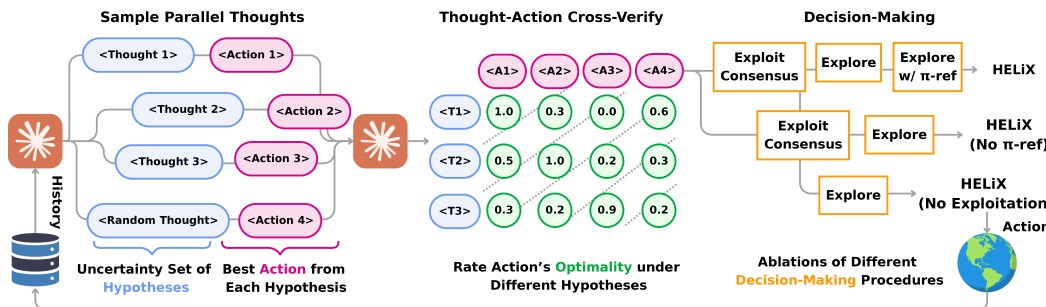

Figure 4: **Thought Sampling with Cross-Verify.** Our algorithm extends the traditional paradigm of model-based exploration to the LLM setting. Here, the "model" is represented by the LLM's intermediate thoughts, which we interpret as their hypotheses about the external world. We ground this thought-then-act behavior in the interactive decision-making framework and introduce a new algorithm that conducts efficient exploration from language feedback.

1. $\pi_{\mathrm{LLM}} : \bigcup_{t=0}^{\infty} (\mathcal{A} \times \mathcal{O})^t \to \Delta(\mathcal{H} \times \mathcal{A})$. This is an LLM with a chain-of-thought prompt that asks it to analyze the current observation through thinking tokens and produce a valid action. We may view this policy as producing the best action conditioned on a hypothesis consistent with the feedback history.

2. $\pi_{\mathrm{ref}} : \varnothing \to \Delta(\mathcal{A})$. This is a user-provided reference policy to sample actions, analogous to a baseline policy. The design of reference policy may vary. In this work, we adopt a random reference policy by asking an LLM to produce a set of random actions that are different from those generated by $\pi_{\mathrm{LLM}}$.

3. $R_{\mathrm{LLM}} : \mathcal{H} \times \mathcal{A} \to [0, 1]$. This is a reward mapping to evaluate how good/bad the action is under a given hypothesis. We implement this by prompting an LLM to score an action conditioned on a sampled hypothesis. This can be viewed as a hypothesis-conditioned reward model.

**Approximation of Feedback-consistent Hypotheses and Policy Space.** HELiX (Algorithm 1) maintains a hypothesis space $\mathcal{H}_t$ at iteration $t$, which contains all hypotheses $\eta$ that are consistent with observed feedback. Then, HELiX searches over all possible policies by computing $\pi_p$ and $\pi_o$. We approximate these two steps with finite sets of candidates, $\hat{\mathcal{H}}_t$ and $\hat{\mathcal{A}}_t$, respectively. We make the assumption that state-of-the-art LLMs are capable of producing valid hypotheses when instructed with a chain-of-thought prompt and history. In other words, they provide hypotheses that are plausible explanations of the interaction history of actions and feedback. At each step, we use $\pi_{\mathrm{LLM}}$ to produce thought-conditioned actions. We first ask the LLM to generate a diverse set of hypotheses. For each hypothesis, we prompt the LLM to generate corresponding optimal actions. Unlike a common chain-of-thought approach that asks LLM to produce only one thought and one action, we ask the LLM to output $N$ thoughts and actions. This set of thoughts accounts for the agent's uncertainty about the environment. In addition, we use $\pi_{\mathrm{ref}}$ to propose $M$ random valid actions. For computational efficiency, we sample these $N$ hypotheses and actions in one LLM call rather than $N$ calls, introducing conditional dependencies between them (the same holds when sampling the $M$ random actions). These LLM calls produce an approximate hypotheses space $\hat{\mathcal{H}}_t$ of size $N$ and an approximate policy space $\hat{\mathcal{A}}_t$ (of deterministic actions) of size $N + M$.

**Thought Cross-verify.** In Algorithm 2, we approximate the minimax and maximization steps in Algorithm 1 with $\hat{\mathcal{H}}_t$ and $\hat{\mathcal{A}}_t$. Concretely, we construct a score matrix $S_t \in [0, 1]^{N \times (N+M)}$ whose entries $[S_t]_{\eta,a}$ correspond to the reward of hypothesis-action pairs $(\eta, a)$. The rows of this score matrix correspond to hypotheses in $\hat{\mathcal{H}}_t$ and columns correspond to actions in $\hat{\mathcal{A}}_t$. This matrix is visualized in the middle portion in Figure 4. We use the reward mapping $R_{\mathrm{LLM}}$ to produce scores. The diagonal entries of $S_t$ are close to 1.0 because the action $a_i$ conditionally sampled from $\eta_i$ should be scored the highest under $\eta_i$. If some action $a$ is deemed optimal across all sampled hypotheses, we follow this consensus choice (Fig. 5 Stage 1). Conversely, when the hypotheses disagree, we

---

**Algorithm 2** HELiX (Practical Version with LLMs)

1: **Input** $T, \pi_{\text{LLM}}, \pi_{\text{ref}}, R_{\text{LLM}}, N, M$
2: initialize $A_0, \eta_0 \sim \pi_{\text{LLM}}()$
3: **for** $t = 0, 1, \ldots, T-1$ **do**
4:      execute $A_t$, observe $O_t$
5:      $\hat{\mathcal{A}}_t, \hat{\mathcal{H}}_t \leftarrow \{\pi_{\text{LLM}}(\{A_\tau, O_\tau\}^t_{\tau=0}) \mid i = 1, \ldots, N\}$ // Sample $N$ thought-action
6:      $\hat{\mathcal{A}}_t \leftarrow \hat{\mathcal{A}}_t \cup \{\pi_{\text{ref}}(\cdot) \mid i = 1, \ldots, M\}$ // Sample $M$ random actions from $\pi_{\text{ref}}$
7:      // Thought-action cross-verify (for checking if Exploitation step should be triggered.)
8:      compute score matrix $[S_t]$ where $[S_t]_{\eta,a} \leftarrow R_{\text{LLM}}(\eta, a)$ for $a \in \hat{\mathcal{A}}_t, \eta \in \hat{\mathcal{H}}_t$
9:      $\hat{\mathcal{A}}^*_t \leftarrow \bigcap_{\eta \in \hat{\mathcal{H}}_t} \arg\max [S_t]_\eta$ // Set of actions optimal to all hypotheses in $\hat{\mathcal{H}}_t$.
10:      **if** $\hat{\mathcal{A}}^*_t \neq \varnothing$ **then**
11:          $A_{t+1} \leftarrow$ tie-break-choose$(\hat{\mathcal{A}}^*_t)$   // Exploitation step: check consensus
12:      **else**
13:          $\tilde{\mathcal{H}}_t \leftarrow \arg\max_{\eta \in \hat{\mathcal{H}}_t} \left( \max [S_t]_\eta \right)$     // Exploration step: UCB-inspired
14:          $A_{t+1} \leftarrow \arg\max_{a \in \hat{\mathcal{A}}_t} \left( \max_{\eta \in \tilde{H}_t} \left[ [S_t]_{\eta,a} - \mathbb{E}_{\tilde{a} \sim \pi_{\text{ref}}} \left[ [S_t]_{\eta,\tilde{a}} \right] \right] \right)$
15:      **end if**
16: **end for**

---

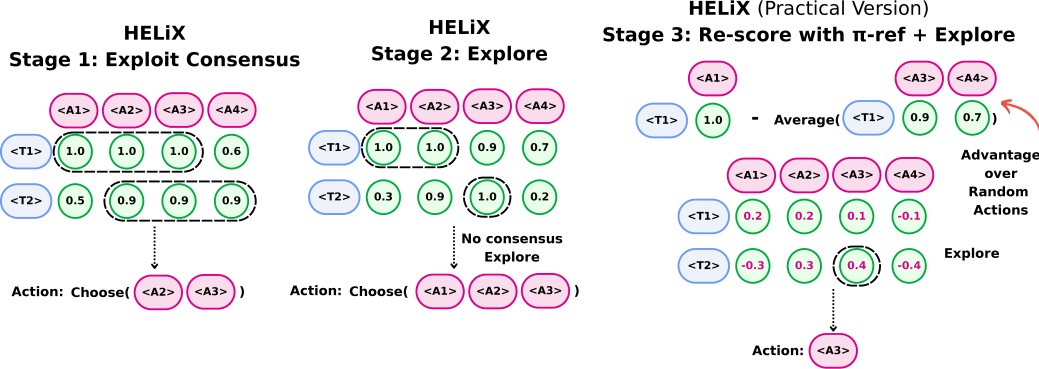

Figure 5: **HELiX Algorithm.** The HELiX algorithm has three steps. First, if the highest-scoring actions across all generated hypotheses coincide, the algorithm performs an exploitation step. Otherwise, it performs an exploration step by retaining only the hypotheses whose optimal actions achieve the highest scores. In the absence of a random policy $\pi_{\text{ref}}$, HELiX chooses an action using a predefined tie-breaking rule. When a random policy $\pi_{\text{ref}}$ is available, the algorithm adjusts the score of each action by subtracting the average score of actions under $\pi_{\text{ref}}$. In the example above, A3 and A4 are random actions sampled from $\pi_{\text{ref}}$.

select the most optimistic action to encourage exploration (Fig. 5 Stage 2). This distinction between consensus and disagreement forms the backbone of our exploration–exploitation strategy.

**Exploitation Step.** Given the score matrix $S_t$, we first check whether the exploitation step in Algorithm 1 is triggered. Specifically, if a given action $a^\star$ satisfies $R_{\text{LLM}}(\eta, a^\star) \geq R_{\text{LLM}}(\eta, a)$ for all $\eta \in H_t$ and $a \in A_t$, then $a^*$ is identified as a consensus action and exploited immediately. This corresponds to the exploitation step in the theoretical Algorithm 1. By Lemma 5, if an action solves the minimax problem, it must also be an optimal action for all remaining hypotheses simultaneously. If there are multiple consensus actions, we perform tie-breaking detailed below. In Figure 5, this step is implemented as a set intersection operation over the sets of highest-scoring actions from each hypothesis.

**Exploration Step.** If no consensus action exists, we conduct the exploration step in Algorithm 1. We first eliminate hypotheses whose highest score is lower than those of other hypotheses. This

implements exploration using the optimism in the face of uncertainty principle (Auer et al., 2002), where we only keep the most optimistic hypotheses. After the hypothesis elimination, if only one hypothesis remains, then we execute the best action under that hypothesis. If there are more than one hypothesis left, we apply a tie-breaking step by re-scoring with a reference policy. The re-scoring or re-centering step is widely used in RL, such as baseline methods (Weaver & Tao, 2013; Sutton & Barto, 2018), ReMax (Li et al., 2023), RLOO (Ahmadian et al., 2024), and GRPO (Guo et al., 2025). This procedure interprets the score as the advantage of an action $a$ relative to those sampled from a reference policy $\pi_{\text{ref}}$, under a given hypothesis $\eta$. There are multiple reasons why an advantage is useful for tie-breaking: 1) LLMs may not score consistently across hypotheses. Comparing score differences can help cancel out these inconsistencies. 2) When we use a uniformly random $\pi_{\text{ref}}$, the advantage implicitly examines the quality of the hypotheses and favors more discriminative ones. A permissive hypothesis that assigns approximately the same score to all actions (e.g., "Fire a shot anywhere on the map") lacks discriminative power. In contrast, a discriminative hypothesis assigns higher scores to actions that align with its intent (e.g., "Fire a shot along the edge of the map"), yielding a higher advantage over random actions. With re-scoring, we favor actions with high advantages over random actions.

**A Note on Tie-breaking.** If ties remain after re-scoring, we further tie-break by preferring hypotheses and actions generated earlier in the output. This is due to empirical observations that LLMs have a preference to produce the best plan and action first, followed by less likely plans and actions (Dracheva & Phillips, 2024).

## F    EXPERIMENT DETAILS

In this section, we present experiment details of HELiX in three environments that require learning from language feedback.

### F.1    BASELINES

We consider the following agents for comparison. In addition to HELiX, we implement two of its variants with slightly different action selection procedures.

**Greedy.** This agent generates one hypothesis and one action, and returns that action immediately. This is a ReAct-style baseline.

**HELiX (No exploitation step).** This baseline agent conducts optimistic exploration without the consensus-based exploitation step. We demonstrate that optimistic exploration alone is insufficient in our setup. We use thought sampling to generate $N + M$ actions and $N$ hypotheses, followed by cross-verification that scores each action under every hypothesis. Unlike in HELiX, we directly select actions with the highest score across all hypotheses. If there are multiple actions, we tie-break by preferring hypotheses and actions generated earlier in the output.

**HELiX (No $\pi_{\text{ref}}$).** This variant of HELiX includes thought sampling, cross-verification, and the exploitation step, but omits the re-scoring step using $\pi_{\text{ref}}$. If the exploitation step is not triggered, we perform the exploration step without re-scoring using random actions sampled from $\pi_{\text{ref}}$. The benefit of using $\pi_{\text{ref}}$ is entirely empirical and depends on its specific instantiation.

### F.2    EXPERIMENTAL SETUP

We conduct experiments in the following three gym environments proposed in Tajwar et al. (2025).

**WORDLE** In each scenario, the environment selects a secret 5-letter word from a predefined dictionary. The agent attempts to guess the word, receiving feedback after each guess indicating correct letters and their positions. In our experiment, we used 50 scenarios to evaluate all agents. To better illustrate Example 2 in Section 4.2, we modify the feedback from the original environment to only contain information about the first incorrect character. For example, if the target word is "totem" and the agent's guess is "apple", the feedback is "The first letter 'a' is incorrect." Considering that this feedback provides less information than typical feedback in wordle, we allow the agents to make 10 attempts before termination.

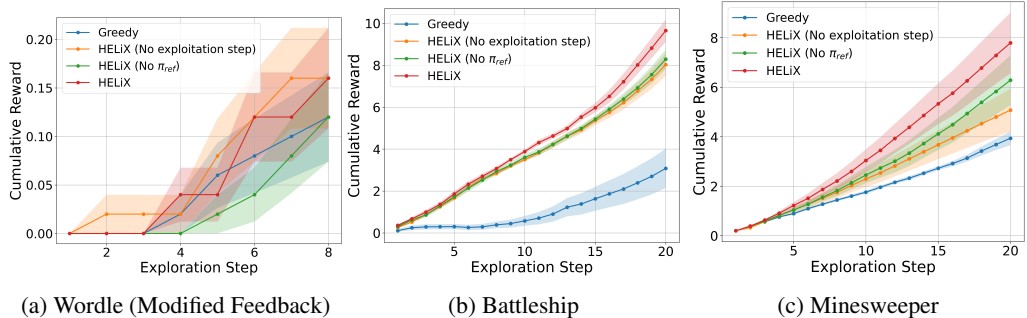

(a) Wordle (Modified Feedback)  (b) Battleship  (c) Minesweeper

Figure 6: We show the cumulative reward that the agent is able to obtain during a fixed number of interactions with the environment. Shaded area represents the standard error of cumulative reward across different scenarios. We use Claude-Sonnet-3.5 v2 for the experiment.

**BATTLESHIP** Battleship is a 2D grid environment where the agent must locate and sink three hidden ships within 20 turns. The agent fires at one cell per turn and receives hit/miss feedback, ship type (5-cell ship, 4-cell ship, and 3-cell ship), and a map showing all previous hits and misses. Success requires strategic exploration to find ships and exploitation to sink them efficiently. We use 20 scenarios (maps of ship layout) to evaluate all agents. We use a hidden per-step reward to evaluate an agent's performance. For instance, the feedback "a ship was hit but not sunk" corresponds to 0.5 point. We do not communicate this numerical reward information to the agent.

**MINESWEEPER** Minesweeper is a 2D grid puzzle with hidden mines. At each turn, the agent chooses to reveal one cell, aiming to uncover all safe cells within 20 turns without hitting a mine. Revealed cells show the number of adjacent mines, and a '0' triggers automatic revelations of surrounding safe cells. Sequential reasoning and updating of hypotheses based on observed clues are essential for success. Hidden rewards are calculated by assigning 0.2 to choosing a square that does not have a mine, and 1.0 to fully solving the game. Invalid moves incur a -0.2 penalty. The agent receives feedback in the form of a partially revealed map after each action.

### F.3 DISCUSSION OF RESULTS

We plot the cumulative reward as a function of the number of environment interaction steps on WORDLE, BATTLESHIP, and MINESWEEPER in Figure 6. We see that for all three environments, the base LLM, where we only greedily choose the first action, performs worse generally. In environments where information-gathering is more necessary, such as BATTLESHIP and MINESWEEPER, agents designed to conduct strategic explorations and exploitations tend to outperform the greedy base LLM by a large margin.

As shown, HELiX consistently outperforms both the greedy baseline and HELiX variants: HELiX (no exploitation step) and HELiX (no $\pi_{\text{ref}}$). In particular, in BATTLESHIP and MINESWEEPER, HELiX performs significantly better than the baselines. Although the theoretical version of our algorithm does not use $\pi_{\text{ref}}$, we have found that across these three environments, performing an explicit re-scoring is beneficial.

Although the initial results are promising, our practical implementation relies on assumptions that warrant discussion. We assume that the LLM can select an optimal action under a given hypothesis. We also assume that the LLM can produce fair scores across hypotheses for different actions. However, these assumptions may not hold for all LLMs (Shojaee et al., 2025), and further investigation is needed to validate them. Additionally, to capture the agent's uncertainty about the environment, we sample a set of hypotheses from the LLM. These hypotheses should be both diverse and faithful in reflecting the history of interactions. The extent to which existing LLMs can propose plausible hypotheses given historical information remains uncertain, with evidence pointing in both directions (Zhou et al., 2024; Si et al., 2024; Ghareeb et al., 2025). Our theory-inspired algorithm highlights key properties an LLM must exhibit to function effectively as a decision-making agent, one that autonomously learns from environment feedback, proposes hypotheses, and explores ac-

cordingly. Further research is needed to verify whether current LLMs possess these properties, and, if not, to determine what forms of training could instill them.

## F.4 DISCUSSION OF COMPUTATIONAL COST

As a practical implementation of our theoretical algorithm, HELiX incurs a computational cost on sampling from LLMs. At a first glance, building the score matrix through cross-verify could incur $K^2$ LLM calls. However, we could leverage parallelization and efficient sampling techniques to reduce the cost significantly.

**Parallelization of Thought Sampling and Verification.** In practical implementations, we sample thoughts and verify thoughts in parallel, reducing $O(K^2)$ to $O(K)$ LLM calls during thought-cross-verify and $O(K)$ to $O(1)$ calls during thought sampling.

**Efficient use of tokens through prefix Caching.** In practical implementations, we could leverage advanced inference techniques like prefix caching where instructions sharing the same prefix sequence can be stored and loaded as KV-cache without re-computing. In HELiX, many LLM calls share common observations and judgments, significantly reducing the actual tokens needed.

Since we mainly use the experiment to illustrate and instantiate one practical implementation, we do not implement prefix caching or parallel sampling. The table below demonstrates that even without such advanced techniques, just by exploring the environment better, we still avoid exponential cost blowup (instead of $9x$ tokens compared to the baseline, we incur $3.73x$ to $4.02x$ the token count of baseline).

| Method | Token Count | Ratio to Baseline |
|---|---|---|
| Baseline | $698, 873$ | 1 |
| HELiX (no exploitation step) (explore optimistically) | $3, 151, 173$ | $4.51$ ($K = 3$ hypotheses) |
| **HELiX** | $2, 812, 856$ | **4.02** ($K = 3$ hypotheses) |

Table 2: Token count comparisons on Battleship.

| Method | Token Count | Ratio to Baseline |
|---|---|---|
| Baseline | $553, 389$ | 1 |
| HELiX (no exploitation step) (explore optimistically) | $2, 538, 838$ | $4.59$ ($K = 3$ hypotheses) |
| **HELiX** | $2, 064, 156$ | **3.59** ($K = 3$ hypotheses) |

Table 3: Token count comparisons on Minesweeper.

To help give a sense of the computational overhead of our practical implementation of HELiX (Algorithm 2), we expand on some key problem-dependent parameters and hyperparameters. For detailed prompt format and reasoning traces, please see Appendix F.6. Given an LLF problem, practical HELiX needs an input of domain_description, action_space, and learning_task_instruction at the start of learning. This constitutes the initial prompt. At each round, given historical context $C$, HELiX goes through 2-3 LLM calls, whose contexts are outlined below:

1. Feedback-consistent hypothesis generation: the context includes $C$ along with the most recent domain_state and instruction to simultaneously propose num_actions diverse hypotheses and actions. This is done in one LLM call.

2. Thought cross-verify: the context includes $C$ along with the num_actions proposed hypotheses and actions from step 1 and instruction to evaluate each action under each hypothesis. This is done in one LLM call.

3. Exploration step: if $\pi_{\text{ref}}$ is enabled, the LLM is asked to propose additional num_ref_actions exploratory actions. The context includes $C$ along with the num_actions proposed actions from step 2 and instructions to propose actions different from those.

The remaining steps do not need LLM calls; these include: consensus check, UCB elimination based on scores from step 2, and advantage-based exploration with tie-breaking. Tunable hyperparameters include num_actions, whether to use $\pi_{\text{ref}}$, and num_ref_actions.

| | |
|---|---|
| Assumption 3 satisfcation (trajectory-averaged) | 96.26% |
| Assumption 3 satisfcation (step-averaged) | 95.98% |

Table 4: The empirical rate at which Assumption 3 is satisfied in Battleship.

### F.5 ASSUMPTION SATISFACTION

To give a sense of how Assumption 3 connects to our experiments, we evaluate a key implication of it: that the shrinking hypothesis set remains consistent with the true hypothesis along the trajectory. Concretely, if Assumption 3 approximately holds, then as the algorithm updates and prunes the hypothesis set, the true hypothesis should almost always remain included. To test this, we manually annotate 10 trajectories from the Battleship experiment. For each step in each trajectory, we check whether the candidate set of hypotheses produced by HELiX contained the true hypothesis. The following table reports (i) the average fraction of steps within a trajectory for which the true hypothesis is included (Assumption 3 satisfaction (trajectory-averaged)"), and (ii) the fraction of all annotated steps across trajectories for which the true hypothesis is included ("Assumption 3 satisfaction (step-averaged)"). Both metrics are above 95%, empirically supporting that Assumption 3 holds to a good approximation in our experiment.

### F.6 REASONING TRACES FOR BATTLESHIP

Here we show a concrete example on how the algorithm works and what the LLM generates.

---

**Transcript on Multi-Hypothesis Evaluation (Battleship)**

**Prompt:** Hit at D2! You have hit a ship, which occupies 4 cells in the grid. Here is how the board looks now:

```
      1    2    3    4    5    6
 A    M    M    M    M    .    M
 B    M    X    X    M    M    .
 C    M    X    X    .    M    .
 D    .    X    X    M    .    .
 E    .    .    X    .    .    .
 F    .    .    X    .    .    .
```

========================
Now you have a new task. You are a given a hypothesis (thought/instruction) and actions. You need to evaluate if how good or bad the action is given the hypothesis.
Hypothesis:
 Alternatively, a 2-cell ship might be placed horizontally in the A5-B6 area, as this is one of the few remaining areas large enough to fit it. A5 would be worth checking.
Rate all the actions indiviually based on whether the action is aligned with the hypothesis.
Action 1: <action>E2</action>
Action 2: <action>E5</action>
Action 3: <action>A5</action>
Make sure the score you assign is between 0 and 1. Please provide your scores in the following format: ...

**Response:**
Action 1 for the Hypothesis:
<think> E2 is not aligned with the hypothesis at all. The hypothesis suggests checking A5 area for the 2-cell ship, but E2 is in a completely different part of the board. Additionally, we already have significant ship presence in the middle columns. </think><score> 0.1 </score>
Action 2 for the Hypothesis:

---

<think> E5 is somewhat in the direction of exploring new areas for remaining ships, but it's not in the specific A5-B6 area mentioned in the hypothesis. It's not directly supporting the hypothesis's suggested strategy. </think><score> 0.3 </score>

Action 3 for the Hypothesis:

<think> A5 is exactly what the hypothesis suggests checking. It's directly aligned with the reasoning that the 2-cell ship might be placed horizontally in the A5-B6 area, and it's one of the few remaining unexplored areas that could fit a 2-cell ship. </think><score> 1.0 </score>

---

Transcript on Action Proposal (Battleship)

Prompt: Hit at A5! You have hit a ship, which occupies 5 cells in the grid.
Here is how the board looks now:

```
     1    2    3    4    5
A    .    .    X    X    X
B    .    X    X    X    X
C    .    M    X    X    .
D    .    .    M    .    .
E    .    .    M    .    .
```

Given the information above, please propose 3 actions, make sure each of them is a valid action in the environment, and is reasonable.
Make sure your thoughts and hypotheses are diverse and unique. However, actions are allowed to be the same if they are the best possible action under that hypothesis.
Please provide your actions in the following format:
Action 1: ¡think¿ ...</think><answer> your first action </answer>
...
Action 3: <think> ...</think><answer> your 3th action </answer>

Response:
Action 1: <think> Looking at the pattern of hits for the 5-cell ship, we can see it extends from A3 to A5. Since 5-cell ships are 5 cells long and we've found 3 cells, we should check A2 as it's likely part of the 5-cell ship.</think><answer> A2 </answer>

Action 2: <think> If the 5-cell ship is not extending westward, it might extend to the east. Even though this is less likely since it would make the ship overlap with known hits, we should consider A1 to eliminate this possibility.</think><answer> A1 </answer>

Action 3: <think> Since we've hit three cells of the ship (A3, A4, A5), and this ship is 5 cells long, if it's not extending horizontally, it must extend vertically. Therefore, checking C5 could reveal if the 5-cell ship extends downward from A5.</think><answer> C5 </answer>

### F.7   PROMPT TEMPLATES

---

**Propose Action Prompt ($\pi_{\text{LLM}}$)**

Given the information above, please propose some hypotheses and act according to those hypotheses.

You can propose at most {num_actions} hypotheses.

Please propose a reasonable number of hypotheses – each hypothesis represents what you think.

Please provide your actions in the following format:

Action 1: <think>...</think><answer>action 1</answer>

...

Action {num_actions}:   <think>...</think><answer>your {num_actions}th action</answer>

---

**Propose Exploration Action Prompt ($\pi_{\text{ref}}$)**

Given the information above, please propose {num_actions} completely different and unexpected actions. These should be valid in the environment but should explore unusual or creative approaches.

Try to think outside the box and propose actions that might not be immediately obvious or conventional.

Here are the actions you have already proposed:

{actions}

Please avoid proposing the same actions.

Please provide your actions in the following format:

Action 1: <think>...</think><answer>your first random/exploratory action</answer>

...

Action {num_actions}: <think>...</think><answer>your {num_actions}th random/exploratory action</answer>

---

**Hypothesis-Conditioned Value Function Prompt ($V_{\text{LLM}}$)**

{task description}

=========================

Now you have a new task. You are a given a hypothesis (thought/instruction) and actions. You need to evaluate how good or bad the action is given the hypothesis.

Hypothesis:
<think>
{hypothesis}
</think>

Rate all the actions indiviually based on whether the action is aligned with the hypothesis.

Action {action_idx}: <action>{action}</action>

Make sure the score you assign is between 0 and 1. Please provide your scores in the following format:

Action 1 for the Hypothesis:
<think>... </think>
<score>...</score>

...

Action {num_actions} for the Hypothesis:
<think>... </think>
<score>...</score>

---

### F.8 CODE IMPLEMENTATION

We provide a high-level code snippet that demonstrates how we implement the algorithm below. We omit the implementation details of methods involving LLM calls.

```python
import numpy as np

class HELiX:

    def select_action(self, observation, hypotheses, actions,
            random_actions):

        if random_actions is not None:
            actions = actions + random_actions  # evaluate on all actions

        # Create a matrix to store scores for each hypothesis-action pair
        score_matrix = np.zeros((len(hypotheses), len(actions)))

        # Fill the score matrix by evaluating each hypothesis-action pair
        for h_idx, hypothesis in enumerate(hypotheses):
            scores = self.evaluate_multi_hypotheses(observation,
                hypothesis, actions)
            score_matrix[h_idx] = scores

        # ======== Exploitation step: consensus check =======
        consensus_action = self.consensus_action(score_matrix, actions)
        if consensus_action is not None:
            return consensus_action

        # ====== UCB elimination ======
        score_matrix, hypotheses, actions = self.
            ucb_hypothesis_elimination(score_matrix.copy(), hypotheses,
            actions)

        # ====== (Re-scoring +) Exploration =====
        best_hypothesis, best_action, best_overall_score,
            best_action_indices = self.tie_breaking(score_matrix,
            hypotheses, actions)

        return best_action

    def consensus_action(self, score_matrix, actions):
        max_scores_per_row = np.max(score_matrix, axis=1)
        action_sets = []
        for i in range(score_matrix.shape[0]):
            action_sets.append(np.where(score_matrix[i] ==
                max_scores_per_row[i])[0].tolist())
        # Convert each sublist to a set
        action_sets = [set(actions) for actions in action_sets]

        # Find the intersection of all sets
        overlapped_actions = reduce(lambda x, y: x.intersection(y),
            action_sets)

        # Convert back to list if needed
        overlapped_actions_list = list(overlapped_actions)

        if len(overlapped_actions_list) == 0:
            return None
        else:
            # randomly choose one
            random_index = np.random.choice(len(overlapped_actions_list))
            return actions[overlapped_actions_list[random_index]]

    def tie_breaking(self, score_matrix, hypotheses, actions,
            random_actions=[]):

        # ====== Optional Re-Scoring =========
        # Calculate average scores only for random actions
        num_regular_actions = len(actions) - len(random_actions)
```

```
 57
 58        # avg(random actions)
 59        action_avg_scores = np.mean(score_matrix[:, num_regular_actions
              :], axis=1, keepdims=True)
 60        normalized_score_matrix = score_matrix - action_avg_scores
 61
 62        # eliminate hypothesis again, to prevent ties
 63        normalized_score_matrix, hypotheses, actions = self.
              ucb_hypothesis_elimination(normalized_score_matrix.copy(),
              hypotheses, actions)
 64
 65        best_hypothesis, best_action, best_overall_score,
              best_action_indices = self.two_tiered_argmax_sampling(
              normalized_score_matrix, hypotheses, actions
 66
 67        )
 68
 69        return best_hypothesis, best_action, best_overall_score,
              best_action_indices
 70
 71    def ucb_hypothesis_elimination(self, score_matrix, hypotheses,
          actions):
 72        # Get the maximum score for each row (hypothesis)
 73        max_scores_per_row = np.max(score_matrix, axis=1)
 74
 75        # Find the highest score value
 76        highest_score = np.max(max_scores_per_row)
 77
 78        # Get indices of rows that have the highest score
 79        selected_row_indices = np.where(max_scores_per_row ==
              highest_score)[0]
 80
 81        # Select the hypotheses corresponding to these rows
 82        selected_hypotheses = [hypotheses[i] for i in
              selected_row_indices]
 83        # we only eliminate hypotheses, not actions
 84
 85        # Create a new score matrix with only the selected rows and
              columns
 86        new_score_matrix = score_matrix[selected_row_indices, :]
 87
 88        return new_score_matrix, selected_hypotheses, actions
 89
 90    def two_tiered_argmax_sampling(self, score_matrix, hypotheses,
          actions):
 91        # we take the highest score hypothesis, then sample its highest
              action
 92        assert score_matrix.shape[0] == len(hypotheses)
 93
 94        best_hypo_idx = np.argmax(np.max(score_matrix, axis=1))  # bias
              towards first hypothesis
 95        best_action_idx = np.argmax(score_matrix[best_hypo_idx, :])
 96
 97        best_action = actions[best_action_idx]
 98        best_hypothesis = hypotheses[best_hypo_idx]
 99        best_overall_score = score_matrix[best_hypo_idx, best_action_idx]
100
101        return best_hypothesis, best_action, best_overall_score, [
              best_action_idx]
```

## G   FAQ FOR REVIEWERS

We compile a list of FAQs from previous interactions with reviewers, with the hope of resolving common questions and providing a clearer perspective on our contributions.

**Q:** This work assumes that the agent has access to an effective verifier. Is this assumption necessary or realistic?

**A (Verifier assumption):** This assumption is motivated by empirical evidence that LLMs are generally **stronger at verification than generation**. It captures the ability of an LLF agent to decode textual feedback and assess its consistency, rather than being a mere mathematical simplification. In practice, for instance, an LLM verifier can be prompted to judge whether an observed (action, feedback) pair is consistent with a text hypothesis, acting as the verifier loss. For the simplest case, the LLM judge outputs 0 if it deems these consistent, and 1 otherwise. This verifier loss satisfies the boundedness and consistency properties (Assumption 2). The unbiased feedback assumption (Assumption 3) can be satisfied by considering a large enough hypothesis space. Our experiments further support that these assumptions hold in practice.

Since classical no-regret literature has no formal structure or tools to analyze text-based interactions, some assumptions must be made to develop rigorous theoretical understanding in this space. Our work takes an initial step towards such an attempt by making assumptions on the verification capability of text models only, rather than on their generation capability. We would like to highlight that the verifier is a natural structure to reduce the problem to solvable problems and hence derive provable algorithms. In particular, we can view the verifier loss as a generalization of a loss of reward fitting in a classical bandit setting (e.g., one way is to identify $O = R$ as the observed reward and $\eta = \hat{r}(\cdot)$ as the reward model, and then use a square loss $\ell(a, O, \eta) = (R - \hat{r}(a))^2$. The unbiased feedback assumption (Assumption 3) is equivalent to using a proper loss function for reward fitting in this case.

Our set of assumptions is not the only path towards a rigorous understanding of LLF problems, and it remains an open question if weaker or other forms of assumptions could be made instead. In the current framework, we can easily relax the verifier assumption to use a $\Delta$-approximately correct verifier to model mistakes LLM can make (modify assumption 3 to have $\mathbb{E}_{O \sim f_\eta(a)}[\ell(a, O, \eta)] \leq \ell_\eta^{\min}(a) + \Delta$), and this will induce a linear bias of $O(\sqrt{\Delta})$ term in the final regret.

**Q:** What is the main technical contributions made by this work?

**A (Framework construction as a technical contribution):** Beyond proving theorems, our primary contribution is the rigorous framework for LLF, together with principled assumptions under which such problems are tractable. We view this foundational structure as critical for enabling future advances in the field.

We would like to emphasize that technical results are not just limited to theorems and lemmas, as those regarding the regret bound you have pointed out. We believe that technical papers include two types of technical contributions: those that define the right problem to set the stage, and those that attempt to solve pre-defined problems.

In fact, in many foundational papers, the primary novelty lies in identifying and formalizing the right abstraction, one that captures the essential difficulty of a new class of problems and allows for rigorous analysis. Once such a framework is set, some theory in the traditional sense (upper bounds, sanity checks in carefully presented examples) shows that it is well-posed, but the heavy lifting is really in the conceptualization. Our paper can be cast in this category. LLF problems have been studied largely empirically, and we contribute a formal problem definition, a complexity measure (TED) to capture learnability, and an efficient algorithm to show the utility of the framework. We believe that this kind of framework with inspired algorithm work constitutes a solid technical contribution, even if the theorem-proving part does not contain particularly extensive results.

**Q:** How does this work compare to other frameworks on learning beyond rewards?

**A (Comparison to existing frameworks):** We provide detailed discussion (Appendix C) of how LLF extends frameworks such as IGL. While IGL emphasizes decodability of realized rewards, LLF leverages richer textual information beyond realized rewards to accelerate learning.

**Q:** Why doesn't the obtained regret rate in Theorem 1 match the usual $\widetilde{O}\sqrt{T}$ rate in the bandit literature?

**A (Regret rate):** Achieving the optimal rate requires assuming favorable loss structures (e.g., squared loss with sub-Gaussian noise). In Theorem 4, we show that when additional structural assumptions are made, we can recover the $\widetilde{O}\sqrt{T}$ regret rate.

