# OpenReview forum: "Provably Learning from Language Feedback"
_ICLR.cc/2026/Conference — Submitted to ICLR 2026_

### Official Review · Reviewer_gXYb · 2025-10-14

**Soundness:** 3
**Presentation:** 4
**Contribution:** 1
**Rating:** 2
**Confidence:** 4

**Summary:**

The paper introduces a theoretical framework for Learning from Language Feedback (LLF), defining notions such as verifier, transfer eluder dimension, and a provable no-regret algorithm called HELiX. The authors also provide empirical tests and claim provable efficiency.

**Strengths:**

The paper attempts to provide a formal bridge between language-based feedback and provable learning theory, which is an important direction.

The introduction of a “transfer eluder dimension” to quantify information in feedback is conceptually interesting.

HELiX’s idea of hypothesis elimination and verifier-guided exploration is an appealing concept.

**Weaknesses:**

1. **Verifier Assumptions (Assumption 2) and Unbiased Feedback (Assumption 3)**

The central assumptions—**the Verifier assumption (Assumption 2)** and **the Unbiased Feedback assumption (Assumption 3)**—are **neither empirically testable nor adequately justified**.

The authors assume access to a verifier loss $\ell$ that perfectly reflects *semantic alignment* between hypotheses and feedback, but no practical method is given for constructing or validating such a verifier. While Appendix G argues that “LLMs are strong verifiers,” this is anecdotal; the authors should provide quantitative evidence (e.g., correlation between LLM-verifier loss and human-judged semantic consistency) to substantiate the claim that current LLMs satisfy the assumptions -- at least for the experiments that author did.

Moreover, the claim that a verifier “can be implemented by prompting an LLM” is **untested**. There is no verification that the resulting LLM-verifier satisfies the required *bounded*, *eluder-finite*, or *unbiased* properties. At minimum, the authors should (i) describe how one could empirically estimate the **transfer eluder dimension** of an LLM-based verifier, and (ii) discuss whether this value is finite. If the transfer eluder dimension is effectively infinite, the regret guarantees become vacuous.

So I would recommend authors propose a constructive diagnostic for testing them (e.g., estimating transfer eluder dimension, or calibrating unbiasedness empirically).

---

2. **Lack of Theoretical Validation**

The main regret bound (**Theorem 1**) provides only an upper bound
$\tilde{O}(T^{3/4})$
under minimal assumptions on $\ell$.

* No **lower-bound analysis** is provided, making the claim that the rate is “optimal under $\ell_2$” might be misleading. Without citing or deriving lower-bound results for LLF or for transfer eluder dimension, it is unclear whether $\sqrt{T}$ is in fact optimal under $\ell_2$ assumption.
* The theoretical section seems like largely re-casts the standard eluder-dimension analysis in a new notation, though the introduction of the *transfer eluder dimension* is conceptually meaningful.

It would be accurate to report the $\tilde{O}(T^{3/4})$ bound as a conservative guarantee, but labeling it “optimal” under L₂ is not justified without matching lower bounds. I am fine with this is not a lower bound matching algorithm as it is meaning to provide a new concept. However, I think claiming lower bound on $\ell_2$ should be justified, or at least provide some references.

---

3. **Disconnect Between Theory and Empirics**

The practical **HELiX (Algorithm 2)** deviates substantially from the theoretical Algorithm 1:

* The **verifier** is replaced by heuristic LLM-scoring functions rather than a loss $\ell$ satisfying Assumptions 2–3.
* **Hypothesis elimination** is implemented via “thought sampling + cross-verify” which lacks a formal connection to the theoretical framework.
* **No regret or transfer-eluder metrics** are reported—the figures show only cumulative reward curves (Fig. 2 & 6).

Consequently, the experiments **do not test the theoretical predictions**. It is unclear whether the improvements observed arise from the principles of Algorithm 1 (e.g., UCB-style exploration) or simply from LLM prompting heuristics

**Recommendation:** Report empirical **regret curves** relative to an oracle or to the best-in-hindsight policy, and evaluate whether empirical performance scales with proxy measures of the *transfer eluder dimension* (e.g., hypothesis diversity or verifier variance).

---

4. **Ambiguous Practicality and Computational Cost**

The paper lacks analysis of the **computational feasibility** of HELiX:

* There is no guidance on how the verifier loss $\ell$ or transfer eluder dimension could be computed or approximated in realistic LLM settings.
* Maintaining the hypothesis set is potentially exponential in |A|, yet the paper does not analyze resource or time complexity.
* The “thought-sampling” and “cross-verify” steps appear extremely compute-intensive—requiring quadratic LLM evaluations per iteration—but the paper omits runtime, model size, or token-budget statistics.

---

5. **Weak Experimental Evidence**

* The environments are **toy-scale** and synthetic (Wordle, Battleship, Minesweeper), providing limited insight into general LLF performance.
* There are **no statistical tests**, **no ablations** on verifier quality, hypothesis size N, or the influence of ref.
* Baselines (“No exploitation step”, “No $\pi_{ref}$”) are underspecified and not clearly motivated by theory.
* Reported metrics (cumulative reward) lack any linkage to the proposed theoretical quantities (regret, eluder dimension, unbiasedness).

Overall, the experiments are insufficient to substantiate the claimed theoretical contributions. The authors should include:

1. Ablations varying verifier fidelity and hypothesis diversity.
2. Explicit regret estimation against a known ground truth.
3. Runtime and scaling analysis.

**Questions:**

Mostly written in weaknesses section.

My expertise lies between the theoretical and empirical aspects of this work, so there may be minor misunderstandings. I am open to revising my evaluation based on clarifications.

---

> ### Comment · Reviewer_xnbi · 2025-11-14
> **Language feedback on this review**
>
> Just quickly commenting in to say that this is an excellent review. It articulates very well many of my thoughts when reading the paper and the points I was getting at in my review. And to the authors, I'm still very open to and scientifically interested in discussing our points throughout the rest of the review period

---

> ### Author Response · Authors · 2025-11-26
>
> We appreciate the thorough review and constructive feedback.
>
> > The authors assume access to a verifier loss that perfectly reflects semantic alignment between hypotheses and feedback...no verification that the resulting LLM-verifier satisfies the required bounded, eluder-finite, or unbiased properties.
>
> We first hope to clarify that we do not assume the verifier to be an exact measure of consistency. For instance, Assumption 3 applies to the expectation of $\ell$ under a distribution of feedback, hence noisy verifiers are allowed. In the current framework, we can easily relax the verifier assumption to use a $\Delta$-approximately correct verifier to model mistakes LLM can make (modify Assumption 3 to have $E_{O\sim f_\eta(a)}[\ell(a,O,\eta)] \le \ell_\eta^{\min}(a) + \Delta$), and this will induce a linear bias of $O(\sqrt{\Delta})$ term in the final regret. To your second point, the verifier is a modeling assumption we choose to make for the LLF framework, but it is not a required component for an algorithm. In particular, HELiX does not need to construct a verifier. The assumptions are also only required for the theoretical guarantees to hold.
>
> > At minimum, the authors should (i) describe how one could empirically estimate the transfer eluder dimension of an LLM-based verifier, and (ii) discuss whether this value is finite. If the transfer eluder dimension is effectively infinite, the regret guarantees become vacuous.
>
> We hope to clarify some potential misunderstanding. Transfer eluder dimension is not a property of a particular LLM, but rather characterizes the relationship between the hypothesis space $H$, the verifier loss $\ell$, and the feedback and reward mapping. This complexity notion is an analysis tool, but does not need to be estimated by an algorithm empirically. In particular, there is no need to estimate transfer eluder dimension for HELiX to run. To your second point, our regret bound holds regardless of whether the transfer eluder dimension is finite. While it is true that the bound would become vacuous if the dimension is infinite, this exactly aligns with our intuition: when the dimension is infinite, the corresponding LLF problem is difficult for efficient learning. We have discussed in Examples 1 and 2 the value of transfer eluder dimension (when it is finite) under different forms of feedback, and mentioned when the value would be infinite (Section 4.2).
>
> (Also, to avoid confusion: transfer eluder dimension is a property of $(H, \ell)$ and the reward mapping; not of a particular LLM).
>
> > The main regret bound (Theorem 1) provides only an upper bound under minimal assumptions on \ell…though the introduction of the transfer eluder dimension is conceptually meaningful.
>
> In the writing, by no means do we claim $\sqrt{T}$ is optimal to LLF with $\ell_2$. We referred to it as the rate for the “classical optimal rates for bandit learning with direct reward feedback” [4]. We are sorry for the confusion and have updated the wording. Regarding your second point, the eluder dimension analysis relies on learning the true reward from revealed rewards, whereas our setting completely forgoes observable reward signals. Though there may be some resemblance of proof structure and techniques, our analysis is novel and cannot be re-cast directly from the standard eluder dimension analysis.
>
> [4] Tor Lattimore, Csaba Szepesvari. “Bandit Algorithms”. Cambridge University Press, July 2020.

---

> > ### Author Response · Authors · 2025-11-26
> >
> > > The verifier is replaced by heuristic LLM-scoring functions rather than a loss  satisfying Assumptions 2–3…No regret or transfer-eluder metrics are reported—the figures show only cumulative reward curves (Fig. 2 & 6).
> >
> > We hope to clarify that the LLM scoring function is not the verifier (but rather, a reward function) nor a replacement of it, and we do not explicitly construct a verifier in our experiments. We rely on trained SOTA LLMs to propose the most-likely $K$ hypotheses according to the observations so far. By doing this, we leverage LLM’s capability to produce hypotheses that minimize $\ell(a,o,\eta)$ without explicitly computing this value. This also allows our algorithm to scale with future generations of LLMs that will have increasing abilities to produce loss-minimizing hypotheses.
> >
> > While we never explicitly implement or query $\ell$ in our practical implementation of HELiX (Algorithm 2), we use the information from $\ell$ implicitly through hypothesis updates. Directly checking for Assumption 3 would require access to $\ell(\cdot,\cdot,h)$ for all $h$, which is infeasible in practice.
> >
> > Instead, to give a sense of how Assumption 3 connects to our experiments, we evaluate a key implication of it: that the shrinking hypothesis set remains consistent with the true hypothesis along the trajectory. Concretely, if Assumption 3 approximately holds, then as the algorithm updates and prunes the hypothesis set, the true hypothesis should almost always remain included. To test this, we manually annotated 10 trajectories from the Battleship experiment. For each step in each trajectory, we checked whether the candidate set of hypotheses produced by HELiX contained the true hypothesis. The following table reports (i) the average fraction of steps within a trajectory for which the true hypothesis is included (“Assumption 3 satisfaction (trajectory-averaged)”), and (ii) the fraction of all annotated steps across trajectories for which the true hypothesis is included (“Assumption 3 satisfaction (step-averaged)”). Both metrics are above 95%, empirically supporting that Assumption 3 holds to a good approximation in our experiment.
> >
> > | Metric                                          |  Value |
> > | ----------------------------------------------- | -----: |
> > | Assumption 3 satisfaction (trajectory-averaged) | 96.26% |
> > | Assumption 3 satisfaction (step-averaged)       | 95.98% |
> >
> > We appreciate your suggestion to make the connection between the theoretical and practical versions of HELiX clearer. Below, we provide a line-by-line walkthrough comparing the two versions:
> >
> > **Line 5 of Algorithm 1 → line 5 of Algorithm 2:** Algorithm 1 maintains a hypothesis space $\mathcal{H}_t$ at iteration $t$, which contains all hypotheses $\eta$ that are consistent with observed feedback. Then Algorithm 1 searches over all possible policies by computing $\pi_p$ and $\pi_o$. Since it is impractical to keep a large hypothesis set and do an exhaustive search, we approximate this step in Algorithm 2 with sampling finite sets of candidates $\hat{\mathcal{H}}_t$ and $\hat{\mathcal{A}}_t$, respectively.
> >
> > **Line 6 of Algorithm 2:** This step is purely motivated by observations that using a random reference policy to generate random actions provides empirical benefits.
> >
> > **Lines 6–8 of Algorithm 1 → lines 7–11 of Algorithm 2:** Algorithm 1 carries out a minimax step to determine whether to stop exploring, when a policy is simultaneously optimal for all remaining hypotheses. To approximate this step, we use a thought cross-verify step to determine if some action $a$ is optimal across all sample hypotheses.
> >
> > **Lines 9–11 of Algorithm 1 → lines 12–14 of Algorithm 2:** Algorithm 1 computes a UCB-style exploration policy via optimism. This is approximated by eliminating hypotheses whose highest score is lower than those of other hypotheses in thought cross-verify.
> >
> > Regarding regret and cumulative reward, we hope to clarify some potential misunderstanding. In the three environments, the optimal reward baseline is deterministic given a hypothesis, so a regret plot can be directly mapped from a cumulative reward curve by subtracting the latter from the cumulative reward of an optimal policy.

---

> > > ### Author Response · Authors · 2025-11-26
> > >
> > > > There is no guidance on how the verifier loss or transfer eluder dimension could be computed or approximated in realistic LLM settings…the paper omits runtime, model size, or token-budget statistics.
> > >
> > > Thank you for your suggestion and going through our algorithm. As addressed in response to your first bullet point, HELiX does not need to construct a verifier or estimate the transfer eluder dimension to operate. You are correct that it would be infeasible to maintain a hypothesis set in practice, so our practical version of HELiX does not need to maintain a large hypothesis set but utilizes thought-sampling to approximate line 5 of Algorithm 1.
> > >
> > > HELiX (Algorithm 2) is a practical implementation of our theoretical algorithm, but by no means is it the only possible implementation. You are correct at pointing out that HELiX incurs a computational cost on sampling that scales with the $K$ number of top hypotheses that we propose at each turn, and if we only look at the algorithm, building the score matrix through “cross-verify” does incur a $K^2$ LLM calls. However, this is not as costly as it might seem due to several reasons:
> > >
> > > 1. **Better algorithms can lead to solving the game faster (i.e., early termination):** In all the games, we allow agents to act many more steps than necessary to solve the game. A better algorithm can often lead to quicker terminations. We verify this with the token count of our algorithm.
> > >
> > > 2. **Parallelization of thought sampling and verification:** In practical implementations, we can also sample thoughts and verify thoughts in parallel, reducing $O(K^2)$ to $O(K)$ during thought-cross-verify and $O(K)$ to $O(1)$ during thought sampling.
> > >
> > > 3. **Efficient use of tokens through prefix caching:** In practical implementations, we could leverage advanced inference techniques like prefix caching where instructions sharing the same prefix sequence can be stored and loaded as KV cache without re-computing. In HELiX, many LLM calls share common observations and judgments, significantly reducing the actual tokens needed.
> > >
> > > Since we mainly use the experiment to illustrate and instantiate one practical implementation, we did not implement prefix caching or parallel sampling. The table below mostly demonstrates that even without such advanced techniques, just by exploring the environment better, we still avoided exponential cost blowup (instead of 9x token to the baseline, we are 3.73x to 4.02x the token count of baseline).
> > >
> > > **Battleship**
> > >
> > > | Method                                                | Token Count | Comparison               |
> > > | ----------------------------------------------------- | ----------: | ------------------------ |
> > > | Baseline                                              |     698,873 | 1x                       |
> > > | HELiX (no exploitation step) (explore optimistically) |   3,151,173 | 4.51x ($K=3$ hypotheses) |
> > > | HELiX                                                 |   2,812,856 | **4.02x** ($K=3$ hypotheses) |
> > >
> > > **Minesweeper**
> > >
> > > | Method                                                | Token Count | Comparison               |
> > > | ----------------------------------------------------- | ----------: | ------------------------ |
> > > | Baseline                                              |     553,389 | 1x                       |
> > > | HELiX (no exploitation step) (explore optimistically) |   2,538,838 | 4.59x ($K=3$ hypotheses) |
> > > | HELiX                                                 |   2,064,156 | **3.73x** ($K=3$ hypotheses) |
> > >
> > > > There is no guidance on how the verifier loss or transfer eluder dimension could be computed or approximated in realistic LLM settings. Maintaining the hypothesis set is potentially exponential in |A|, yet the paper does not analyze resource or time complexity.
> > >
> > > We hope to clarify some potential misunderstanding here. The transfer eluder dimension is only used in theoretical analysis and not required for the HELiX algorithm to run. Furthermore, practical HELiX (Algorithm 2) does not maintain a large hypothesis set. The concern about resource requirements applies to a literal implementation of Algorithm 1, but Algorithm 2 avoids maintaining a set whose size is exponential in $|A|$.

---

> > > > ### Author Response · Authors · 2025-11-26
> > > >
> > > > > The environments are toy-scale and synthetic...Baselines (“No exploitation step”, “No \pi_{ref}”) are underspecified and not clearly motivated by theory.
> > > >
> > > > We mainly use the experiment to illustrate and instantiate one practical implementation of HELiX. The focus of our paper is on offering a theoretical framework for people to mathematically analyze the paradigm of learning from language feedback and what formal quantities are needed to make an algorithm “provably” learning. This goes beyond a single application task (or even a collection of tasks) where you see the numbers go up and up. A concrete implementation/instantiation of our algorithm can be task-specific, but the general learning principles, which we have explored in this paper, are universal across applications.
> > > >
> > > > We picked the three tasks for demonstration purposes of how an empirical instantiation of our theoretical algorithm would look like. All three tasks focus on exploration, presenting incomplete information of the game state and by carefully choosing actions, information is revealed:
> > > >
> > > > * **Battleship**: hidden displays of enemy ship locations
> > > > * **Minesweeper**: hidden locations of the mine
> > > > * **Wordle**: hidden word
> > > >
> > > > They are fairly representative of a broad class of partial / hidden information tasks that we want to investigate, where smarter actions chosen by the agent might not incur high immediate reward but will help gain higher long-term reward.
> > > >
> > > > Thank you for pointing out the lack of reference to $\pi_{\text{ref}}$ in the main text. We will add a pointer to the detailed $\pi_{\text{ref}}$ ablation and discussion in Appendix E.
> > > >
> > > > We hope that our responses have fully addressed your concerns and you will consider re-evaluating the merits of this paper.

---

> ### Comment · Reviewer_gXYb · 2025-11-26
>
> ## a) Verifier-related comments
>
> ### (i) Unbiasedness & empirical validation
>
> My original point was not that the verifier must be perfect.  Rather, Assumption 3 explicitly requires the verifier to be *unbiased* with respect to the true hypothesis. In authors experiments, the LLM effectively plays the role of this verifier. Therefore, the assumption becomes an empirical question: **how unbiased/noisy is the LLM when used as a verifier?**
>
> Authors' newly added table showing “Assumption 3 satisfaction” in Battleship is helpful and partially resolves this concern. I strongly suggest including that table in the camera-ready version, as it is essential evidence that the theoretical assumptions have empirical grounding.
>
> To clarify:
> - HELiX does *not* need to explicitly compute ℓ,
> - but the *theory* assumes a verifier satisfying Assumption 3,
> - and since the *experiments implicitly -- I think explicitly -- rely on the LLM as this verifier*, some quantification of verifier accuracy/noise is necessary.
>
> ### (ii) Relationship between ℓ and the LLM
>
> Because the paper uses the LLM to score hypotheses/actions, the induced ℓ is *implicitly* determined by the LLM’s internal behavior. Even if ℓ is not explicitly computed, it is still LLM-dependent. I agree that we cannot expect a theoretical bound to hold exactly for an untrained verifier. But this makes it more important to quantify the LLM’s approximation error $\Delta$ in practice, since $\Delta$ directly affects whether the regret guarantee is meaningful.
>
> ---
>
> ## b) Optimality of the regret bound
>
> Thank you for clarifying that you plan to update the writing. However, it is still not fully clear what specific revision will be made. The current draft states:
>
> > “the bound can be tightened to match the optimal order $\sqrt{T}$.”
>
> But $\sqrt{T}$ is not the optimal rate (and used "match" terminology in paper) in the general LLF setting—only in the squared-loss case. To be clear, this is totally fine -- at least I am not interested in the optimality of the analysis --  but the terminology “optimal” should be used only when formally appropriate.
>
> Regarding transfer eluder dimension:
> I remain mixed. It is standard in RL theory, yet authors do provide some efforts for the LLF setting, which I really appreciate. I am just saying it looks like a bit standard.
>
> ---
>
> ## c) Clarification on “LLM scoring vs verifier”
>
> I am confused by the following statement in the rebuttal:
>
> > “The LLM scoring function is not the verifier (but rather, a reward function) nor a replacement of it.”
>
> But in the paper, authors wrote:
>
> > “Rate all the actions individually based on whether the action is aligned with the hypothesis.”
>
> This is, functionally, **a verifier operation**—evaluating action–feedback consistency given a hypothesis.
>
> Additionally, the authors mention:
>
> > “We rely on trained SOTA LLMs to propose the most-likely K hypotheses according to the observations…”
>
> This again seems to use the LLM to approximate the minimization of $\ell(a, o, \eta)$, which is *exactly* a verifier’s role.
>
> So my confusion remains:
> - If the LLM is used to judge hypothesis–feedback consistency, then it *is* acting as a verifier.
> - If the LLM is not the verifier, then what object corresponds to ℓ in the experiments?
> - If $\ell$ is not explicitly computed, but implicitly induced by the LLM, shouldn’t the transfer eluder dimension depend on the LLM?
>
> Some clarification in the main text would be extremely helpful. Right now, the relationship among LLM → $\ell$ → transfer eluder dimension is unclear.
>
> ---
>
> ## d) Regret vs cumulative reward
>
> The authors wrote:
>
> > “the optimal reward baseline is deterministic… so regret can be computed from cumulative reward.”
>
> Yes, that is true.
> However, as a reviewer, I care about *sublinear behavior* and whether the *empirical regret curve matches the theoretical T^{3/4}-type scaling*. Cumulative reward plots do not convey this.
>
> Therefore:
> - **Please plot regret curves explicitly.**
> They are trivial to compute and directly test theoretical predictions.

---

> ### Comment · Reviewer_gXYb · 2025-11-26
>
> ## e) ELiX, verifier construction, and transfer eluder dimension
>
> You responded:
>
> > “HELiX does not need to construct a verifier or estimate the transfer eluder dimension.”
>
> This is not what I was arguing. I agree HELiX does not *need* either of these to run.
>
> My concern is different:
> - If HELiX uses the LLM as a verifier,
> - and $\ell$ is implicitly derived from the LLM,
> - then, for scientific completeness, the paper should discuss what the transfer eluder dimension looks like for an LLM-based verifier.
>
> Right now, there is an internal disagreement:
> 1. **The theory requires a verifier with finite transfer eluder dimension.**
> 2. **The experiments use an LLM as the verifier.**
> 3. **The authors say HELiX does not construct a verifier.**
> 4. **But HELiX appears to rely on LLM outputs that functionally *are* verifier decisions.**
>
> This mismatch should be addressed clearly in the paper.
>
> ---
>
> ## f) Choice of environments
>
> You wrote that the three tasks are “representative of a broad class of partial / hidden information tasks.”
>
> I am not fully convinced:
> - These tasks are not widely used as LLF benchmarks.
> - Tasks such as *Training a Generally Curious Agent* (from LLF-Bench) have limited adoption and only ~6 citations.
> - It is also unclear why LLMs are needed for Wordle, Minesweeper, or Battleship—these tasks are not typically solved using LLM-based verification.
> - Maybe a dumb question -- why can't author use some agentic task because they requires a) language feedback and b) many exploration, such as Software engineering or finance?
>
> I am not saying the tasks are invalid, but I do believe the paper would be stronger with:
> - either more standard LLF tasks,
> - or a clearer justification for why LLMs should be used for these domains at all.
>
> ---
>
> The remaining core confusion is:
>
> **How exactly does the LLM relate to the verifier loss ℓ and the transfer eluder dimension, given that HELiX uses the LLM to generate and score hypotheses?**
>
> The rebuttal suggests that:
> - HELiX doesn’t need a verifier,
> - doesn’t compute ℓ,
> - doesn’t estimate eluder dimension,
> - and the LLM isn’t the verifier.
>
> But the algorithm *appears* to use the LLM for precisely the role the theory calls “verifier.”
>
> Some clarification of this conceptual relationship is needed for the paper to be coherent. I think I might have misunderstanding in this connections so would you please explain this for me -- sorry for missing some link between verifier/LLM and HELiX.
>
> ---
>
> Regarding computational cost, I think the authors’ current response misses the scientific point. The question is not whether HELiX is “too costly to be practical” — rather, for scientific clarity, the paper should **explicitly quantify and report the computational cost** of the practical HELiX implementation. Even if the cost is higher, that is perfectly acceptable if it corresponds to significantly better performance. What matters is that readers can understand the trade-off. I appreciate the token-count tables provided in the rebuttal; please include these (or an expanded version) in the camera-ready paper so that future researchers can rely on them when evaluating the practicality of HELiX or building on your implementation.

---

> ### Author Response · Authors · 2025-12-03
>
> We appreciate your fast response. We are glad that you have found our added tables helpful. We will add both of them to the paper!
> > Regarding computational cost, I think the authors’ current response misses the scientific point. The question is not whether HELiX is “too costly to be practical” — rather, for scientific clarity, the paper should explicitly quantify and report the computational cost of the practical HELiX implementation. Even if the cost is higher, that is perfectly acceptable if it corresponds to significantly better performance. What matters is that readers can understand the trade-off.
>
> To help give a sense of the computational overhead of our practical implementation of HELiX (Algorithm 2), we expand on some key problem-dependent parameters and hyperparameters. For detailed prompt format and reasoning traces, please see Appendix F.4. Given an LLF problem, practical HELiX needs an input of `domain_description`, `action_space`, and `learning_task_instruction` at the start of learning. This constitutes the initial prompt. At each round, given historical context $C$, HELiX goes through 2-3 LLM calls, whose contexts are outlined below:
> 1. Feedback-consistent hypothesis generation: the context includes $C$ along with the most recent `domain_state` and instruction to simultaneously propose `num_actions` diverse hypotheses and actions. This is done in one LLM call.
> 2. Thought cross-verify: the context includes $C$ along with the `num_actions` proposed hypotheses and actions from step 1 and instruction to evaluate each action under each hypothesis. This is done in one LLM call.
> 3. Exploration step: if $\pi_{\text{ref}}$ is enabled, the LLM is asked to propose additional `num_ref_actions` exploratory actions. The context includes $C$ along with the `num_actions` proposed actions from step 2 and instructions to propose actions different from those.
> The remaining steps do not need LLM calls; these include: consensus check, UCB elimination based on scores from step 2, and advantage-based exploration with tie-breaking. Tunable hyperparameters include `num_actions`, whether to use $\pi_{\text{ref}}$, and`num_ref_actions`.
>
> > How exactly does the LLM relate to the verifier loss ℓ and the transfer eluder dimension, given that HELiX uses the LLM to generate and score hypotheses?
>
> In practical HELiX (Algorithm 2, line 5), an LLM is instructed to output a set of diverse, feedback-consistent hypotheses. This is intended as an approximation to the feedback-consistent hypothesis set in line 5 of Algorithm 1. Importantly, this feedback-consistent hypothesis generation process does not directly implement the verifier loss $\ell$: the LLM is not asked to evaluate individual (hypothesis, feedback) pairs or assign them a scalar loss $\ell$. The only place where the LLM outputs a numerical “score” is in the cross-verify step, where it is asked to rate each action $a$ under each generated hypothesis $\eta$. This “score” is interpreted as the reward $r_\eta(a)$, not the verifier loss $\ell(a, o, \eta)$. One can reasonably view the LLM as implicitly implementing some underlying verifier when it is constrained to generate feedback-consistent hypotheses, but Algorithm 2 neither requires nor uses an LLM as a pure, explicit verifier. Rather than approximating $\ell$ itself, the practical approximation occurs at the level of the hypothesis sets that the LLM produces.
>
> Consequently, in our experiments, the transfer eluder dimension is induced implicitly by the LLM-generated hypothesis sets: it is the length of the longest sequence of feedback observations for which the remaining LLM-generated hypotheses still has an $\epsilon$-sized reward gap. Yes, this quantity is LLM-dependent, just as the theoretical transfer eluder dimension would be dependent on $\ell$ if $\ell$ were explicitly constructed by an LLM. As in most work using eluder-type dimensions, we do not attempt to enumerate all possible feedback sequences (which would be exponentially many) and hence do not compute this quantity empirically.

---

> > ### Author Response · Authors · 2025-12-03
> >
> > > But this makes it more important to quantify the LLM’s approximation error \Delta in practice, since \Delta directly affects whether the regret guarantee is meaningful.
> >
> > Following our explanation above, Algorithm 2 does not use the LLM to directly implement the verifier loss $\ell$. Instead, the LLM is used to approximate the remaining hypothesis set. Thus, we do not attempt to estimate an empirical approximation error $\Delta$ for $\ell$ in our experiments. Rather, appears in the theory as a relaxation of the unbiased-verifier assumption: if we assume a $\Delta$-approximately correct verifier to model potential LLM mistakes (formalized by modifying Assumption 3 to $E_{O\sim f_\eta(a)}[\ell(a,O,\eta)] \le \ell_\eta^{\min}(a) + \Delta$), then the regret bound acquires an additional linear bias term of order $O(\sqrt{\Delta})$. In other words, $\Delta$ controls how much the regret guarantee degrades under verifier error, but it is a theoretical robustness parameter rather than a quantity we can directly measure for the LLM in our current experimental setup.
> >
> > > Thank you for clarifying that you plan to update the writing. However, it is still not fully clear what specific revision will be made.
> >
> > We have updated the paper with the modified phrasing:
> >
> > While the order $\widetilde{O}(T^{3/4})$ on the time horizon $T$ may appear suboptimal compared to classical $\widetilde{O}(\sqrt{T})$ optimal rates for bandit learning with direct reward feedback, this slower rate is in fact a principled consequence of our minimal assumptions. Specifically, our analysis makes no structural assumptions on the verifier loss $\ell$ beyond boundedness.  If we have more structural knowledge of $\ell$, say, that it is a squared loss, then the bound can be tightened to match the optimal order $\widetilde{O}(\sqrt{T})$ {\color{red}in classical bandit learning} (see Theorem 4 in Appendix B.4).
> >
> > > Please plot regret curves explicitly.
> >
> > We will add regret curves in Appendix E.
> >
> > > Regarding transfer eluder dimension: I remain mixed. It is standard in RL theory, yet authors do provide some efforts for the LLF setting, which I really appreciate. I am just saying it looks like a bit standard.
> >
> > Our theoretical analysis is based on eluder-dimension-based analyses used for optimism-under-uncertainty-based exploration methods. If the reviewer is looking for a new paradigm for analysing such methods, that is not one of our contributions. What our paper does contribute is the following: formalizing the LLF problem mathematically, and defining all LLF components in the eluder dimension framework including a new version of eluder dimension. Note that defining the transfer eluder dimension is nontrivial as it requires carefully relating the key concepts of our formalization to the key steps in the eluder dimension based analysis. These include hypothesis, verifier, hypothesis-induced reward function and reward-"informativeness".
> >
> > > I am not saying the tasks are invalid, but I do believe the paper would be stronger with: either more standard LLF tasks, or a clearer justification for why LLMs should be used for these domains at all.
> >
> > We are actively working on the experiments and will add additional tasks from LLF-Bench and a theorem proving task to the paper once the experiments finish running.

---

### Official Review · Reviewer_afE5 · 2025-10-28

**Soundness:** 4
**Presentation:** 3
**Contribution:** 3
**Rating:** 6
**Confidence:** 3

**Summary:**

The paper "Provably Learning from Language Feedback" proposes a theoretical analysis of in-context learning with LLMs for stateless RL tasks. To do so, authors introduce the transfer eluder dimension, which quantifies the hardness of the Learning from Language Feedback at hand. This theoretical quantity, based on the eluder dimension, allows to show that learning from feedbacks is at least as easy as from associated feedbacks, whenever the feedback is discriminative (i.e., is able to distinguish true hypotheses from false ones based on feedbacks to the same extent than based on rewards). Finally, the paper gives an illustrative algorithm that iteratively excludes hypotheses (and their associated actions) from the pool of candidates, in a similar way as UCB approaches, but based on language rewards rather than on observed rewards. Under some assumptions regarding the ability of a verifier agent to assess the consistency of the feebacks provided in response to a selected action for a given hypothesis, authors show that the regret of their algorithm is sublinear. Some experiments empirically complement this analysis.

**Strengths:**

- Important work to better understand in-context learning of LLMs in one step RL problems. This is a continuously growing field, with many works proposing to use LLM agents in a loop to achieve complex tasks, often greatly lacking of formalization. This theoretical work looks to propose a significant step to close that gap.

- Theoretical assumptions and demonstrations look reasonnable and well sounded (I did not check all the proofs though)

- Authors made an important effort for illustrating abstract concepts with examples

**Weaknesses:**

-  Pedagogy and Positioning : While it is quite well written globally, I had a bit of difficulty fully grasping this paper. First, I believe the positioning should make it clearer that the work is set in a one-step RL (bandit) framework. It took me some time to realize that the paper does not consider the more common multi-step RL setting that I am more familiar with. I appreciate the authors’ effort to provide illustrative examples for the various abstract concepts — these are genuinely helpful. However, I think the discussions that justify the different formulations could be expanded and made more pedagogical. I found it rather challenging to interpret several elements of the paper. For instance, the conclusion of Example 1 could be more detailed to clarify the main takeaway. Similarly, the definitions are quite dense, and understanding their precise meaning requires effort. A stronger pedagogical focus in these sections would be highly appreciated.

- Extension to the non deterministic setting: While authors claim their framework is directly applicable for stochastic environments, I feel it is not fully trivial

- Extension to the multi-turn return setting : Authors provide insights for the extension to a contextual setting (which should better further developped by the way). But nothing is said about the classical multi-turn return setting of the RL litterature (beyond bandits). Would it be possible to extend the work for cumulative discounted rewards / feedbacks ? This looks crucial for many complex tasks, which cannot explore efficiently if only taking utterances globally.

- Experiments lack baselines and explanations. At least, I would have expected a classical UCB or Thompson Sampling baseline based rewards. Or even an epsilon-greedy exploration scheme (with or without a LLM as the agent). Many other agentic approaches could have also been experimented. For instance approaches based on TextGrad, which proposes backward steps in the form of a textual feedback.  Some ablations are not well described either. For instance, what $\pi_{ref}$ stands for ?  Experiments could have also been performed on more challenging environments, such as theorem proving for instance.

**Questions:**

1. Algorithm Helix assumes the availability of the optimal policy for any hypothesis from the pool. For simple settings where hypothesis are simply defined as an action (such as in the multiarm bandit setting), this is trivial. But I suspect that many other settings would hinder the applicability of the proposed algorithm. Please discuss.

2. How would the approach compare to baselines mentionned in weaknesses ?

3. How confidence levels can be set in Helix ? Is there a decreasing scheme as in epsilon greedy approaches ? Note that there exists a scheduling decrease that allow epsilon-greedy to be optimal (while unavaible in practical settings).

4. Could author better precise what they mean by "rubric". I feel this is quite unclear, as examples given in discussion are quite complex, while what is considered for experiments is extremly simple.

5. what $\pi_{ref}$ stands for in the abblations ?

6. How would perform the proposed algorithm in more complex environements (e.g., contextual / stochastic / implying reasonning) ?

7. Would it be possible to extend the work to the multi turn RL setting (implying a sequence of interactions with an environment) ?

8. Is the reward mapping availability a common practical assumption ?




Minor remarks:

- p7. "In addition to this example, one can check that... " ==> please mention that this can be found in appendix C4
- top of p7 : fist ==> first

---

> ### Author Response · Authors · 2025-11-26
>
> Thank you for the positive review and the detailed suggestions on pedagogy and baselines.
>
> > Pedagogy and Positioning…A stronger pedagogical focus in these sections would be highly appreciated.
>
> Thank you for the detailed feedback on the writing! We will explicitly label the setting as one-step RL/bandits in the introduction and Section 3.1, and fix all typos.
>
> > Extension to the non deterministic setting…Extension to the multi-turn return setting…
>
> We already include contextual extension (Remark 1; Appendix D.2). We assumed $r*$ is deterministic for ease of exposition. Extending to stochastic $r*$ only requires an additional step handling the reward noise. As long as the noise is sub-Gaussian, standard concentration analysis can be applied, similar to that in the bandit setting [1].
>
> We completely agree that it would be an exciting future direction to extend our LLF formulation to multi-turn stateful interactions. A naive way to apply our current setup to an RL setting is to regard all possible (finite) histories as contexts. However, the state space in this case would be exponentially large, and graceful regret analysis would be prohibitive. A more careful treatment of the components of our framework, such as history-dependent verifiers, would be necessary to allow for meaningful analysis. While these extensions are outside the scope of this paper, we provide the theoretical ground to build them on. Thanks for your insightful suggestions! We will add a short paragraph about the mentioned extensions to the discussion section.
>
> [1] Sébastien Bubeck, Vianney Perchet, Philippe Rigollet. “Bounded regret in stochastic multi-armed bandits.” COLT 2013.
>
> > Algorithm Helix assumes the availability of the optimal policy for any hypothesis from the pool…Please discuss.
>
> In our tasks, $\pi_\eta$ is simply the greedy policy with respect to $r_\eta$. In richer domains, HELiX can use approximate or LLM-proposed policies per hypothesis. Even if the exact greedy policy $\pi_\eta$ is not available, our analysis can be extended to applying an $\epsilon$-optimal policy instead. This approximation will only incur a small additional regret term controlled by $\epsilon$.
>
> > I would have expected a classical UCB or Thompson Sampling baseline based rewards…Experiments could have also been performed on more challenging environments, such as theorem proving for instance.
>
> The greedy baseline we implemented corresponds to “draw actions from the argmin of cumulative verifier loss” (Section 4.4), which lacks exploration. In principle, one can use LLMs to convert feedback into reward scores and then run exploration techniques like UCB. However, this deviates from any existing UCB or Thompson sampling algorithm that directly learns from reward signals. To convert them to meaningful baselines requires additional design bridging the feedback–reward gap, such as the mechanism used to decode rewards from feedback.
>
> > How confidence levels can be set in Helix? Is there a decreasing scheme as in epsilon greedy approaches? Note that there exists a scheduling decrease that allows epsilon-greedy to be optimal (while unavailable in practical settings).
>
> We used the following schedule for $\theta_t$ in the experiments:
> $$\max\{( \frac{1}{t^2}, \min_{a\in\mathcal{A}}\inf{|r_\eta(a)-r*(a)|: \eta\in\mathcal{H}, \eta\ne \eta*})\}.$$
>
> Theoretically, $\theta_t$ is chosen to be of order $1/t^2$ (see Equation (2)) for the regret bound to hold. We’d like to clarify that even though the $\epsilon$-greedy policy can converge to an optimal policy under a decreasing $\epsilon$ schedule, the regret bound incurred by such an algorithm will not be optimal.
>
> > Could the author better precise what they mean by "rubric".
>
> We use rubric to refer to the criteria implicit in a hypothesis that the verifier uses to judge consistency of feedback with actions (Section 3.3).
>
> > what $\pi_{ref}$ stands for in the ablations?
>
> The $\pi_{\text{ref}}$ ablation is detailed in Appendix E; we will add a pointer in the main text.
>
> > Is the reward mapping availability a common practical assumption?
>
> The availability of a reward mapping is a design choice of our formulation. This assumption is equivalent to having a function class parameterized by hypotheses. Knowing a realizable function class is a standard assumption in learning theory, e.g. in classical bandit settings [2,3]. We briefly discuss an alternative formulation without this assumption in Appendix D.3.
>
> [2] Yasin Abbasi-Yadkori, David Pal, Csaba Szepesvári. “Improved Algorithms for Linear Stochastic Bandits.” NIPS 2011.
>
> [3] Yue Wu, Jiafan He, Quanquan Gu. “Uniform-PAC Guarantees for Model-Based RL with Bounded Eluder Dimension.” UAI 2023.

---

> > ### Comment · Reviewer_afE5 · 2025-11-27
> >
> > Thanks for authors for their answers.
> >
> > I still think that further results with more classical baselines and in other settings (e.g., stochastic, multiturn which goes far beyond simply using the history : credit assignement) would be useful to strenghten the paper
> >
> > But I was already positive for this paper that brings interesting things for the community. I still think the same. I keep my score unchanged.
> >
> > Regards

---

### Official Review · Reviewer_xnbi · 2025-11-01

**Soundness:** 2
**Presentation:** 2
**Contribution:** 2
**Rating:** 2
**Confidence:** 4

**Summary:**

The paper studies how to do learning with language feedback and proposes a no-regret algorithm for the tasks. They review and formalize the LLF setting in section 3, defining the setup for the policy to maximize an unknown reward function given only token-based feedback from the actions. The emphasis here is that distinct from the usual RL/bandit setup, the policy only has information about the reward values through natural language feedback. From here, they also define a hypothesis space and verifier to complete the setup Then, S4.1 proposes to quantify information in the feedback by an extension of the eluder dimension to characterize how well the feedback reduces uncertainty about the unknown reward function, and the rest of that section builds up to their UCB-style algorithm HELiX, furnishing a regret bound in Thm 1. Experimentally they evaluate on wordle, battleship, and minesweeper.

**Strengths:**

Language feedback is a rich additional source of information and under-explored topic, as the language models and tasks have only recently enabled progress in this area. In a research landscape where many language tasks and methods are ill-defined and not possible to make theoretical statements about, I am generally supportive of more formalization in the LLF direction so more precise mathematical statements can be made about improvement and learning in language spaces. The paper has a nice arc building up to HELiX and the resulting regret bound, and the experiments in Figure 2 demonstrate how HELiX is better than a greedy baseline on the tasks.

**Weaknesses:**

Despite being generally positive about formalizing this space, the experimental results are holding me back from giving an acceptance at this point. I am very open to discussing this.

Before getting into the specific formal setting of the paper, Wordle, Battleship, and Minesweeper seem like well-studied tasks where other methods and experimental settings have been investigated. However, the results in Figure 2 are isolated from these, and not compared to what I would consider state-of-the-art AI methods on these settings. Reporting the ablations across cumulative reward is interesting for comparing bandit-like methods, but seems slightly disconnected from actually knowing how well the performance on these settings are. This makes it difficult to understand how significant of an experimental advance the paper provides and leads me to an interpretation that the main contribution of the paper is more in the theoretical contribution instead of being experimentally SOTA. Slightly beyond this, why only select these three tasks, and not evaluate on all of the tasks from LLF-Bench?

This could be okay, but the domain of the paper is an experimental topic, and "provable learning from LF" is a strong title. On this, "provable" still needs some scoping, e.g., only to the setting they formalized.

And lastly, there are many other pieces of related work that use language-based feedback, such as many of the coding agents (SWEAgent, OpenHands, and the citation graph around these), as well as DSPy and TextGrad. All of these settings also iterative improve a textual object based on text-based feedback in many cases. They are not mentioned in the submitted paper. I believe they would minimally be interesting to conceptually connect to. They could be interpreted as using language feedback "without" provable guarantees, so it would be very insightful for these communities to understand how LLF with provable guarantees can help improve them. And it would be an extremely strong paper to compare exactly to some of their exact and previously-published experimental settings. My guess is that it may be difficult to get it exactly in the setting considered in the paper, but maybe they could be minimally be considered as baselines on the tasks in this paper?

**Questions:**

I am very open to discussing my experimental concerns and connections to agents/DSPy/TextGrad throughout the rest of the review period

---

> ### Author Response · Authors · 2025-11-26
>
> We appreciate the encouragement to better connect to prior empirical works using language feedback.
>
> > leads me to an interpretation that the main contribution of the paper is more in the theoretical contribution instead of being experimentally SOTA.
>
> Your interpretation is correct. The focus of our paper is on offering a theoretical framework for people to mathematically analyze the paradigm of learning from language feedback and what formal quantities are needed to make an algorithm “provably” learning. This goes beyond a single application task (or even a collection of tasks) where you see the numbers go up and up. A concrete implementation/instantiation of our algorithm can be task-specific, but the general learning principles, which we have explored in this paper, are universal across applications. To understand this, stochastic gradient descent (SGD) is a general learning algorithm, but for different tasks, you have dataset constructions, learning rate choices, gradient clipping threshold, etc. These empirical “tricks” for each application are orthogonal to the contributions of proposing, analyzing, and understanding SGD.
>
> > Slightly beyond this, why only select these three tasks, and not evaluate on all of the tasks from LLF-Bench?
>
> We picked the three tasks for demonstration purposes of how an empirical instantiation of our theoretical algorithm would look like. All three tasks focus on exploration, presenting incomplete information of the game state and by carefully choosing actions, information is revealed:
>
> * **Battleship**: hidden displays of enemy ship locations
> * **Minesweeper**: hidden locations of the mine
> * **Wordle**: hidden word
>
> They are fairly representative of a broad class of partial / hidden information tasks that we want to investigate, where smarter actions chosen by the agent might not incur high immediate reward but will help gain higher long-term reward.
>
> > On this, "provable" still needs some scoping, e.g., only to the setting they formalized.
>
> Thank you for the suggestion! We will include explanations in the paper to make clear our intentions with the title. We also hope to clarify some potential misunderstanding: every logical proof is based on a premise – a premise is the setting that we formalized. Proof without a premise (aka the formalized setting) does not exist and is called an axiom (aka, self-evident). We follow a long tradition of machine learning papers where provably learning is used in the title:
>
> | Title                                                                        | Citation                                                                                                                                       |
> | ---------------------------------------------------------------------------- | ---------------------------------------------------------------------------------------------------------------------------------------------- |
> | Provably efficient learning with typed parametric models                     | Emma Brunskill, Bethany R. Leffler, Lihong Li, Michael L. Littman, Nicholas Roy; Journal of Machine Learning Research. 10(68):1955−1988, 2009. |
> | Provable learning of noisy-or networks                                       | Sanjeev Arora, Rong Ge, Tengyu Ma, Andrej Risteski. STOC 2017.                                                                                 |
> | Provably efficient reinforcement learning with linear function approximation | Chi Jin, Zhuoran Yang, Zhaoran Wang, and Michael I Jordan. In Conference on Learning Theory, pages 2137–2143. PMLR, 2020.                      |
>
> Our contributions focus on the idea of learnability – under what formal conditions is learning from natural language feedback possible and whether we can derive an upper bound on regret using recently developed theoretical tools in reinforcement learning. We will make this connection to prior literature clear in the related work and revise the title to “Formalizing Learning from Language Feedback with Provable Guarantees”.

---

> > ### Author Response · Authors · 2025-11-26
> >
> > > And lastly, there are many other pieces of related work that use language-based feedback… (SWEAgent, OpenHands, and the citation graph around these), as well as DSPy and TextGrad. All of these settings also iteratively improve a textual object based on text-based feedback in many cases. They are not mentioned in the submitted paper. I believe they would minimally be interesting to conceptually connect to.
> >
> > Thank you for the suggestion! Yes, The empirical successes of DSPy, TextGrad, and software agents like (SWEAgent, OpenHands) motivated us to write the paper. As you correctly point out, all of them focus on the paradigm of iteratively improving a textual object based on text feedback in many cases. This is a larger phenomenon / paradigm, and we are trying to formalize this with theoretical tools from RL. All your listed software packages can be regarded as specific instantiations / implementations of the learning from feedback paradigm. Our goal is largely orthogonal to their efforts of offering a specific implementation. We aim to suggest that the general family of hypothesis-elimination style algorithms (which have nice theoretical properties and guarantees, but often are studied in cases where rewards are numerical) can be re-purposed to work with language feedback. We offer one specific implementation in the paper (Algorithm 2) to show that our theoretical algorithm can be implemented.
> >
> > We will add this discussion in the related work.
> >
> > > My guess is that it may be difficult to get it exactly in the setting considered in the paper, but maybe they could be minimally be considered as baselines on the tasks in this paper?
> >
> > Thank you for the suggestion and understanding! The baseline we chose is closely mimicking what DSPy and TextGrad are doing when it comes to leveraging natural language feedback, where the LLM is given the task, solution, and feedback and prompted to propose a new solution (TextGrad, DSPy’s COPRO fall exactly under this paradigm). Some of DSPy’s optimizers (GEPA, MIPROv2) however leverage numerical reward information (aka task score) to do additional ranking and selection between solutions, which is beyond our setting.
> >
> > Beyond the algorithm side, we also offer a large amount of discussions on what constitutes good feedback. In previous literature, hand-wavy terms such as feedback is most useful when it contains “explanations of past mistakes”, “suggestions of future behaviors”. However, these are high-level intuitions. On line 333, the table under example 2, we are able to actually quantify what “explanation” and “suggestion” could mean, and how much they affect learning complexity of the problem. Using our framework, we successfully formalized intuitions around learning from feedback, and mapped words with semantic meanings to mathematical objects. This was not studied in DSPy, TextGrad, and OpenHands previously.
> >
> > We will include this discussion into the paper. Thank you for the suggestion!
> >
> > We hope that our responses have fully addressed your concerns and you will consider re-evaluating the merits of this paper.

---

> ### Comment · Reviewer_xnbi · 2025-11-26
>
> Thank you for the response! I will consider all of that throughout the rest of the process. Here are my quick thoughts inline:
>
> >> Slightly beyond this, why only select these three tasks, and not evaluate on all of the tasks from LLF-Bench?
> >
> > We picked the three tasks for demonstration purposes [... representing a] broad class of partial / hidden information tasks that we want to investigate
>
> My surprise here is because most of the paper doesn't seem scoped to just partial/hidden information tasks, e.g., this is not mentioned in the abstract/intro/motivation. So it seems like a disconnection to focus the experiments on these new LLF tasks while there is LLF-Bench with other tasks following the same interface. In other words, because it's an LLF paper, it seems reasonable to evaluate on LLF-Bench.
>
> > We follow a long tradition of machine learning papers where provably learning is used in the title
>
> I disagree with the tradition :) (I find "provable/y" potentially misleading and generic for outsiders to the subcommunity)
>
> But I acknowledge wanting to follow it, and the new title is clearer.
>
> >  Proof without a premise (aka the formalized setting) does not exist and is called an axiom
>
> 😂
>
> >  The baseline we chose is closely mimicking what DSPy and TextGrad are doing [...] TextGrad, DSPy’s COPRO fall exactly under this paradigm
>
> The biggest deficiency I still see on the experimental side of the paper is that despite being conceptually motivated and similar to TextGrad/DSPY/other agentic settings, there are no comparisons to their methods or tasks. To address this, I would recommend evaluating with exactly TextGrad and DSPy's COPRO as baselines, it would strengthen the comparison here. And the experiments would be even stronger if it's possible to use a previously-published task that uses TextGrad/COPRO.

---

> ### Author Response · Authors · 2025-11-26
>
> Hi,
>
> Thank you for the very quick response! We are actively working on the experiments and will add additional tasks from LLF-Bench, a theorem proving task, and comparisons to TextGrad / DSPy (COPRO) to the paper once the experiments finish running.

---

### Official Review · Reviewer_sdh9 · 2025-11-03

**Soundness:** 3
**Presentation:** 4
**Contribution:** 3
**Rating:** 8
**Confidence:** 3

**Summary:**

This paper studies a model of online learning with rich feedback called "learning from language feedback" (LLF), motivated by online decision making applications where the decision maker can receive language-based feedback from e.g., LLMs. In the proposed model, there is some ground truth hypothesis \eta^* unknown and is known to be in class H; and the learning agent repeatedly takes an action a_t and receives an observation o_t from the observation distribution f_{\eta^*}(a_t). Its goal is the maximize its cumulative reward \sum_{t=1}^T r_{\eta^*}(a_t), where the mapping r_{\eta}(a) is known ahead of time. The paper proposes a new concept called the ``transfer eluder dimension'' that measures the statistical complexity of the LLF problem. It further shows that:
- (Section 4.3) under some assumptions, LLF is no harder than learning from reward feedback (structured bandits)
- (Section 4.2) in some settings, LLF can be exponentially easier than learning from reward feedback

 It also proposes the HeLiX algorithm, which enjoys a sublinear regret, given that the transfer eluded dimension is finite. Experiments show that HeLiX can outperform baselines such as greedy.

**Strengths:**

- The paper provides a nice conceptual framework to understand the benefit of learning from language feedback, which is relevant in modern applications where LLM are good at providing language feedback beyond rewards

- A general statistical measure, transfer eluder dimension is proposed, that can handle general hypothesis classes.

- The initial set of theoretical results are somewhat complete, in the sense that it compares with reward-based feedback, as well as showing when it can be significantly more useful

**Weaknesses:**

- Assumption 3 seems important, although maybe necessary for clear development of theoretical results. Do you have a sense if this is satisfied in your experiments?

- (Clarity) Can the authors clarify what are some roadblocks in proving \sqrt{T} regret bound for HELiX under the general feedback setting (beyond the square loss assumption)?

- I agree with the final remark by the authors that there are some LLF problems with infinite transfer Eluder dimension and are trivially solvable. I wonder if we can show something like a converse of Theorem 1, e.g. for every d, there exists a class of hypotheses, observation mapping, and reward mapping with transfer Eluder dimension <= d, such that the regret of any algorithm is at least Omega( \sqrt{d} T^{3/4} ). A related question is, is O(T^{3/4}) regret bound in Theorem 1 fundamental to some loss \ell?

**Questions:**

- In Sec. 4.4 and Alg. 1, the \pi's are meant to be action a's?

- Without the consensus exploitation step, can HELiX still have a regret bound as in Theorem 1? That step seems rather nonstandard to me. In other words, why don't we just use a pure optimism-based algorithm?

---

> ### Author Response · Authors · 2025-11-26
>
> We thank the reviewer for their positive assessment and their insightful questions.
>
> > Assumption 3 seems important, although maybe necessary for clear development of theoretical results. Do you have a sense if this is satisfied in your experiments?
>
> We rely on trained SOTA LLMs to propose the most-likely $K$ hypotheses according to the observations so far. By doing this, we leverage LLMs’ capability to produce hypotheses that minimize $\ell(a,o,\eta)$ without explicitly computing this value. This also allows our algorithm to scale with future generations of LLMs that will have increasing abilities to produce loss-minimizing hypotheses.
>
> While we never explicitly implement or query $\ell$ in our practical implementation of HELiX (Algorithm 2), we use the information from $\ell$ implicitly through hypothesis updates. Directly checking for Assumption 3 would require access to $\ell(\cdot,\cdot,h)$ for all $h$, which is infeasible in practice.
>
> Instead, to give a sense of how Assumption 3 connects to our experiments, we evaluate a key implication of it: that the shrinking hypothesis set remains consistent with the true hypothesis along the trajectory. Concretely, if Assumption 3 approximately holds, then as the algorithm updates and prunes the hypothesis set, the true hypothesis should almost always remain included. To test this, we manually annotated 10 trajectories from the Battleship experiment. For each step in each trajectory, we checked whether the candidate set of hypotheses produced by HELiX contained the true hypothesis.
>
> The following table reports (i) the average fraction of steps within a trajectory for which the true hypothesis is included (“Assumption 3 satisfaction (trajectory-averaged)”), and (ii) the fraction of all annotated steps across trajectories for which the true hypothesis is included (“Assumption 3 satisfaction (step-averaged)”). Both metrics are above 95%, empirically supporting that Assumption 3 holds to a good approximation in our experiment.
>
> | Metric                                          | Value  |
> | ----------------------------------------------- | ------ |
> | Assumption 3 satisfaction (trajectory-averaged) | 96.26% |
> | Assumption 3 satisfaction (step-averaged)       | 95.98% |
>
> To make the connection between the theoretical and practical versions of HELiX clearer, we compare the two versions below in a line-by-line walkthrough:
>
> * **Line 5 of Algorithm 1 → line 5 of Algorithm 2:** Algorithm 1 maintains a hypothesis space $\mathcal{H}_t$ at iteration $t$, which contains all hypotheses $\eta$ that are consistent with observed feedback. Then Algorithm 1 searches over all possible policies by computing $\pi_p$ and $\pi_o$. Since it is impractical to keep a large hypothesis set and do an exhaustive search, we approximate this step in Algorithm 2 with sampling finite sets of candidates $\hat{\mathcal{H}}_t$ and $\hat{\mathcal{A}}_t$, respectively.
>
> * **Line 6 of Algorithm 2:** This step is purely motivated by observations that using a random reference policy to generate random actions provides empirical benefits.
>
> * **Lines 6–8 of Algorithm 1 → lines 7–11 of Algorithm 2:** Algorithm 1 carries out a minimax step to determine whether to stop exploring, when a policy is simultaneously optimal for all remaining hypotheses. To approximate this step, we use a thought cross-verify step to determine if some action $a$ is optimal across all sampled hypotheses.
>
> * **Lines 9–11 of Algorithm 1 → lines 12–14 of Algorithm 2:** Algorithm 1 computes a UCB-style exploration policy via optimism. This is approximated by eliminating hypotheses whose highest score is lower than those of other hypotheses in thought cross-verify.
>
> > Can the authors clarify what are some roadblocks in proving $\sqrt{T}$ regret bound for HELiX under the general feedback setting (beyond the square loss assumption)?
>
> Theorem 1 assumes only boundedness of $\ell$; without curvature or variance conditions, the standard confidence-set machinery yields $O(T^{3/4})$. When $\ell$ has some structure, we already proved $\tilde{O}(\sqrt{T})$ – see Theorem 4 (squared loss) in Appendix B.4.
>
> > I wonder if we can show something like a converse of Theorem 1…such that the regret of any algorithm is at least $\Omega(\sqrt{d} T^{3/4})$. A related question is, is $O(T^{3/4})$ regret bound in Theorem 1 fundamental to some loss $\ell$?
>
> We agree that it is an exciting direction for future work to derive such lower bounds. This exemplifies how our work offers avenues for future theoretical research in LLF. We hope that the formal framework proposed here would enable discussing questions about verifiers and how they affect maximal learning efficiency in the future.

---

> ### Author Response · Authors · 2025-11-26
>
> > In Sec. 4.4 and Alg. 1, the $\pi$'s are meant to be action $a$'s?
>
>
> No, $\pi$ denotes a policy (a distribution over actions) – Section 3.1. We slightly abused notation to use $r_\eta(\pi) = \sum_{a} r_\eta(a)\pi(a)$ to denote the expected reward under policy $\pi$. We will make this clear in Section 4.4 and Algorithm 1, sorry for the confusion.
>
>
> > Without the consensus exploitation step, can HELiX still have a regret bound as in Theorem 1? That step seems rather nonstandard to me. In other words, why don't we just use a pure optimism-based algorithm?
>
>
> The bound already holds without the consensus step; Appendix B.2 states that the full HELiX regret is upper-bounded by that of the version without exploitation, and Lemma 6 shows that if consensus occurs, subsequent per-step regret is zero. The consensus step is helpful in non-discriminative LLF where feedback can reveal an optimal action without revealing rewards (Remark D.1), avoiding needless exploration.

---

### Author Response · Authors · 2025-12-04
**Rebuttal Summary and Final Remarks**

We thank all the Reviewers and the Area Chair for their time and feedback throughout the review process. We are encouraged that the Reviewers recognized:
- the novelty and timeliness of our theoretical framework for learning from language feedback (LLF), which is well-motivated by growing empirical interest;
- the introduction of a new complexity measure, the transfer eluder dimension, which formalizes the intuition that LLF can be exponentially more sample-efficient than learning from rewards alone;
- and the proposal of an intuitive and implementable algorithm, HELiX, based on hypothesis elimination and verifier-guided exploration, which both enjoys general regret guarantees and demonstrates superior empirical performance on a range of text-based tasks.

In our rebuttal, we clarified several key points that we hope are helpful for the final assessment:
- **Verifier assumption**: Our verifier assumption can be naturally extended to **allow noise and small approximation error** in the verifier loss. In practice, **HELiX does not explicitly implement a verifier**; instead, it implicitly uses a verifier to **construct an approximation to the hypothesis set** by sampling several feedback-consistent hypotheses. While directly checking for Assumption 3 is infeasible, we annotated Battleship trajectories to verify an implication of Assumption 3, namely, that the candidate set of hypotheses produced by HELiX contains the true hypothesis. Our results show that in our experiments, **this implication holds in at least 95% of cases**.
- **Connection between theory and experiments**: The practical version of HELiX used in our experiments closely approximates the theoretical version while reducing computational cost to a feasible level. The experiments are intended primarily to **instantiate one concrete implementation of HELiX** and to illustrate the behavior of our theoretical framework in realistic text-based settings, whereas the main focus of the paper is to provide **a rigorous framework for mathematically analyzing LLF** and to **identify the formal quantities needed** to make an algorithm “provably learning” from language feedback.
- **Computational cost analysis**: We provide a detailed breakdown of the computational overhead of practical HELiX **in terms of both problem-dependent parameters and algorithmic hyperparameters**. In particular, we compute the token budget incurred by HELiX and show that it **scales gracefully relative to the baseline**.
- **Framing of contribution**: Our primary contribution is the first rigorous framework for LLF, together with principled assumptions under which such problems are tractable. To better reflect this emphasis, we have **revised the title to “Formalizing Learning from Language Feedback with Provable Guarantees”** and will further refine the framing in the related work section.
- **Baseline comparisons**: Our **baseline is designed to closely mirror how existing empirical work with language feedback typically leverages an LLM**: by providing the task, solution, and feedback, and prompting the model to generate an improved solution. HELiX can be layered on top of such approaches as an **orthogonal exploration module**. In parallel, we are **actively developing additional experiments** that more directly compare HELiX to these methods, in order to further substantiate its practical fidelity.

We appreciate the opportunity to engage in this discussion, which has helped us improve the clarity and impact of our work. We hope our responses address the Reviewers’ concerns and reinforce confidence in the significance and potential positive influence of this contribution.

---

### Meta-Review · Area_Chair_LM7t · 2026-01-03

**Summary:**

This paper proposes a rigorous theoretical framework for Learning from Language Feedback (LLF), introducing the transfer eluder dimension and a no-regret algorithm (HELiX), and was recognized by several reviewers as a timely and conceptually important contribution; however, substantial concerns remain regarding assumptions, theory–practice alignment, and empirical validation that must be addressed before acceptance. For a successful resubmission, I recommend the authors to address all reviewers suggestions below:
(1) clearly and explicitly scope the setting as one-step/bandit LLF in the title, abstract, and introduction, and avoid claims that suggest broader multi-step RL applicability;
(2) precisely clarify the conceptual relationship between the verifier, the verifier loss ℓ, LLM scoring, and the practical HELiX implementation, including an explicit explanation of when and how the LLM is functionally acting as a verifier;
(3) empirically quantify verifier quality and bias (e.g., unbiasedness diagnostics, approximation error Δ) and include the reported assumption-validation tables in the main paper;
(4) refine theoretical claims to avoid “optimality” language without matching lower bounds, and clearly distinguish general bounds from squared-loss special cases;
(5) strengthen experiments by adding regret plots (not just cumulative reward), expanding to additional LLF-Bench or agentic tasks, and including stronger and more relevant baselines (e.g., TextGrad, DSPy/COPRO, and reward-decoded UCB-style methods where appropriate);
(6) provide clearer ablations analyzing the impact of verifier fidelity, hypothesis set size, exploration components, and reference policies;
(7) report computational cost and scalability in a standardized manner (token counts, number of LLM calls, parallelization assumptions); and
Addressing these items would substantially strengthen the paper’s coherence, credibility, and impact for resubmission.

**Reviewer Concerns:**

I believe most reviewer comments will be addressed if the author delivered the promised edits (listed above). Unfortunately, until the date of my review (Jan 2nd, 2026) most of the promised edits were not found in the updated draft, so I cannot recommend acceptance.

**Reviewer Scores:**

I believe neither of the reviewers who gave score 2 would upgrade them all the way to acceptance.

---

### Decision · Program_Chairs · 2026-01-26

Reject